

# Malagasy *Conostigmus* (Hymenoptera: Ceraphronoidea) and the secret of scutes

István Mikó[1], Carolyn Trietsch[1],*, Emily L. Sandall[1],*,
Matthew Jon Yoder[2], Heather Hines[1,3] and Andrew Robert Deans[1]

[1] Frost Entomological Museum, Department of Entomology, Pennsylvania State University, University Park, PA, USA
[2] Illinois Natural History Survey, University of Illinois at Urbana-Champaign, Champaign, IL, USA
[3] Department of Biology, Pennsylvania State University, University Park, PA, USA
* These authors contributed equally to this work.

Corresponding author
István Mikó, izm2@psu.edu

## ABSTRACT

We revise the genus *Conostigmus Dahlbom* 1858 occurring in Madagascar, based on data from more specimens than were examined for the latest world revision of the genus. Our results yield new information about intraspecific variability and the nature of the atypical latitudinal diversity gradient (LDG) observed in Ceraphronoidea. We also investigate cellular processes that underlie body size polyphenism, by utilizing the correspondence between epidermal cells and scutes, polygonal units of leather-like microsculpture. Our results reveal that body size polyphenism in Megaspilidae is most likely related to cell number and not cell size variation, and that cell size differs between epithelial fields of the head and that of the mesosoma. Three species, *Conostigmus ballescoracas Dessart, 1997*, *C. babaiax* Dessart, 1996 and *C. longulus Dessart, 1997*, are redescribed. Females of *C. longulus* are described for the first time, as are nine new species: *C. bucephalus* Mikó and Trietsch sp. nov., *C. clavatus* Mikó and Trietsch sp. nov., *C. fianarantsoaensis* Mikó and Trietsch sp. nov., *C. lucidus* Mikó and Trietsch sp. nov., *C. macrocupula*, Mikó and Trietsch sp. nov., *C. madagascariensis* Mikó and Trietsch sp. nov., *C. missyhazenae* Mikó and Trietsch sp. nov., *C. pseudobabaiax* Mikó and Trietsch sp. nov., and *C. toliaraensis* Mikó and Trietsch sp. nov. A fully illustrated identification key for *Malagasy Conostigmus* species and a Web Ontology Language (OWL) representation of the taxonomic treatment, including specimen data, nomenclature, and phenotype descriptions, in both natural and formal languages, are provided.

## INTRODUCTION

With 162 extant species, *Conostigmus* Dahlbom, 1858 is the second most species-rich genus of Megaspilidae (Ceraphronoidea), a hymenopteran family showing a reverse latitudinal diversity gradient (LDG) in species richness (*Johnson & Musetti, 2004*; *Noyes, 1989*). Since ceraphronoid faunistic and taxonomic studies mostly focus on the
Holarctic fauna, it is possible that sample bias is the reason for this atypical distribution. This might be especially true for *Conostigmus*, given that the only taxonomic review of the genus focused exclusively on non-Nearctic, non-European species (n = 36) and was based on only 145 specimens (*Dessart, 1997*). The large number of Malagasy specimens examined in the present study will not only double the number of specimens of non-Nearctic, non-European *Conostigmus* but will also provide a reasonable data set for comparing the Malagasy fauna with that of a similarly sized area in Europe, the Atlantic Archipelago (British Isles; *Broad & Livermore, 2014*). Madagascar is considered a hotspot of biodiversity (*Myers et al., 2000*), and if reverse LDG is false and based on sample bias, we would be able to document more species in Madagascar than in the AA.

Ceraphronoids likely belong to the basal apocritan Evaniomorpha, and exhibit mostly ancestral sets of phenotypes (*Heraty et al., 2011*; *Vilhelmsen, Mikó & Krogmann, 2010*). The complexity of the ceraphronoid ovipositor system and male genitalia is unparalleled among Apocritans (*Mikó et al., 2013*; *Ernst, Mikó & Deans, 2013*), and the leather-like microsculpture covering their head and mesosoma (*Mikó, Yoder & Deans, 2011*; *Burks, Mikó & Deans, 2016*) is hypothesized to be an ancestral trait in Insecta (*Hinton, 1970*).

Besides the ten-fold interspecific body length variability from 0.37 mm (*Microceraphron subterraneus* Szelényi, 1935) to 4.5 mm, (*Megaspilus armatus* Say, 1836), up to four-fold intraspecific variability has been reported in Ceraphronoidea (*Mackauer & Chow, 2015*; *Dessart & Gärdenfors, 1985*; *Liebscher, 1972*). Similar intraspecific variability is common among microhymenoptera and can be stimulated by alternative host species with different nutritional quality (*Nalepa & Grisell, 1993*; *Medal & Smith, 2015*) gregariousness with variable brood sizes (*Harvey et al., 1998*), and climatic differences, such as temperature (*Wu et al., 2011*). Ceraphronoids are parasitoids on insect parasitoid and predator larvae (*Haviland, 1920*; *Withycombe, 1924*; *Kamal, 1939*) and have a broad host range (*Gilkeson, McLean & Dessart, 1993*; *Sullivan & Völkl, 1999*). *Dendrocerus carpenteri*, for example, has been reared from >70 aphidiine (Braconidae) species (*Fergusson, 1980*). Based on the few studies with appropriate rearing conditions, gregariousness might also be not uncommon among Ceraphronoidea (*Kamal, 1939*; *Bennett & Sullivan, 1978*; *Mackauer & Chow, 2015*; *Cooper & Dessart, 1975*; *Dessart, 1997*; *Liebscher, 1972*).

Environmental factors impact development and determine final adult body size (*Nijhout & Callier, 2015*) by altering different cellular processes. Temperature and oxygen level usually impact cell size (*Azevedo, French & Partridge, 2002*; *Harrison & Haddad, 2011*; *Heinrich et al., 2011*), while nutrition mostly regulates cell division (*Emlen, Lavine & Ewen-Campen, 2007*). *Conostigmus* species are relatively large ceraphronoids (0.8–2.2 mm) making it feasible to observe and examine scutes (*Meyer, 1842*; *Cals, 1974*; *Krell, 1994*), elements of the aforementioned leather-like sculpture. Scutes likely have a one-to-one correspondence to epidermal cells in arthropods (*Moretto, Minelli & Fusco, 2015*; *Hinton, 1970*; *Cals, 1973*; *Cals, 1974*; *Blaney & Chapman, 1969*) and thus should provide information about the cellular basis of body size polyphenism.

Research related to geographic distribution and polyphenism requires a stable taxonomic framework. We revise the Malagasy *Conostigmus*, Dahlbom, 1858 and use this system to explore the apparently anomalous ceraphronoid diversity patterns and possible reasons for body size polyphenism.

## MATERIALS AND METHODS

Specimens for the present study (Table S1) were obtained from Malaise trap samples and were loaned to the authors from the California Academy of Sciences (CAS). Morphological characters were scored with an Olympus SZX16 stereo-microscope equipped with an Olympus SDF PLAPO 2XPFC objective, resulting in 230× magnification. Specimens are deposited in the CAS, in the Frost Entomological Museum (FEM) and in the Royal Museum of Central Africa (MRAC) (Table S1).

Brightfield images of dried specimens were taken with an Olympus BX43 compound microscope equipped with an Olympus DP73 digital camera. Image stacking was performed with Zerene Stacker (Version 1.04 Build T201404082055; Zerene Systems LLC, Richland, WA, USA) and extended focus images were annotated and modified with Adobe Photoshop 6™ (Adobe Systems, San Jose, CA, USA) using Adjust/Filter/Unsharp mask and Image/Adjustments/Exposure (Gamma correction) tools.

Metasomata were removed from the specimens and placed in 35% $H_2O_2$ for 20 min, rinsed in distilled water for 30 min and dehydrated with 25 and 50% ethanol for 15–15 min, then transferred to a glycerol droplet on a concavity slide (Sail Brand Ltd., West Yorkshire, UK) and dissected. This protocol preserves muscle tissue while bleaching melanized structures, making them transparent for confocal laser scanning microscopy (CLSM).

Sample preparation for CLSM followed *Mikó & Deans (2013)*: male genitalia were temporarily mounted between two coverslips (1.5 μm, 22 × 60) in a glycerin droplet, which did not reach the edge of the coverslip. We used Blu-tack (Bostik, Wauwatosa, WI, USA) as spacer as this material does not interact with glycerol and provides an adjustable, appropriate distance between the coverslips. Specimens were examined with an Olympus FV10i desktop CLSM using a 60X objective.

Soft and sclerotized anatomical structures in arthropods tend to fluoresce with different intensities at different wavelength intervals (*Mikó & Deans, 2013*). CLSM tissue-specific contrast is gained by exciting specimens using multiple excitation wavelengths and/or recording the fluorescence on multiple channels assigned to different laser wavelength intervals. In previous research (*Mikó et al., 2013*; *Popovici et al., 2014*; *Ernst, Mikó & Deans, 2013*), specimens were excited with only one blue laser (480 nm) and the auto-fluorescence was detected with two channels (500–580 and 580–800 nm). Although the resulting micrographs had differences in their intensity patterns, data from the two channels largely overlapped. In the present paper, we use two different lasers (631 and 499 nm) and set filters (644 and 520 nm, respectively; narrow green and narrow red presets in Olympus Fluoview viewer software version 4.2) with narrow wavelength windows that result in a much higher tissue-specific contrast, almost perfectly separating muscle tissue and skeletal components (Fig. 1).

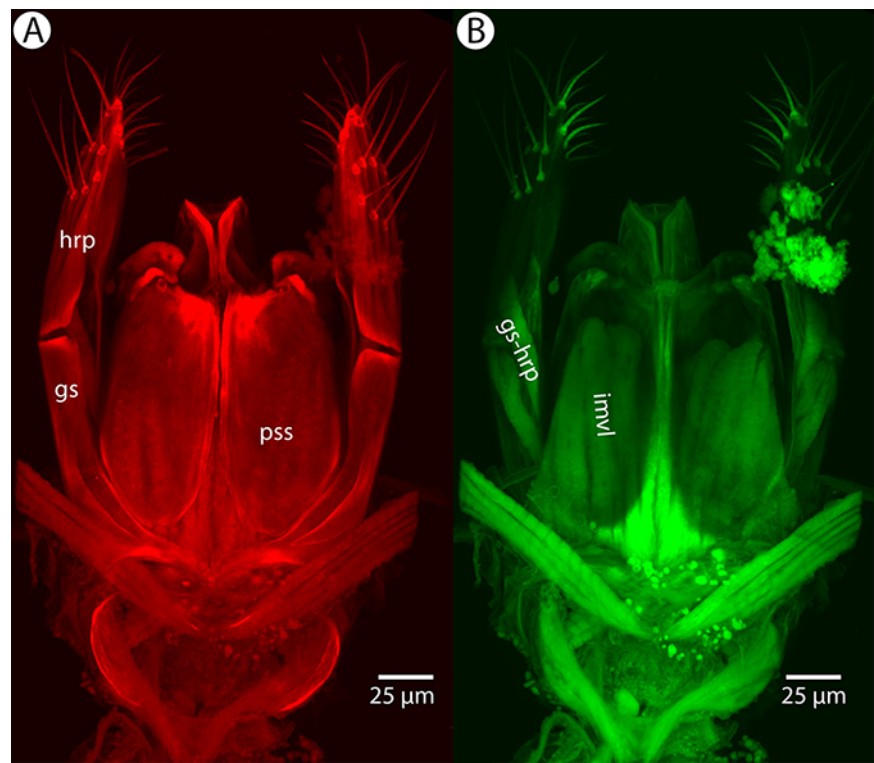

**Figure 1 CLSM volume rendered micrographs showing the skeletomuscular system of the male genitalia of *Conostigmus longulus Dessart, 1997*.** (A) Excitation wavelength = 631 nm, emission wavelength = 647 nm. (B) Excitation wavelength = 499 nm, emission wavelength = 520 nm. Abbreviations: pss, parossiculus; gs, gonostipes; hrp, harpe; gs-hrp, proximal gonostyle/volsella complex-harpal muscle; imvl, medial gonostyle volsella complex-volsellal muscle.

For the morphometric analysis on scute patterns, extended focal images of the frons and the mesocutellar-axillar complex of 14 *Conostigmus longulus Dessart, 1997* specimens were taken using an Olympus BX43 compound microscope equipped with an Olympus DP73 digital camera on 200× magnification. Extended focal images were generated using the online "extended focal imaging" (efi) tool of an Olympus Cellsens™ software. Measurements (Table S2) were taken using the same software. First, a 9,636 μm²; rectangular area was assigned on the extended focal images for recording scute pattern. The lateral vertices of the medially-positioned rectangle were adjacent with the scutoscutellar sulcus on the mesonotum, while the rectangle was positioned medially on the frons with equal distance from the anterior ocellus and the intertorular carina. Scutes overlapping this area (including scutes adjacent to the margin of the rectangle) were counted and the longest diameter of each scute was measured. Measurements were taken on the images while constantly checking their accuracy on live view at 200–500× magnification (Figs. 2A and 2B). Body length largely depends on the relative orientations of the tagmata. The head of most species is flattened dorsoventrally and attached at its posterior end to the thorax (compare the position of the head on Figs. 31A and 31B). We used the interorbital distance (IOS, http://purl.obolibrary.org/obo/HAO_0000432) to infer body size in our statistical analysis.

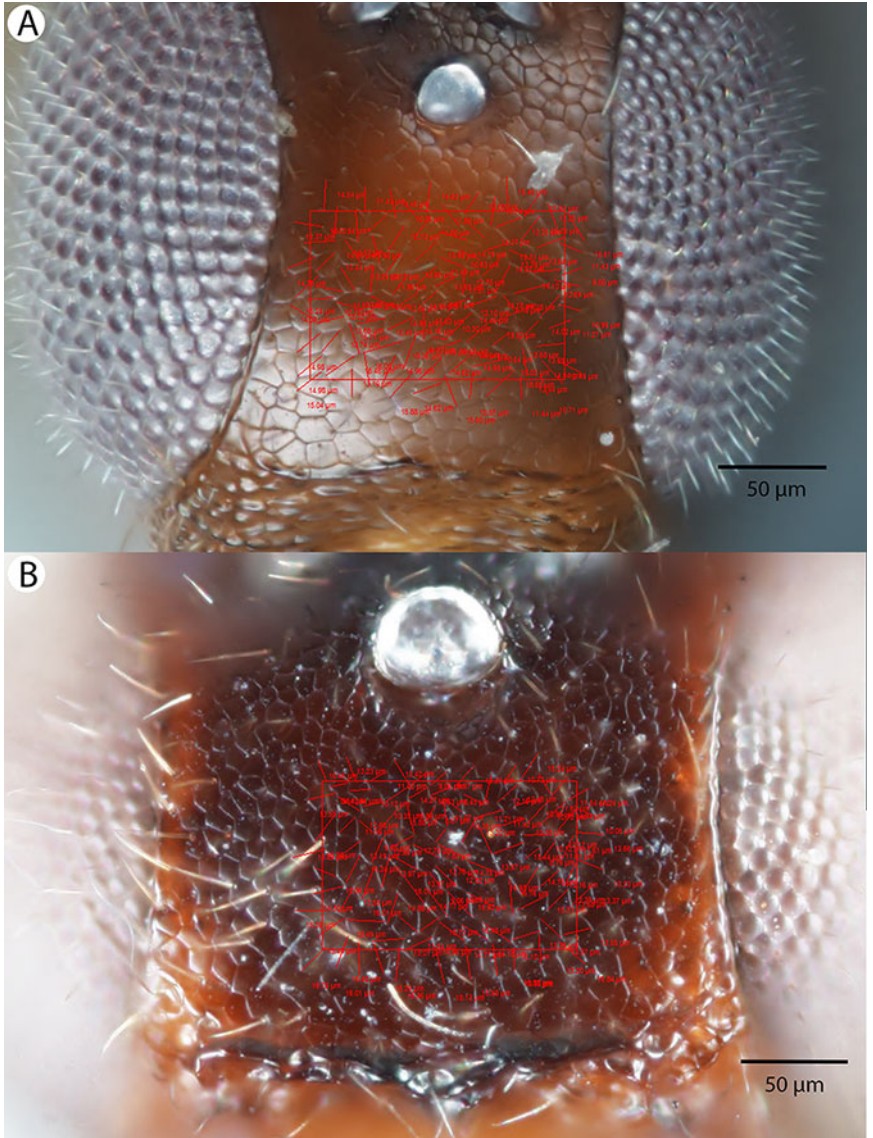

**Figure 2 Brightfield image showing the median region of the cranium of *Conostigmus longulus Dessart, 1997*, anterior view.** The measured scute lengths and values and borders of measured rectangle are annotated in red color. Size of rectangular area = 9,636 $\mu m^2$.

Volume rendered CLSM micrographs and media files and scaleable vector (.msi) annotated extended focal images of the frons and mesoscutum, complete with scute measurements, IOS and rectangles can be accessed from Figshare (http://dx.doi.org/10.6084/m9.figshare.3848544; http://dx.doi.org/10.6084/m9.figshare.3848547.v1).

Abbreviations of anatomical structures used in Figures are listed in Table S3.

Intraspecific variation in scute diameter and relative size was scrutinized by linear regression analyses. Intra-individual variation was scrutinized via Mann-Whitney sum-rank test (*Mann & Whitney, 1947*). The relationship between morphometric variables was inspected through linear regression analyses. Statistical analyses were carried out and Boxplots were generated in R version 3.2.2 (*R Development Core Team, 2015*) (Fig. S1).

**Table 1 Mesoscutellar average scute size, length and scute number in *Conostigmus longulus Dessart, 1997*.**

| Mesoscutellum | ID | aCS | mCL | nS |
|---|---|---|---|---|
| | 2041918 | 119.02 | 17.13 | 81 |
| | 2003474 | 109.56 | 17.22 | 88 |
| | 2044193 | 107.12 | 14.52 | 90 |
| | 2044825 | 101.48 | 15.19 | 95 |
| | 2040900 | 101.48 | 16.01 | 95 |
| | 2009756 | 101.48 | 16.01 | 95 |
| | 2002193 | 101.48 | 16.01 | 95 |
| | 2044755 | 94.52 | 14.09 | 102 |
| | 2040771 | 91.82 | 13.02 | 105 |
| | 2053554 | 90.95 | 13.635 | 106 |
| Median | | 101.48 | 15.60 | 95 |
| Mean | | 101.89 | 15.28 | 95.2 |
| SD | | 2.188891 | 1.437595 | 7.743097 |

Note:
ID, CASENT identifier for specimen; aCS, average cell size in $\mu m^2$; mCL, median cell length in $\mu m$; nS, number of scutes (cells) in a rectangular area of 9,636 $\mu m^2$.

**Table 2 Frontal average scute size, length and scute number in *Conostigmus longulus Dessart, 1997*.**

| ID | IOS | aCS | mCL | nS |
|---|---|---|---|---|
| 2003474 | 182.91 | 97.38 | 12.78 | 99 |
| 2009756 | 157.42 | 98.38 | 13.655 | 98 |
| 2040771 | 174.12 | 98.38 | 12.195 | 98 |
| 2040900 | 171.04 | 98.38 | 13.12 | 98 |
| 2002193 | 251.06 | 133.90 | 16.195 | 72 |
| 2053688 | 195.22 | 100.43 | 13.695 | 96 |
| 2053554 | 189.95 | 97.38 | 13.55 | 99 |
| 2053308 | 175.44 | 99.39 | 12.32 | 97 |
| 2046100 | 208.85 | 93.60 | 11.43 | 103 |
| 2041918 | 213.25 | 95.46 | 12.68 | 101 |
| 2044193 | 263.82 | 95.46 | 13.37 | 101 |
| 2044511 | 138.51 | 93.60 | 13.02 | 103 |
| 2044755 | 171.04 | 97.38 | 12.75 | 99 |
| 2044825 | 255.02 | 94.52 | 12.82 | 102 |
| Median | 186.4 | 97.38 | 12.92 | 99 |
| Mean | 196.3 | 99.55 | 13.11 | 97.57 |
| SD | 37.95319 | 10.10979 | 1.081914 | 7.673273 |

Note:
ID, CASENT identifier for specimen; IOS, interorbital space (referring to body size) in $\mu m$; aCS, average scute (cell) size in $\mu m^2$; mCL: median cell length in $\mu m$; nS, number of scutes (cells) in a rectangular area of 9,636 $\mu m^2$. Note the outlier 2002193 with significantly larger average cell size and median cell length as well as a lower number of scutes.

Taxonomic nomenclature, specimen information, OTU concepts, original brightfield images and natural language (NL) phenotype representations were compiled in mx (http://purl.org/NET/mx-database). Taxonomic history, description,

and material examined sections (Table S1) were rendered from the same software. Terminology of the phenotype statements used in descriptions, identification key and diagnoses are mapped to the Hymenoptera Anatomy Ontology (HAO, available at http://hymao.org), Phenotypic Quality Ontology (PATO, available at http://purl.obolibrary.org/obo/pato.owl), Biospatial Ontology (BSPO, available at http://purl.obolibrary.org/obo/bspo.owl) and Common Anatomy Reference Ontology (CARO, available at http://obofoundry.org/).

Natural language phenotypes are represented in "Entity attribute: value" format. Semantic statements for phenotype descriptions were created in Protégé 5.0.0-beta-16 (http://protege.stanford.edu/) using the Web Ontology Language (OWL) Manchester syntax (http://www.w3.org/TR/owl2-manchester-syntax/) following *Balhoff et al. (2013)*, *Mikó et al. (2015)* and *Mikó et al. (2014)* (Table S4). The OWL (http://www.w3.org/TR/owl2-overview/; accessed 4 February 2014) representation of the full data set was deposited as a single Resource Description Framework (RDF)-XML file (http://www.w3.org/TR/REC-rdf-syntax/; accessed 12 March) in the Github repository (https://github.com/hymao/hymao-data/blob/master/miko2016_malagasy.owl).

The electronic version of this article in Portable Document Format (PDF) will represent a published work according to the International Commission on Zoological Nomenclature (ICZN), and hence the new names contained in the electronic version are effectively published under that Code from the electronic edition alone. This published work and the nomenclatural acts it contains have been registered in ZooBank, the online registration system for the ICZN. The ZooBank LSIDs (Life Science Identifiers) can be resolved and the associated information viewed through any standard web browser by appending the LSID to the prefix http://zoobank.org/. The LSID for this publication is: LSID: urn:lsid:zoobank.org:pub:41133330-D364-4ABF-AFB3-DEE013E0EDF9. The online version of this work is archived and available from the following digital repositories: PeerJ, PubMed Central and CLOCKSS.

## RESULTS

### Body size polyphenism in *Conostigmus longulus* Mikó and Trietsch sp. nov.

#### *Intraspecific variation in scute diameter and relative scute size*

Fourteen specimens were measured and are represented in analysis of frons measurement distribution. Ten of these specimens were also measured on the mesoscutellum and make up the mesoscutellar analysis. Four specimens could not be measured for both regions due to inaccessibility of all scutes required for mesoscutellar measurements (*i.e.*, the specimen preparation obscured these parts). One specimen was found to have measurements four standard deviations from the mean, and was removed from subsequent analyses. Removal of this influential point from linear regression and further statistical analyses was justified by the reduction in statistical power caused by its inclusion in our small sample size (*Osborne & Overbay, 2008*).

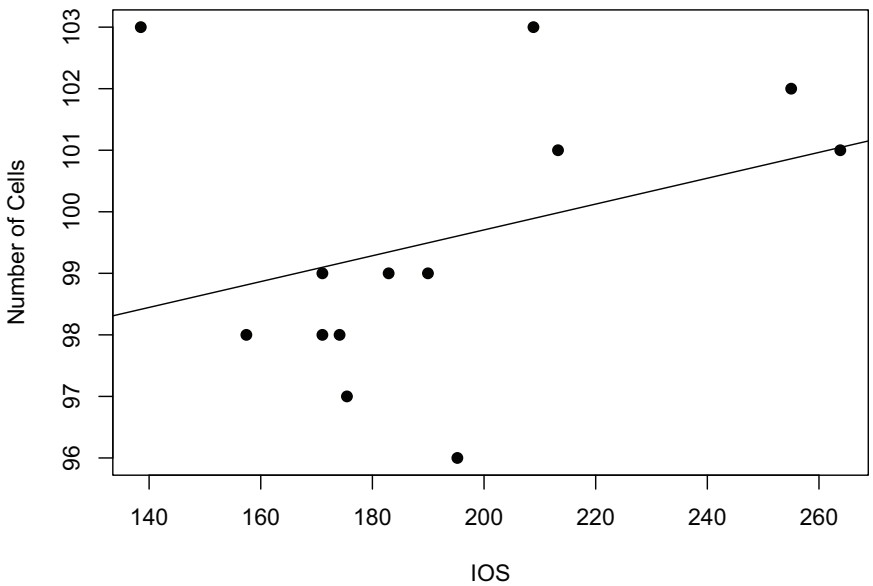

**Figure 3 Relationship between body size and number of cells as linear regression.** Interorbital space length (IOS), measured in µm, stands as a proxy for body size. Number of cells refers to the number of scutes/cells of a standard sized rectangular area. The size of these cells is not correlated to significant body size differences (Fig. S4).

Measurements were tested for normalcy and found to not follow the normal distribution, even after removal of the outlier from analyses. The scutes in a 9,636 µm$^2$ rectangular region of the frons and mesoscutellum were counted and measured. It was found that maximum scute length varied from 6.6 to 19.5 µm. on the frons (Table 1). On the mesoscutellum, maximum scute length varied from 8.8 to 23.4 µm (Table 2). Median cell length of each specimen was used as a variable in statistical analyses. Linear regression analyses were carried out independently on both the frons and mesoscutellar fields. There was a weak negative correlation between median scute length and scute number in the frons region ($R^2$ = 0.1369023). In the mesoscutellar region, scute length had a stronger negative correlation with scute number ($R^2$ = 0.7149943).

### Intraindividual variation in scute size: frons vs. mesoscutellum

In individuals where average scute size and diameter were measured on both the frons and the mesoscutellum, we found a variation in median cell length ranging from 11.43 to 17.22 µm. Wilcoxon rank-sum test revealed a significant difference in cell length between the frons and mesoscutellar regions (p-value = 0.0004011). Measurements of average scute size varied from 90.95 to 119.02 µm$^2$. There was no significant difference in average scute size between the frons and mesoscutellar regions in this sample when analyzed by Wilcoxon rank-sum test (p-value = 0.0809).

### Body size vs. scute size

Interorbital space was used as a measure of body size for all 14 specimens examined. Linear regression analysis for the relationship between scute size and body size was carried

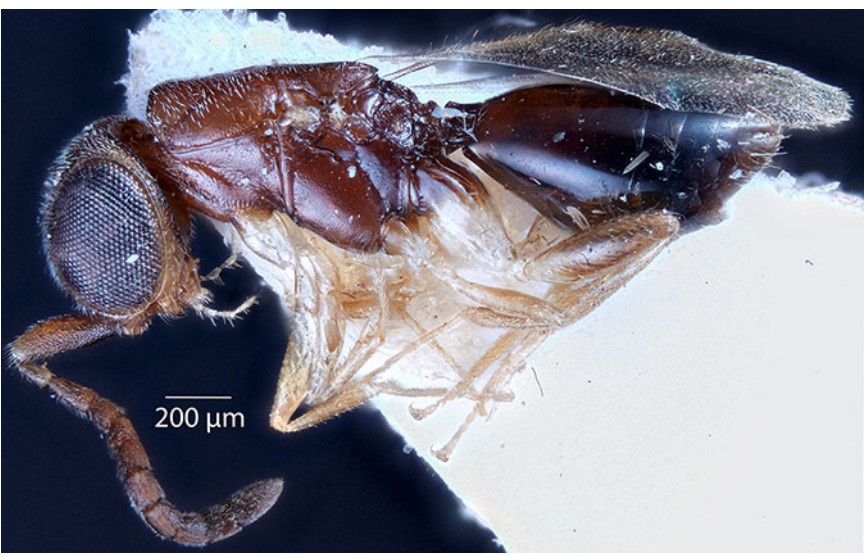

**Figure 4 Brightfield image showing the lateral habitus of *Conostigmus babaiax* Dessart, 1997.**

out using measurements for interorbital space and the measurements of average frons scute size, which were available for all measured specimens (Fig. 3). Correlation between median cell length and IOS was extremely weak and negative ($R^2 = 0.001341984$). The correlation between cell number and IOS was much stronger and weakly positive ($R^2 = 0.1114898$).

## Taxonomic treatment of Malagasy *Conostigmus*

*Conostigmus* Dahlbom, 1858

**Diagnosis:** It is not possible to provide a diagnosis for all megaspiline species that are currently classified as *Conostigmus*. The present treatment includes those Malagasi megaspilid specimens that have a sternaulus, an OOL longer than POL, a mesosoma that is higher than wide and lacks a bifurcated anteromedian projection of the metanoto-propodeo-metapecto-mesopectal complex, a harpe, which is independent from the gonostyle and/or whose parossiculi are discontinuous medially and are independent from the gonostyle. *Megaspilus* has bifurcated anteromedian projection of the metanoto-propodeo-metapecto-mesopectal complex; *Dendrocerus* Ratzeburg and *Creator* Alekseev has OOL shorter than POL and medially continuous parossiculi that are continuous with gonostyle and lack sternaulus; *Trichosteresis* Förster has a harpe that is fused with the gonostyle and the mesosoma is wider than high in *Platyceraphron* Kieffer.

*Conostigmus babaiax* Dessart, 1997
Figures 4, 5 and 6

**Diagnosis:** *Conostigmus babaiax* Dessart, 1996 shares the presence of a prognathous head (dorsal-most point of occipital carina is dorsal to posterior ocellus in lateral view) and the presence of transverse scutes on the ventral region of frons with *Conostigmus longulus*

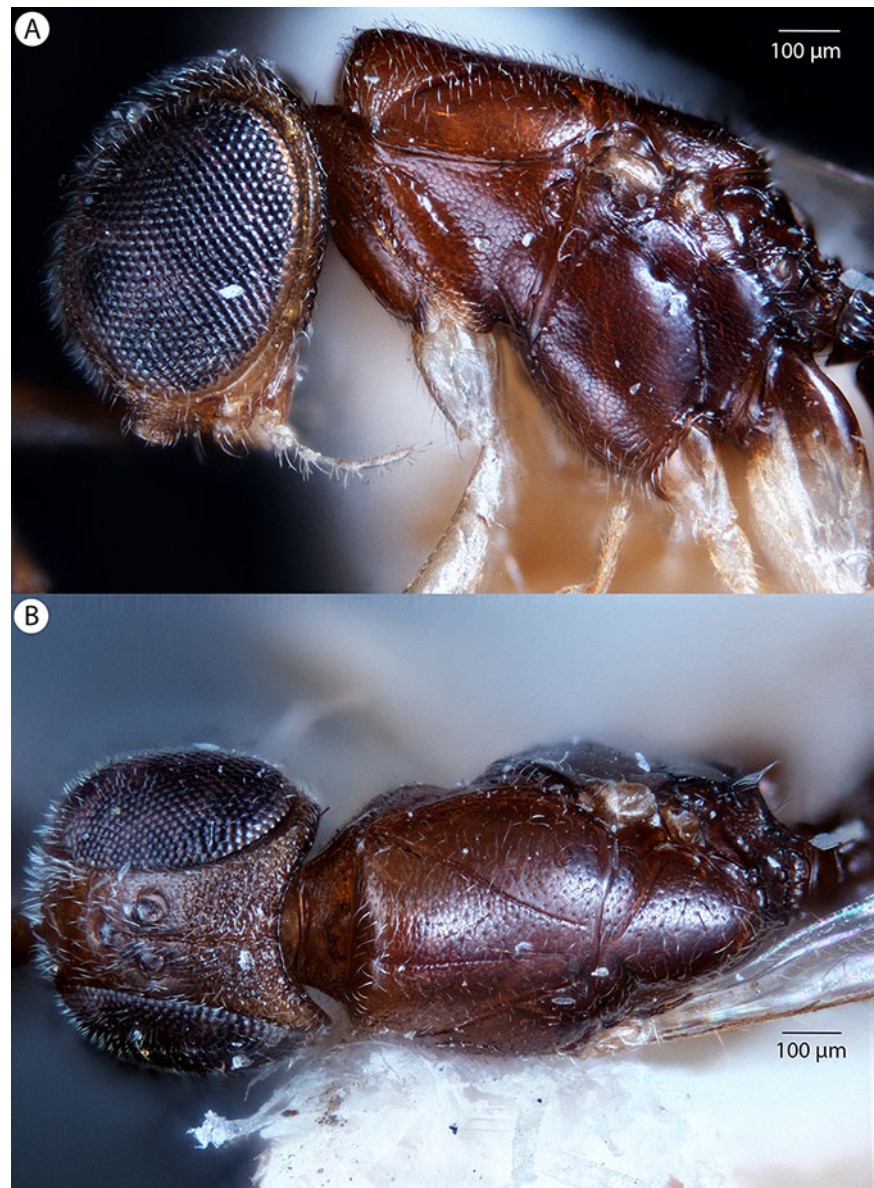

**Figure 5** Brightfield image showing the head and mesosoma of *Conostigmus babaiax Dessart, 1997*. (A) Lateral view. (B) Dorsal view.

*Dessart, 1997*, *C. toliaraensis* sp. nov. and *C. pseudobabaiax* sp. nov. *Conostigmus babaiax, C. toliaraensis* sp. nov. and *C. pseudobabaiax* differ from all other *Conostigmus* species by the presence of ventromedian and ventrolateral white, setiferous patches on the frons. The lateral ocellar line (LOL) is longer than ocular ocellar line (OOL) in *Conostigmus babaiax* and the LOL is shorter than OOL in *C. toliaraensis* sp. nov. and *C. pseudobabaiax* sp. nov.

**Description:** Body length: 2,200 μm. Color intensity pattern: NOT CODED. Color hue pattern: scape, pedicel, F1–F3, head, anterior mesosoma ochre, F4–F9, posterior mesosoma, metasoma brown, legs except darker proximal regions of meso and metacoxae yellow. Occipital carina sculpture: smooth. Median flange of occipital carina count:

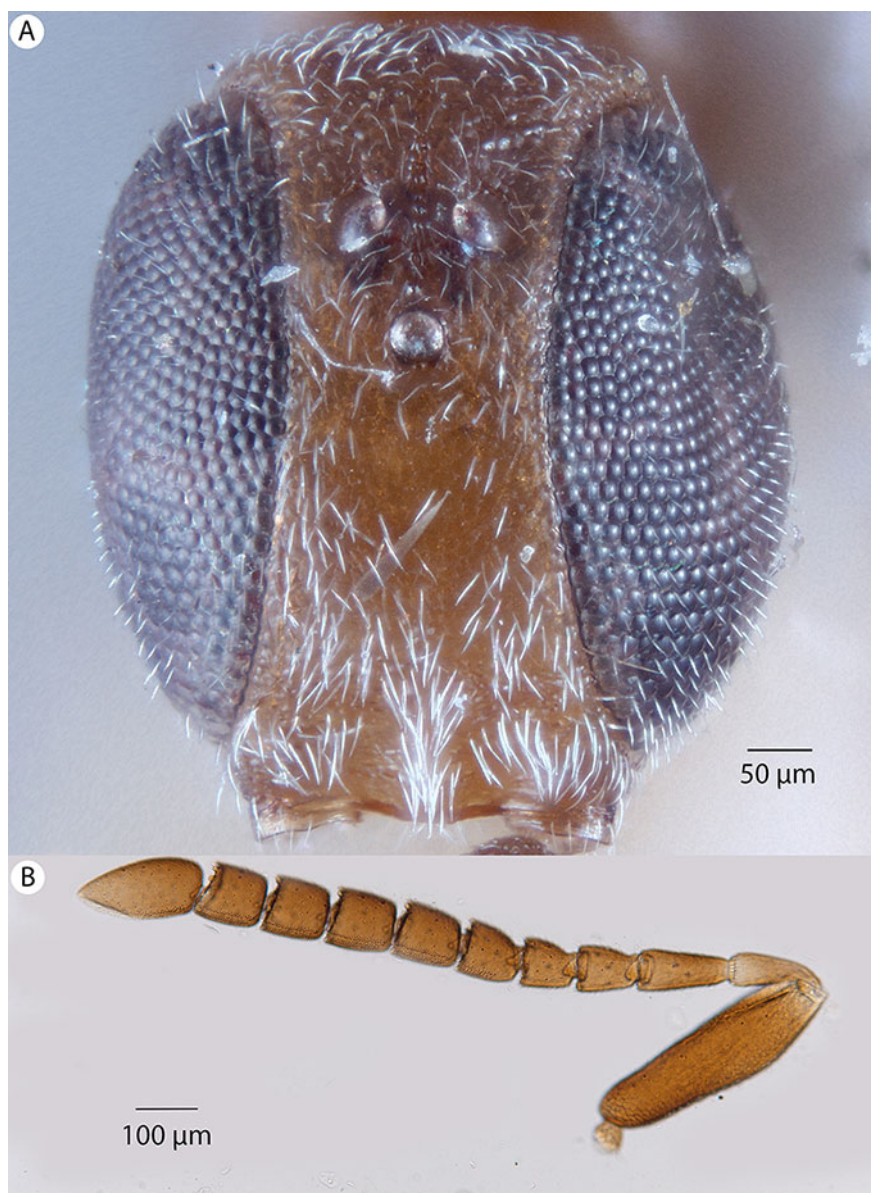

**Figure 6 Brightfield image showing the head and female antenna of *Conostigmus babaiax Dessart, 1997*.** (A) Head, anterior view. (B) Female antenna.

absent. Submedial flange of occipital carina count: absent. Dorsal margin of occipital carina vs. dorsal margin of lateral ocellus in lateral view: occipital carina is dorsal to lateral ocellus in lateral view. Preoccipital lunula count: NOT CODED. Preoccipital carina count: absent. Preoccipital carina shape: NOT CODED. Preoccipital furrow count: present. Preoccipital furrow anterior end: Preoccipital furrow ends inside ocellar triangle. Postocellar carina count: absent. Male OOL: posterior ocellar line (POL): LOL: NOT CODED. Female OOL: POL: LOL: 0.85:0.85:1.00. Head width vs. interorbital space (HW/IOS) Male: NOT CODED. HW/IOS Female: 2.65. Setal pit on vertex size: smaller than diameter of scutes. Transverse frontal carina count: absent. Transverse scutes on frons count: present. Rugose region on frons count: absent. Randomly sized areolae around

setal pits on frons count: absent. Antennal scrobe count: absent. Ventromedian setiferous patch and ventrolateral setiferous patch count: present. Facial pit count: no external corresponding structure present. Supraclypeal depression count: absent. Supraclypeal depression structure: NOT CODED. Intertorular carina count: present. Intertorular area count: present. Median region of intertorular area shape: flat. Ventral margin of antennal rim vs. dorsal margin of clypeus: not adjacent. Torulo-clypeal carina count: absent. Subtorular carina count: absent. Mandibular tooth count: 2. Female flagellomere one length vs. pedicel: 1.09. Female ninth flagellomere length: F9 less than F7 + F8. Sensillar patch of the male flagellomere pattern: NOT CODED. Length of setae on male flagellomere vs. male flagellomere width: NOT CODED. Male flagellomere one length vs. male second flagellomere length: NOT CODED. Male flagellomere one length vs. pedicel length: NOT CODED. Ventrolateral invagination of the pronotum count: present. Scutes on posterior region of mesoscutum and dorsal region of mesoscutellum convexity: flat. Notaulus posterior end location: adjacent to transscutal articulation. Median mesoscutal sulcus posterior end: not adjacent to transscutal articulation (ends anterior to transscutal articulation). Scutoscutellar sulcus vs. transscutal articulation: adjacent. Axillular carina count: absent. Axillular carina shape: NOT CODED. Epicnemium posterior margin shape: anterior discrimenal pit absent; epicnemial carina interrupted medially. Epicnemial carina count: present only laterally. Sternaulus count: absent. Sternaulus length: NOT CODED. Speculum ventral limit: not extending ventrally of pleural pit line. Mesometapleural sulcus count: present. Metapleural carina count: present. Transverse line of the metanotum-propodeum vs. antecostal sulcus of the first abdominal tergum: adjacent sublaterally. Lateral propodeal carina count: present. Lateral propodeal carina shape: inverted "U" (left and right lateral propodeal carina are adjacent to the antecostal sulcus of the first abdominal tergumsubmedially). Anteromedian projection of the metanoto-propodeo-metapecto-mesopectal complex count: absent. S1 length vs. shortest width: S1 wider than long. Transverse carina on petiole shape: concave. Distal margin of male S9 shape: NOT CODED. Proximolateral corner of male S9 shape: NOT CODED. Cupula length vs. gonostyle-volsella complex length: NOT CODED. Proximodorsal notch of cupula count: NOT CODED. Proximodorsal notch of cupula shape: NOT CODED. Proximolateral projection of the cupula shape: NOT CODED. Proximodorsal notch of cupula width vs. length: NOT CODED. Distodorsal margin of cupula shape: NOT CODED. Dorsomedian conjunctiva of the gonostyle-volsella complex length relative to length of gonostyle-volsella complex: NOT CODED. Dorsomedian conjunctiva of the gonostyle-volsella complex count: NOT CODED. Distal end of dorsomedian conjunctiva of the gonostyle-volsella complex shape: NOT CODED. Parossiculus count (parossiculus and gonostipes fusion): NOT CODED. Apical parossicular seta number: NOT CODED. Distal projection of the parossiculus count: NOT CODED. Distal projection of the penisvalva count: NOT CODED. Dorsal apodeme of penisvalva count: NOT CODED. Harpe length: NOT CODED. Distodorsal setae of sensillar ring of harpe length vs. harpe width in lateral view: NOT CODED. Distodorsal setae of sensillar ring of harpe orientation: NOT CODED. Sensillar ring area of harpe orientation: NOT CODED. Lateral setae of harpe count: NOT CODED. Lateral setae of

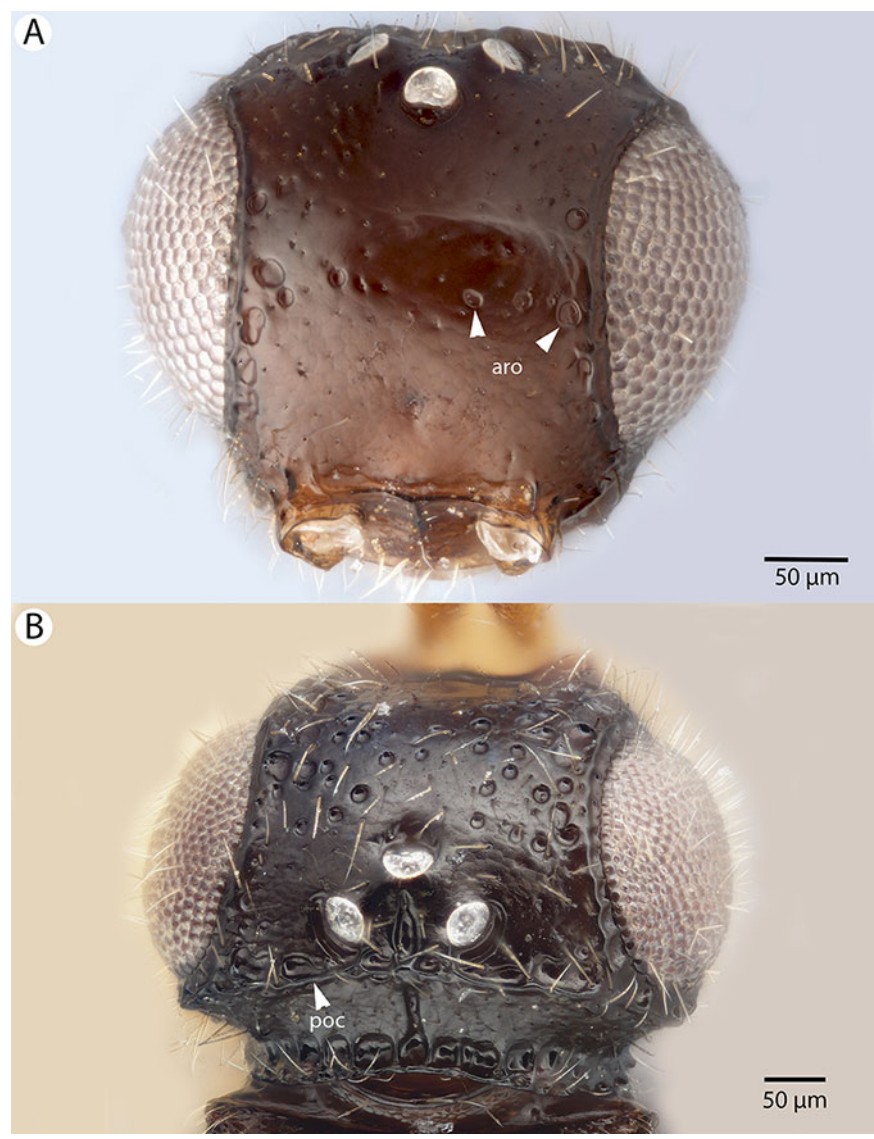

**Figure 7 Brightfield image showing the head of *Conostigmus ballescoracas Dessart, 1997*.** (A) Anterior view. (B) Dorsal view (ar, randomly sized areolae around setal pits; poc, preoccipital carina).

harpe orientation: NOT CODED. Distal margin of harpe in lateral view: shape: NOT CODED. Lateral margin of harpe shape: NOT CODED.

**Material examined:** Holotype female: MADAGASCAR: PSUC_FEM 000006723, COLL. MUS. Congo Madagascar: Mandraka II-1944 A. Seyrig HOLOTYPUS Holotype Prep. micros-copique n 9508/051 (deposited in MRAC).

*Conostigmus ballescoracas Dessart, 1997*

Figures 7, 8 and 9

**Diagnosis:** *Conostigmus ballescoracas Dessart, 1997* differs from other *Conostigmus* species by the presence of a strong preoccipital carina that is continuous with the orbital carina,

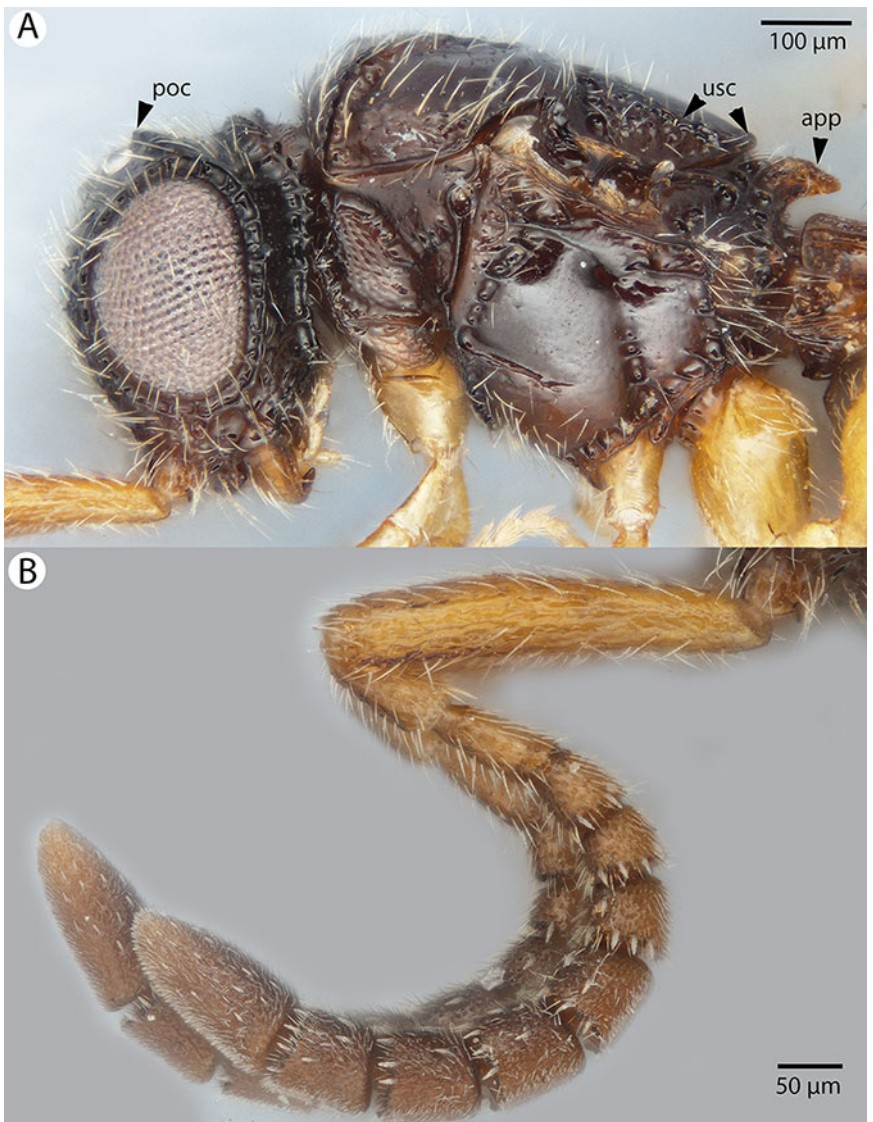

**Figure 8 Brightfield image of *Conostigmus ballescoracas* Dessart, 1997.** (A) Head and mesosoma lateral view. (B) Female antenna lateral view (poc, preoccipital carina; app, anteromedian projection of the metanoto-propodeo-metapecto-mesopectal complex; usc, u-shaped carina surrounding posteriorly and laterally the disc of the mesoscutellum).

the presence of randomly sized areolae around the setal pits on the frons (shared with the ceraphronid *Masner lubomirus* Mikó & Deans, 2013) and the posteromedially adjacent axillular carinae (left and right axillar carinae continuous posteriorly forming a U-shaped carina that surrounds the mesoscutellar disc).

**Description:** Body length: 1,650–1,875 µm. Color intensity pattern: NOT CODED. Color hue pattern: Cranium black; mesosoma, metasoma, F4–F9 brown; rest of antenna, legs and mandible ochre; Cranium brown; mesosoma except legs, metasoma ochre; F4–F9 brown; rest of antenna ochre, legs yellow. Occipital carina sculpture: crenulate. Median flange of occipital carina count: absent. Submedial flange of occipital

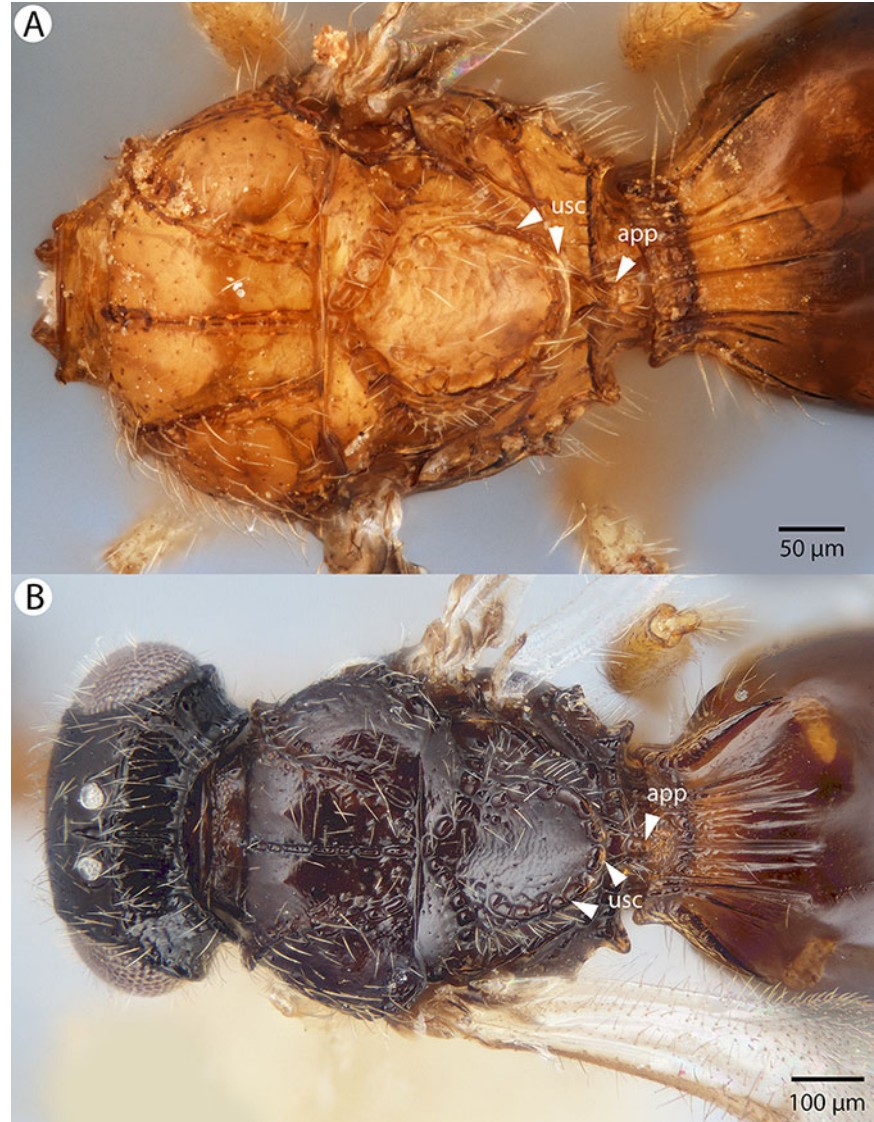

**Figure 9 Brightfield image of *Conostigmus ballescoracas Dessart, 1997*.** (A) Mesosoma and anterior metasoma, dorsal view. (B) Head, mesosoma and anterior metasoma, dorsal view (app, anteromedian projection of the metanoto-propodeo-metapecto-mesopectal complex; usc, u-shaped carina surrounding posteriorly and laterally the disc of the mesoscutellum).

carina count: absent. Dorsal margin of occipital carina vs. dorsal margin of lateral ocellus in lateral view: occipital carina is ventral to lateral ocellus in lateral view. Preoccipital lunula count: present. Preoccipital carina count: present. Preoccipital carina shape: complete. Preoccipital furrow count: present. Preoccipital furrow anterior end: Preoccipital furrow ends inside ocellar triangle. Postocellar carina count: absent. Male OOL: POL: LOL: NOT CODED. Female OOL: POL: LOL: 2.5–3.0:1.9–2.0:1.0. HW/IOS Male: NOT CODED. HW/IOS Female: 1.7–1.8. Setal pit on vertex size: smaller than diameter of scutes. Transverse frontal carina count: absent. Transverse scutes on frons count: absent. Rugose region on frons count: absent. Randomly sized areolae around setal pits on frons count: present. Antennal scrobe count: absent. Ventromedian setiferous

patch and ventrolateral setiferous patch count: absent. Facial pit count: facial pit present. Supraclypeal depression count: absent. Supraclypeal depression structure: NOT CODED. Intertorular carina count: present. Intertorular area count: present. Median region of intertorular area shape: convex. Ventral margin of antennal rim vs. dorsal margin of clypeus: adjacent. Torulo-clypeal carina count: absent. Subtorular carina count: absent. Mandibular tooth count: 2. Female flagellomere one length vs. pedicel: 0.8–0.9. Female ninth flagellomere length: F9 less than F7 + F8. Sensillar patch of the male flagellomere pattern: NOT CODED. Length of setae on male flagellomere vs. male flagellomere width: NOT CODED. Male flagellomere one length vs. male second flagellomere length: NOT CODED. Male flagellomere one length vs. pedicel length: NOT CODED. Ventrolateral invagination of the pronotum count: present. Scutes on posterior region of mesoscutum and dorsal region of mesoscutellum convexity: flat. Notaulus posterior end location: adjacent to transscutal articulation. Median mesoscutal sulcus posterior end: adjacent to transscutal articulation. Scutoscutellar sulcus vs. transscutal articulation: adjacent. Axillular carina count: present. Axillular carina shape: left and right carina continuous posteromedially forming a U-shape carina on the mesoscutellar axillar complex. Epicnemium posterior margin shape: anterior discrimenal pit present; epicnemial carina curved. Epicnemial carina count: complete. Sternaulus count: present. Sternaulus length: elongate, exceeding 3/4 of mesopleuron length at level of sternaulus. Speculum ventral limit: not extending ventrally of pleural pit line. Mesometapleural sulcus count: present. Metapleural carina count: present. Transverse line of the metanotum-propodeum vs. antecostal sulcus of the first abdominal tergum: adjacent sublaterally. Lateral propodeal carina count: present. Lateral propodeal carina shape: NOT CODED. Anteromedian projection of the metanoto-propodeo-metapecto-mesopectal complex count: present. S1 length vs. shortest width: S1 wider than long. Transverse carina on petiole shape: straight. Distal margin of male S9 shape: NOT CODED. Proximolateral corner of male S9 shape: NOT CODED. Cupula length vs. gonostyle-volsella complex length: NOT CODED. Proximodorsal notch of cupula count: NOT CODED. Proximodorsal notch of cupula shape: NOT CODED. Proximolateral projection of the cupula shape: NOT CODED. Proximodorsal notch of cupula width vs. length: NOT CODED. Distodorsal margin of cupula shape: NOT CODED. Dorsomedian conjunctiva of the gonostyle-volsella complex length relative to length of gonostyle-volsella complex: NOT CODED. Dorsomedian conjunctiva of the gonostyle-volsella complex count: NOT CODED. Distal end of dorsomedian conjunctiva of the gonostyle-volsella complex shape: NOT CODED. Parossiculus count (parossiculus and gonostipes fusion): NOT CODED. Apical parossicular seta number: NOT CODED. Distal projection of the parossiculus count: NOT CODED. Distal projection of the penisvalva count: NOT CODED. Dorsal apodeme of penisvalva count: NOT CODED. Harpe length: NOT CODED. Distodorsal setae of sensillar ring of harpe length vs. harpe width in lateral view: NOT CODED. Distodorsal setae of sensillar ring of harpe orientation: NOT CODED. Sensillar ring area of harpe orientation: NOT CODED. Lateral setae of harpe count: NOT CODED. Lateral setae of harpe orientation: NOT CODED. Distal margin of harpe in lateral view: shape: NOT CODED. Lateral margin of harpe shape: NOT CODED.

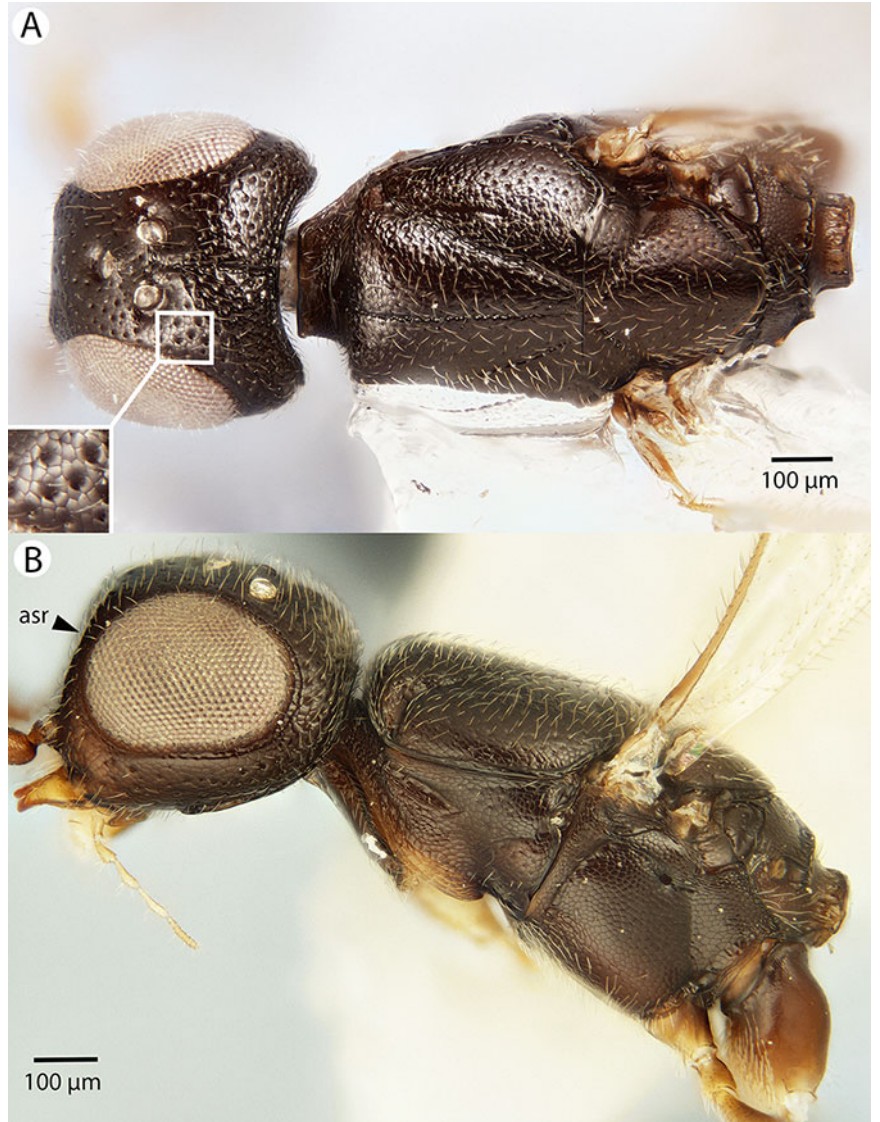

**Figure 10 Brightfield image showing the head and mesosoma of *Conostigmus bucephalus* Mikó and Trietsch sp. nov.** (A) Dorsal view. (B) Lateral view.

**Material examined:** Holotype female: CONGO: PSUC_FEM 8883 Congo Belge: P.N.A 7-XIII-1953 H. Synave 6853 Massif Ruwenzori Mont Ngulingo pres Nyamgaleke, 2,500 m, ex P.N.A HOLOTYPE Prep. micros-copique n 9507/241 (deposited in MRAC).

Other material (2 females): MADAGASCAR: 2 females. CASENT 2001391, 2016542 (CAS).

*Conostigmus bucephalus* Mikó and Trietsch sp. nov.

Figures 10, 11 and 12

**Diagnosis (female):** *Conostigmus bucephalus* sp. nov. differs from other *Conostigmus* species in the presence of the antennal scrobe and the size of impressions around the setal pits on the head: impressions are larger than scutes on the cranium and mesonotum

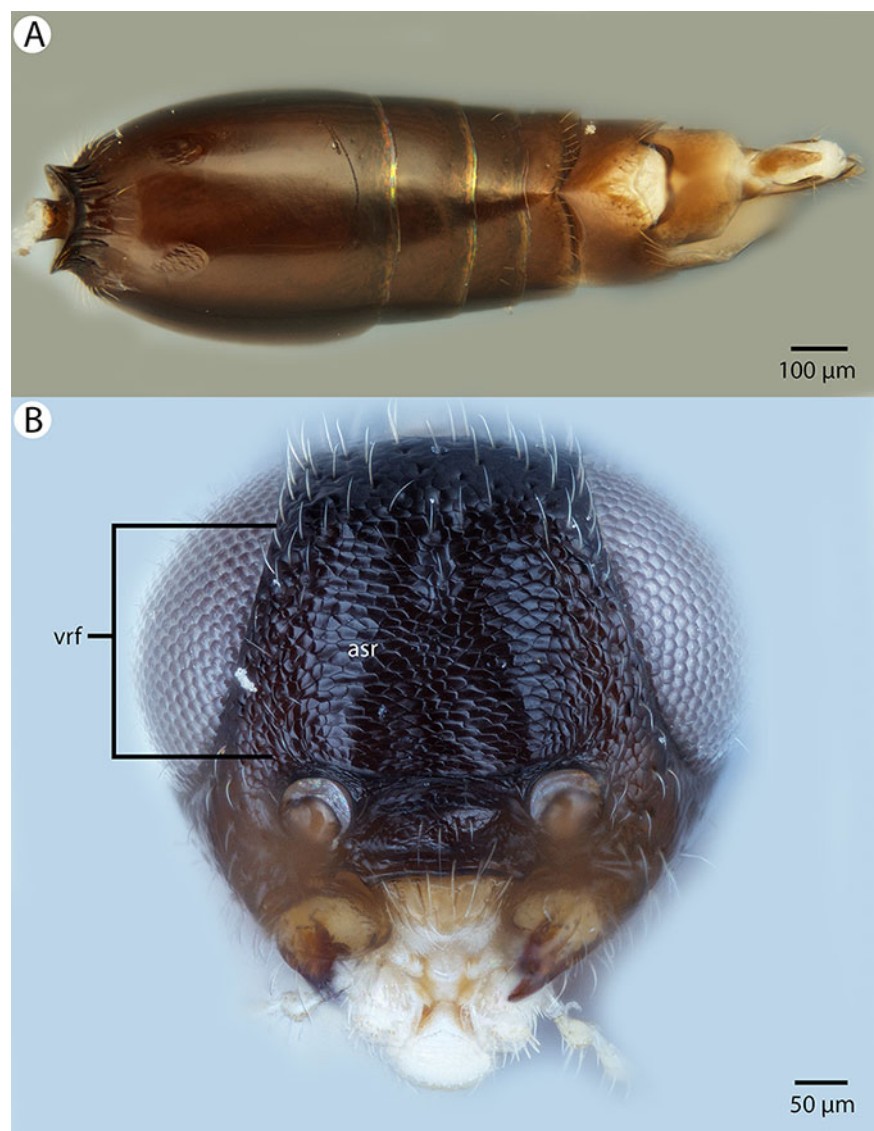

**Figure 11** Brightfield image showing the head and metasoma of *Conostigmus bucephalus* **Mikó and Trietsch sp. nov.** (A) Metasoma, dorsal view. (B) Head, anterior view (vrf, ventral region on frons, asr, antennal scrobe).

in *Conostigmus bucephalus* sp. nov. whereas in other Malagasy species depressions are smaller than scutes on the cranium and mesonotum.

**Description:** Body length: 2,575 μm. Color intensity pattern: distal scape, legs except hind coxa lighter than metasoma. Color hue pattern: Cranium, mesosoma brown; antenna, legs except brown metacoxa, metasoma ochre. Occipital carina sculpture: crenulate. Median flange of occipital carina count: absent. Submedial flange of occipital carina count: absent. Dorsal margin of occipital carina vs. dorsal margin of lateral ocellus in lateral view: occipital carina is dorsal to lateral ocellus in lateral view. Preoccipital lunula count: absent. Preoccipital carina count: absent. Preoccipital carina shape: NOT CODED. Preoccipital furrow count: present. Preoccipital furrow anterior end: Preoccipital furrow ends posterior to ocellar

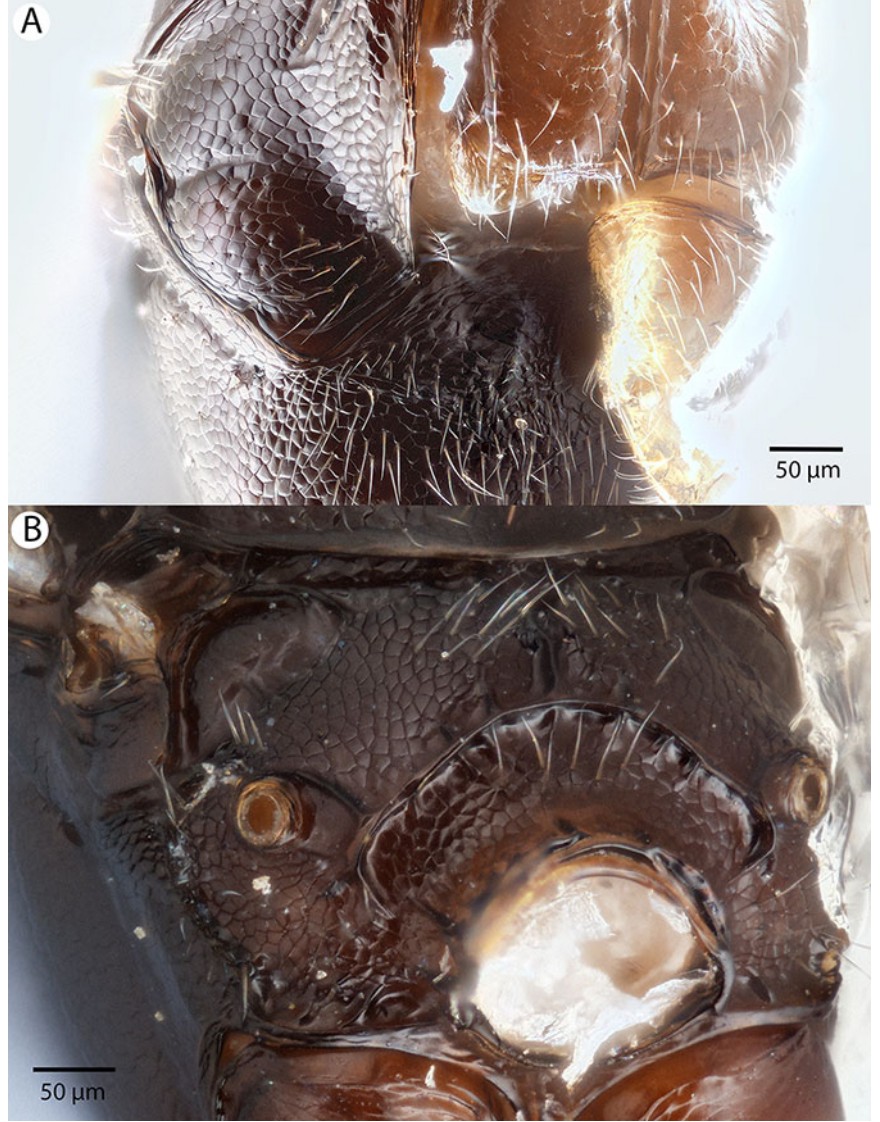

**Figure 12 Brightfield image showing the mesosoma of _Conostigmus bucephalus_ Mikó and Trietsch sp. nov.** (A) Pronotum, part of propleuron and part of mesopectus, anterolateral view. (B) Part of mesomoma, posterior view (vrf, ventral region on frons with transverse scutes, asr, antennal scrobe).

triangle. Postocellar carina count: absent. Male OOL: POL: LOL: NOT CODED. Female OOL: POL: LOL: 1.0:1.2:1.0. HW/IOS Male: NOT CODED. HW/IOS Female: 2.2. Setal pit on vertex size: larger than diameter of scutes. Transverse frontal carina count: absent. Transverse scutes on frons count: present. Rugose region on frons count: absent. Randomly sized areolae around setal pits on frons count: absent. Antennal scrobe count: present. Ventromedian setiferous patch and ventrolateral setiferous patch count: absent. Facial pit count: no external corresponding structure present. Supraclypeal depression count: present. Supraclypeal depression structure: present medially, inverted U-shaped. Intertorular carina count: present. Intertorular area count: absent. Median region of intertorular area shape: NOT CODED. Ventral margin of antennal rim vs. dorsal margin of

clypeus: adjacent. Torulo-clypeal carina count: absent. Subtorular carina count: absent. Mandibular tooth count: 2. Female flagellomere one length vs. pedicel: 0.7. Female ninth flagellomere length: F9 less than F7 + F8. Sensillar patch of the male flagellomere pattern: NOT CODED. Length of setae on male flagellomere vs. male flagellomere width: NOT CODED. Male flagellomere one length vs. male second flagellomere length: NOT CODED. Male flagellomere one length vs. pedicel length: NOT CODED. Ventrolateral invagination of the pronotum count: present. Scutes on posterior region of mesoscutum and dorsal region of mesoscutellum convexity: flat. Notaulus posterior end location: adjacent to transscutal articulation. Median mesoscutal sulcus posterior end: adjacent to transscutal articulation. Scutoscutellar sulcus vs. transscutal articulation: adjacent. Axillular carina count: absent. Axillular carina shape: NOT CODED. Epicnemium posterior margin shape: anterior discrimenal pit absent; epicnemial carina interrupted medially. Epicnemial carina count: present only laterally. Sternaulus count: absent. Sternaulus length: NOT CODED. Speculum ventral limit: not extending ventrally of pleural pit line. Mesometapleural sulcus count: present. Metapleural carina count: present. Transverse line of the metanotum-propodeum vs. antecostal sulcus of the first abdominal tergum: adjacent sublaterally. Lateral propodeal carina count: present. Lateral propodeal carina shape: inverted "Y" (left and right lateral propodeal are adjacent medially posterior to antecostal sulcus of the first abdominal tergum, and connected to the antecostal sulcus by a median carina representing the median branch of the inverted "Y"). Anteromedian projection of the metanoto-propodeo-metapecto-mesopectal complex count: absent. S1 length vs. shortest width: S1 wider than long. Transverse carina on petiole shape: concave. Distal margin of male S9 shape: NOT CODED. Proximolateral corner of male S9 shape: NOT CODED. Cupula length vs. gonostyle-volsella complex length: NOT CODED. Proximodorsal notch of cupula count: NOT CODED. Proximodorsal notch of cupula shape: NOT CODED. Proximolateral projection of the cupula shape: NOT CODED. Proximodorsal notch of cupula width vs. length: NOT CODED. Distodorsal margin of cupula shape: NOT CODED. Dorsomedian conjunctiva of the gonostyle-volsella complex length relative to length of gonostyle-volsella complex: NOT CODED. Dorsomedian conjunctiva of the gonostyle-volsella complex count: NOT CODED. Distal end of dorsomedian conjunctiva of the gonostyle-volsella complex shape: NOT CODED. Parossiculus count (parossiculus and gonostipes fusion): NOT CODED. Apical parossicual seta number: NOT CODED. Distal projection of the parossiculus count: NOT CODED. Distal projection of the penisvalva count: NOT CODED. Dorsal apodeme of penisvalva count: NOT CODED. Harpe length: NOT CODED. Distodorsal setae of sensillar ring of harpe length vs. harpe width in lateral view: NOT CODED. Distodorsal setae of sensillar ring of harpe orientation: NOT CODED. Sensillar ring area of harpe orientation: NOT CODED. Lateral setae of harpe count: NOT CODED. Lateral setae of harpe orientation: NOT CODED. Distal margin of harpe in lateral view: shape: NOT CODED. Lateral margin of harpe shape: NOT CODED.

**Etymology:** The species epithet *bucephalus* (Ancient Greek: bouκ′εφαγος = ox-head) refers to the unique shape of the head that is certainly impacted by the distinct antennal scrobe (asr: Fig. 10B), which is diagnostic for this species.

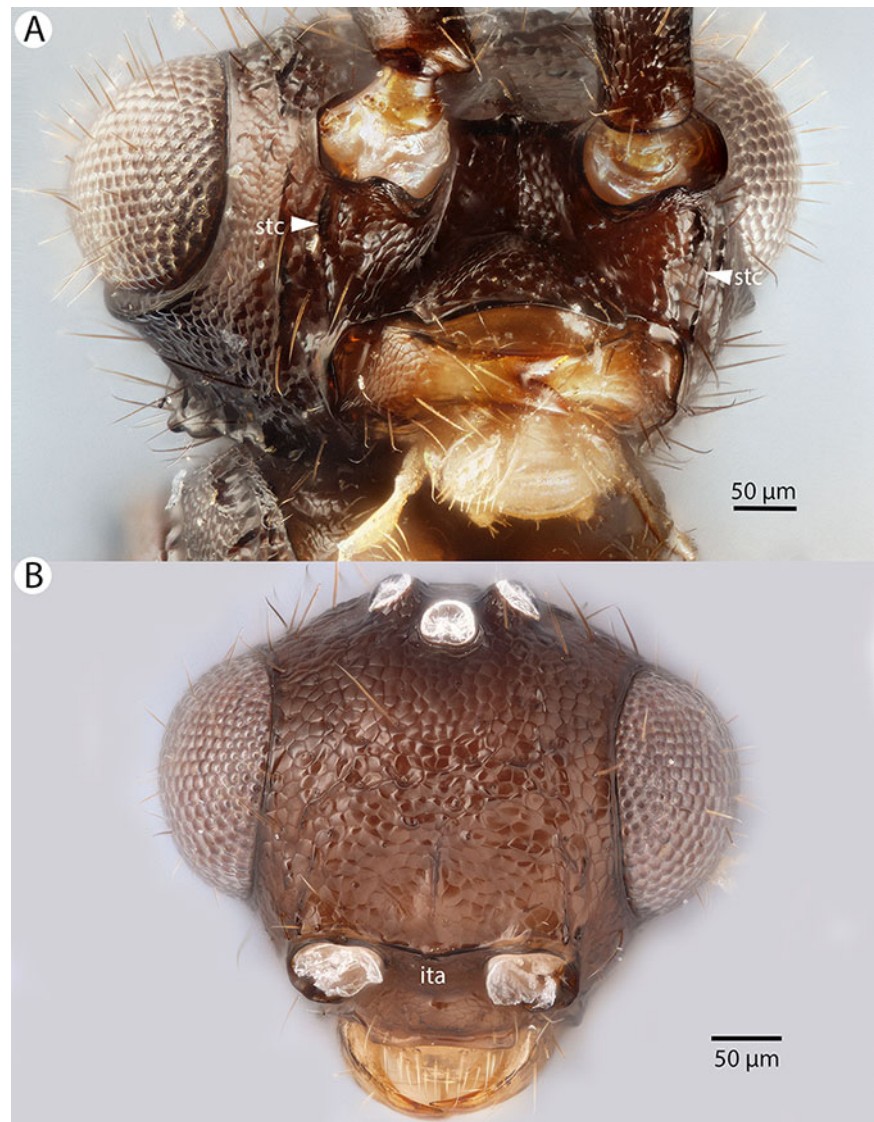

**Figure 13 Brightfield image showing the head of *Conostigmus clavatus Dessart, 1997*.** (A) Ventral view. (B) Anterior view (stc, subtorular carina; ita, intertorular area).

**Comments:** Due to the antennal scrobe that accommodates the scape for almost its entire length, the head is nearly cube shaped in lateral view (Fig. 10B).

**Material examined:** Holotype female: CASENT 2053589 MADAGASCAR: Province Fianarantsoa, Parc National Ranomafana, radio tower at forest edge, elev 1,130 m 20 March–3 April 2003 21°15.05′S, 47°24.43′E collector: R. HarinH́ala California Acad of Sciences malaise, mixed tropical forest MA-02-09B-56 (deposited in CAS).

*Conostigmus clavatus* Mikó and Trietsch sp. nov.

Figures 13, 14, 15, 16 and 17

**Diagnosis:** *Conostigmus clavatus* sp. nov. shares the presence of the axillular carina, bulging eyes and medially convex intertorular area (and intertorular carina) with

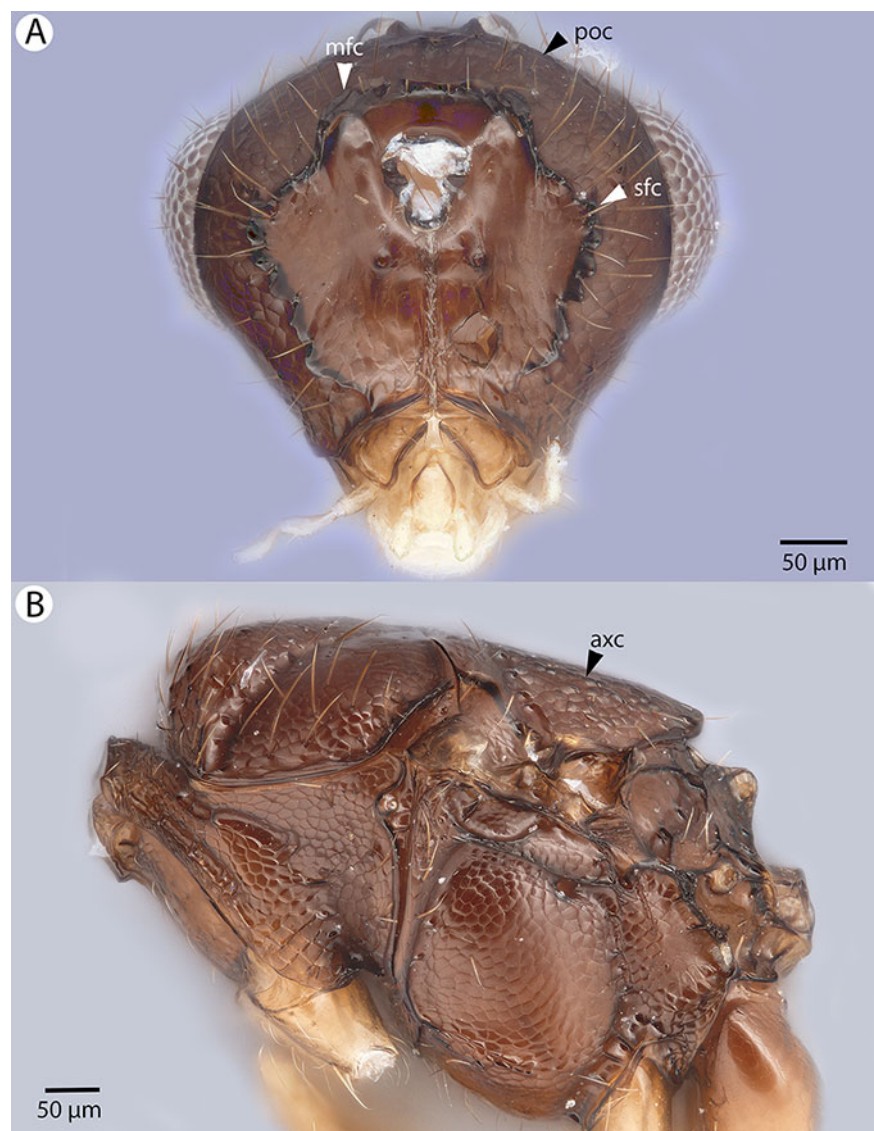

**Figure 14** Brightfield image showing the head and mesosoma of *Conostigmus clavatus Dessart, 1997*. (A) Head, posterior view. (B) Mesosoma, lateral view (poc, preoccipital carina; axc, axillular carina; mfc, median flange of occipital carina; sfc, submedial flange of occipital carina).

*C. uninasutus* Alekseev, 1994 and *C. binasutus* Dessart & Cancemi, 1987 and differs from them in the enlarged distal-most female flagellomere (length of F9 = length of F6 + length of F7 + length of F8, Fig. 16A).

**Description:** Body length: 2,325–2,500 μm. Color intensity pattern: metasoma lighter than mesosoma and cranium. Color hue pattern: Dark brown except pedicel, proximal 1/5th of scape, fore and middle leg, mandible ochre/yellowish. Occipital carina sculpture: crenulate. Median flange of occipital carina count: present. Submedial flange of occipital carina count: present. Dorsal margin of occipital carina vs. dorsal margin of lateral ocellus in lateral view: occipital carina is ventral to lateral ocellus in lateral view. Preoccipital lunula count: present. Preoccipital carina count: present; absent. Preoccipital carina shape: interrupted dorsally and represented by irregular, not continuous carinae.

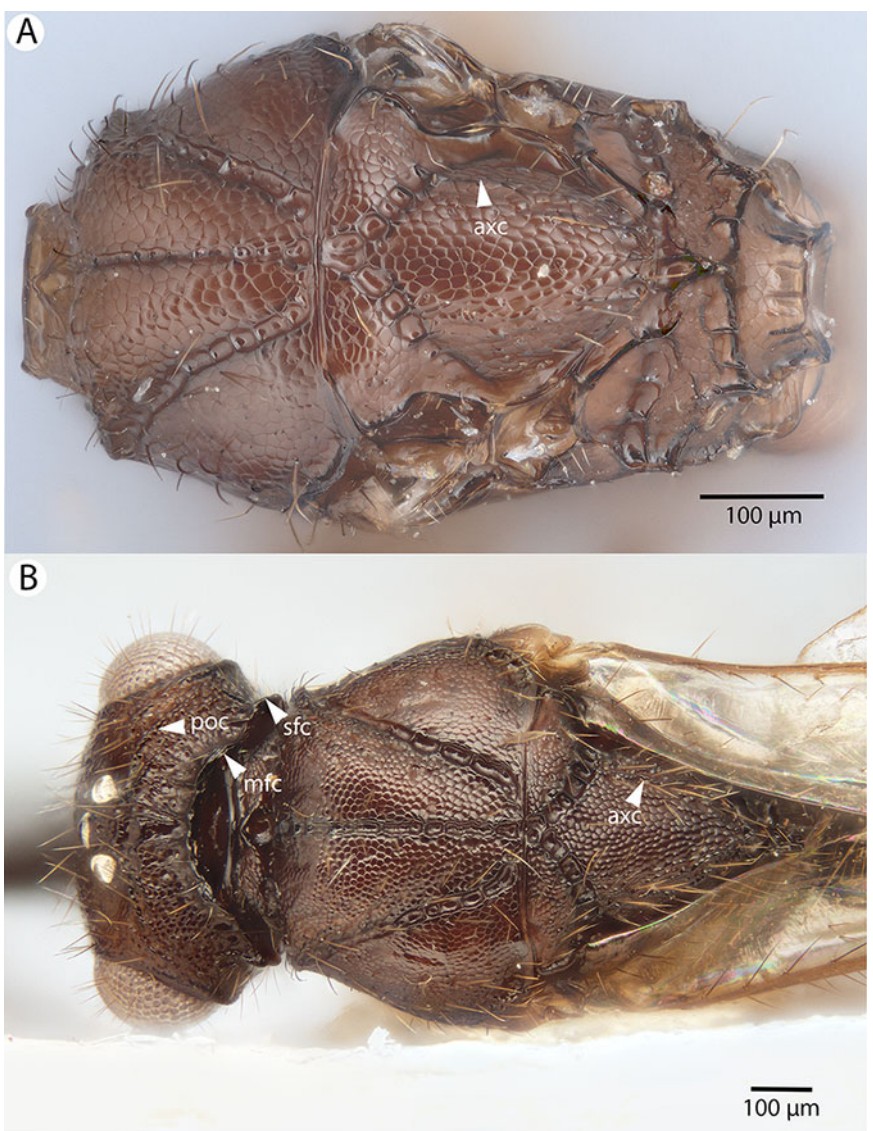

**Figure 15** Brightfield image showing the head and mesosoma of *Conostigmus clavatus Dessart, 1997*. (A) Mesomoma, dorsal view. (B) Head and mesosoma, dorsal view (poc, preoccipital carina; axc, axillular carina; mfc, median flange of occipital carina; sfc, submedial flange of occipital carina).

Preoccipital furrow count: present. Preoccipital furrow anterior end: Preoccipital furrow ends posterior to ocellar triangle. Postocellar carina count: absent. Male OOL: POL: LOL: 2.9–3.6:2.1–2.2:1. Female OOL: POL: LOL: 3.4:2.1–2.2:1.0. HW/IOS Male: 1.6–1.7. HW/IOS Female: 1.6–1.7. Setal pit on vertex size: smaller than diameter of scutes. Transverse frontal carina count: absent. Transverse scutes on frons count: absent. Rugose region on frons count: present; absent. Randomly sized areolae around setal pits on frons count: absent. Antennal scrobe count: absent. Ventromedian setiferous patch and ventrolateral setiferous patch count: absent. Facial pit count: median facial keel present. Supraclypeal depression count: present. Supraclypeal depression structure: absent medially, represented by two grooves laterally of facial pit. Intertorular carina count: present. Intertorular area count: present. Median region of intertorular area shape:

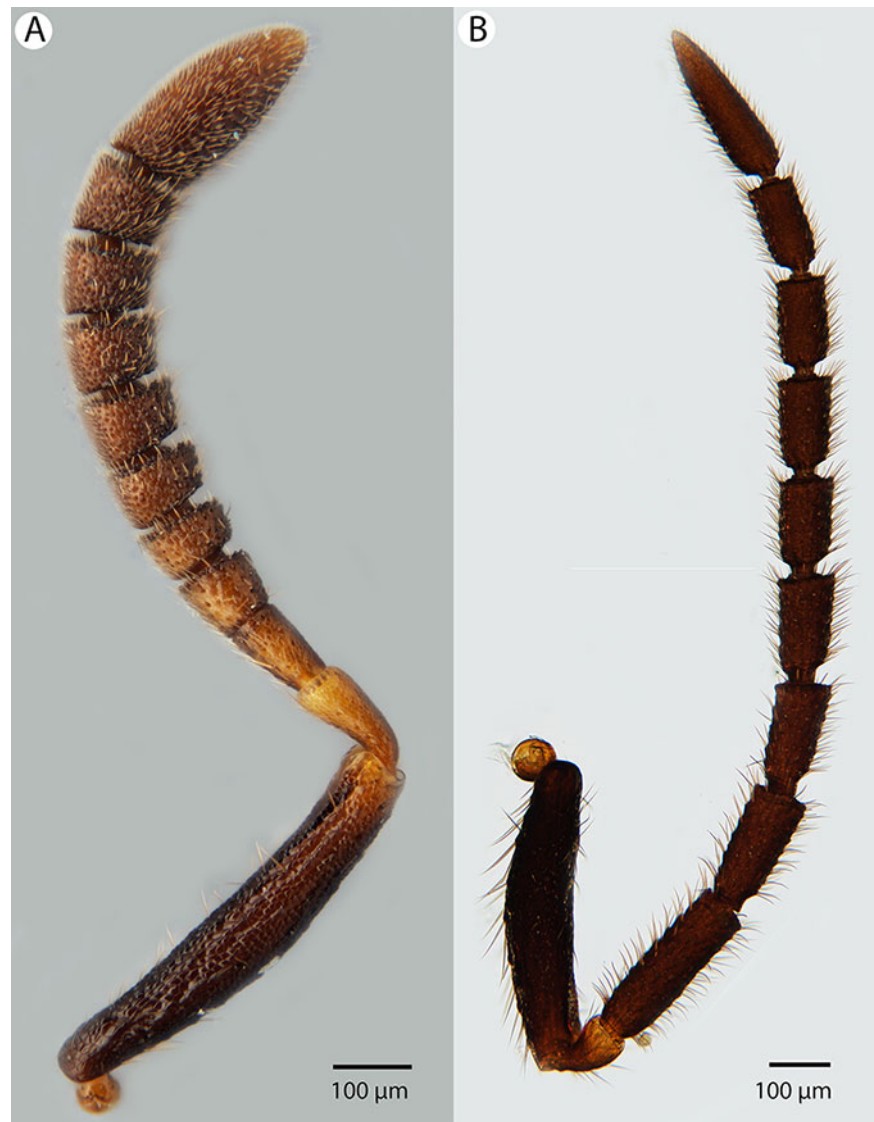

**Figure 16** **Brightfield image showing the antenna of *Conostigmus clavatus Dessart, 1997*.** (A) Female. (B) Male.

convex. Ventral margin of antennal rim vs. dorsal margin of clypeus: not adjacent. Torulo-clypeal carina count: absent. Subtorular carina count: present. Mandibular tooth count: 2. Female flagellomere one length vs. pedicel: 0.9. Female ninth flagellomere length: F9 = F6 + F7 + F8. Sensillar patch of the male flagellomere pattern: F5–F9. Length of setae on male flagellomere vs. male flagellomere width: setae shorter than width of flagellomeres. Male flagellomere one length vs. male second flagellomere length: 1.2–1.3. Male flagellomere one length vs. pedicel length: 2.1–2.4. Ventrolateral invagination of the pronotum count: present. Scutes on posterior region of mesoscutum and dorsal region of mesoscutellum convexity: flat. Notaulus posterior end location: adjacent to transscutal articulation. Median mesoscutal sulcus posterior end: adjacent to transscutal articulation. Scutoscutellar sulcus vs. transscutal articulation: adjacent. Axillular carina

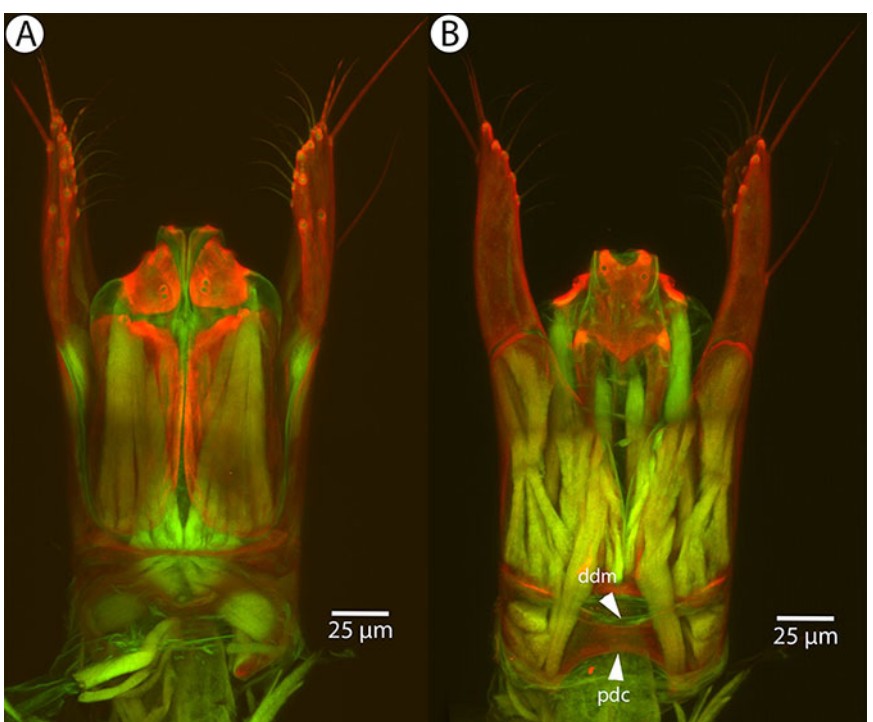

**Figure 17** CLSM volume rendered micrographs showing the male genitalia of *Conostigmus clavatus* **Mikó and Trietsch sp. nov.** (A) Ventral view (B) Dorsal view (ddm, distodorsal margin of cupula; pdc, proximodorsal notch of cupula).

count: present. Axillular carina shape: The left and right carina are separated posteromedially. Epicnemium posterior margin shape: anterior discrimenal pit present; epicnemial carina curved. Epicnemial carina count: complete. Sternaulus count: present. Sternaulus length: short, not reaching 1/2 of mesopleuron length at level of sternaulus. Speculum ventral limit: not extending ventrally of pleural pit line. Mesometapleural sulcus count: present. Metapleural carina count: present. Transverse line of the metanotum-propodeum vs. antecostal sulcus of the first abdominal tergum: adjacent sublaterally. Lateral propodeal carina count: present. Lateral propodeal carina shape: NOT CODED. Anteromedian projection of the metanoto-propodeo-metapecto-mesopectal complex count: present. S1 length vs. shortest width: S1 wider than long. Transverse carina on petiole shape: straight. Distal margin of male S9 shape: convex. Proximolateral corner of male S9 shape: acute. Cupula length vs. gonostyle-volsella complex length: cupula less than 1/2 the length of gonostyle-volsella complex in lateral view. Proximodorsal notch of cupula count: present. Proximodorsal notch of cupula shape: arched. Proximolateral projection of the cupula shape: acute. Proximodorsal notch of cupula width vs. length: as long as wide. Distodorsal margin of cupula shape: concave. Dorsomedian conjunctiva of the gonostyle-volsella complex length relative to length of gonostyle-volsella complex: dorsomedian conjunctiva extending 2/3 of length of gonostyle-volsella complex in dorsal view. Dorsomedian conjunctiva of the gonostyle-volsella complex count: present. Distal end of dorsomedian conjunctiva of the

gonostyle-volsella complex shape: blunt. Parossiculus count (parossiculus and gonostipes fusion): present (not fused with the gonostipes). Apical parossicular seta number: one; two. Distal projection of the parossiculus count: absent. Distal projection of the penisvalva count: absent. Dorsal apodeme of penisvalva count: absent. Harpe length: harpe shorter than gonostipes in lateral view. Distodorsal setae of sensillar ring of harpe length vs. harpe width in lateral view: setae as long or shorter than harpe width. Distodorsal setae of sensillar ring of harpe orientation: distomedially. Sensillar ring area of harpe orientation: medially. Lateral setae of harpe count: present. Lateral setae of harpe orientation: oriented distally. Distal margin of harpe in lateral view: shape: blunt. Lateral margin of harpe shape: widest point of harpe is at its articulation site with gonostyle-volsella complex.

**Etymology:** The species epithet *clavatus* refers to the enlarged apical female flagellomere, resembling a club (F9 > F8 + F7 + F6).

**Comments:** *Conostigmus clavatus*, *C. binasutus* and *C. uninasutus* share numerous morphological traits with *Megaspilus* Westwood, 1929 including the presence of bulging eyes, the crenulate and distinct ocular suture, the presence of the axillular carina and the large body size (>2,000 μm). Females of all *Conostigmus* species exhibit a distinct clava with three rows of ventral, female specific basiconic sensilla (distally gradually widening flagellum). The antenna of female *Megaspilus* is filiform and lacks ventral basiconic sensilla (I. Mikó, 2010, personal observation).

**Material examined:** Holotype male: MADAGASCAR: Province Fianarantsoa, Parc National Ranomafana, Vohiparara at broken bridge, Malaise trap in high altitude rainforest, 22–28.11.2001, R. Harin'Hala, CASENT 2044514 (deposited in CAS). Paratypes (seven males, four females): MADAGASCAR: seven males, four females. CASENT 2002179, 2032775, 2044150, 2045085, 2045509, 2045602, 2045755, 2046024, 2046178–2046179, 2053642 (deposited in CAS, MRAC).

*Conostigmus fianarantsoaensis* Mikó and Trietsch sp. nov.
Figures 18, 19, 20 and 21

**Diagnosis:** *Conostigmus fianarantsoensis* sp. nov. is most similar to *C. madagascariensis* sp. nov. among Malagasy *Conostigmus* and differs from it by the following characters: mandible with one tooth (mandible with two teeth in *C. madagascariensis*); flagellar setae shorter than the flagellomere width (in *C. madagascariensis*, flagellar setae are distinctly longer than the flagellomere width); blunt proximolateral projection of cupula (acute in *C. madagascariensis*); V-shaped (notched) proximodorsal notch of cupula (U-shaped (arched) in *C. madagascariensis*); blunt distal end of dorsomedial conjunctiva of gonostyle/volsella complex (acute in *C. madagascariensis*); and the acute distal margin of harpe in lateral view (blunt in *C. madagascariensis*).

**Description:** Body length: 1,150–2,300 μm. Color intensity pattern: metasoma and mandible lighter than mesosoma. Color hue pattern: F3–F8, cranium, mandible, metasoma, tegula brown; legs, except brown proximal region of metacoxa and distal region of metafemur, scape, pedicel, F1–F4 yellow. Occipital carina sculpture: crenulate.

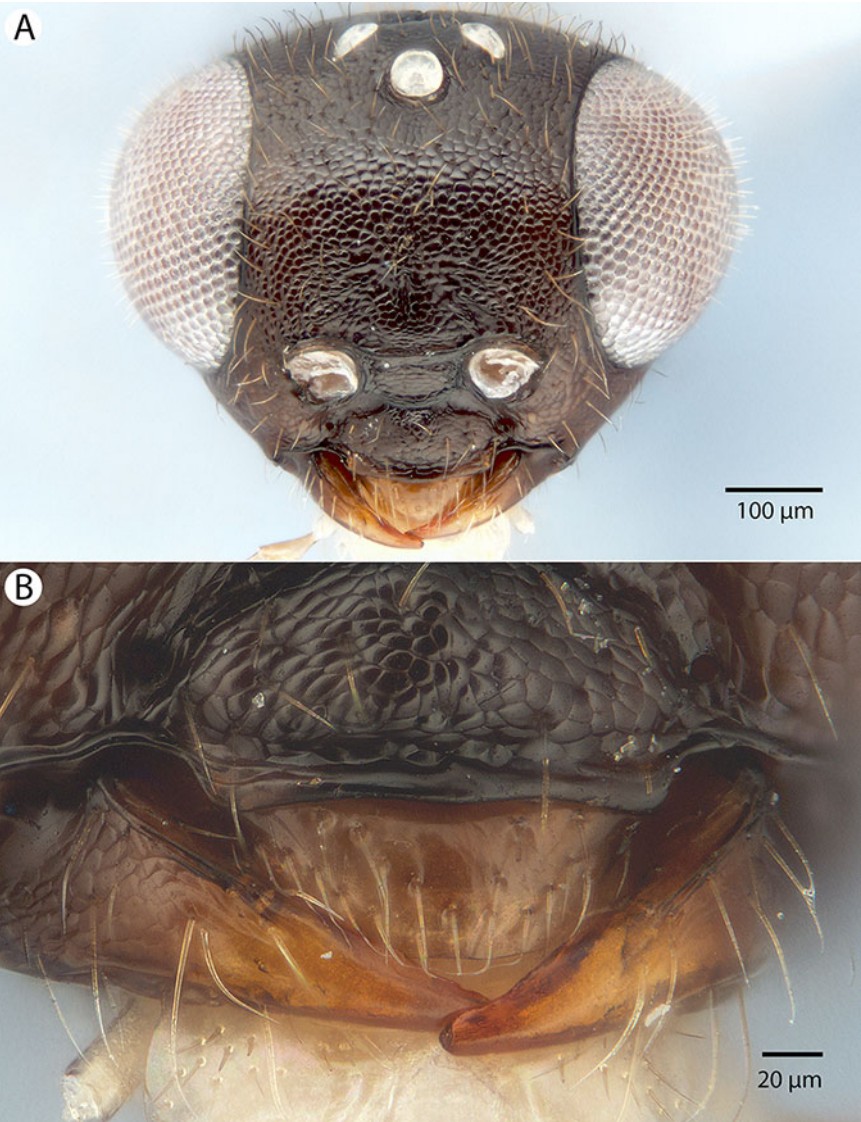

**Figure 18 Brightfield image showing the head of *Conostigmus madagascariensis* Mikó and Trietsch sp. nov.** (A) Head, anterior view. (B) Mandible, labrum and clypeus, anteroventral view.

Median flange of occipital carina count: absent. Submedial flange of occipital carina count: absent. Dorsal margin of occipital carina vs. dorsal margin of lateral ocellus in lateral view: occipital carina is ventral to lateral ocellus in lateral view. Preoccipital lunula count: present. Preoccipital carina count: absent. Preoccipital carina shape: NOT CODED. Preoccipital furrow count: present. Preoccipital furrow anterior end: Preoccipital furrow ends posterior to ocellar triangle. Postocellar carina count: absent. Male OOL: POL: LOL: 1.4–1.8:1.5–1.8:1. Female OOL: POL: LOL: 1.7–2.3:1.7–1.8:1.0. HW/IOS Male: 1.6–1.9. HW/IOS Female: 2.0–2.2. Setal pit on vertex size: smaller than diameter of scutes. Transverse frontal carina count: absent. Transverse scutes on frons count: absent. Rugose region on frons count: absent. Randomly sized areolae around setal pits on frons count: absent. Antennal scrobe count: absent. Ventromedian setiferous patch and

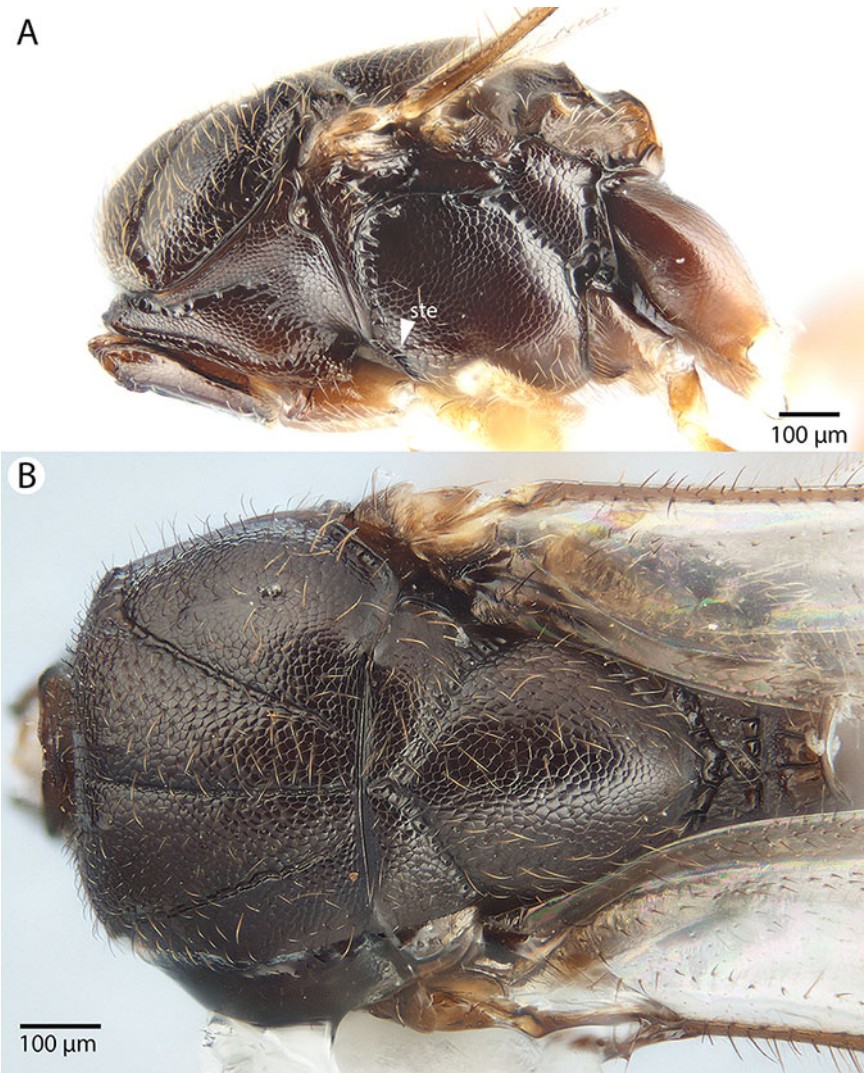

**Figure 19 Brightfield image showing the mesosoma of *Conostigmus madagascariensis* Mikó and Trietsch sp. nov.** (A) mesosoma, lateral view. (B) Mesosoma, dorsal view (ste, sternaulus).

ventrolateral setiferous patch count: absent. Facial pit count: facial pit present. Supraclypeal depression count: present. Supraclypeal depression structure: absent medially, represented by two grooves laterally of facial pit. Intertorular carina count: present. Intertorular area count: present. Median region of intertorular area shape: flat. Ventral margin of antennal rim vs. dorsal margin of clypeus: not adjacent. Torulo-clypeal carina count: present. Subtorular carina count: absent. Mandibular tooth count: 1. Female flagellomere one length vs. pedicel: 0.8–1.16. Female ninth flagellomere length: F9 less than F7 + F8. Sensillar patch of the male flagellomere pattern: F5–F9. Length of setae on male flagellomere vs. male flagellomere width: setae shorter than width of flagellomeres. Male flagellomere one length vs. male second flagellomere length: 1.2–1.4. Male flagellomere one length vs. pedicel length: 2.9–3.3. Ventrolateral invagination of the pronotum count: present. Scutes on posterior region of mesoscutum and dorsal region of mesoscutellum convexity: flat. Notaulus posterior end location: adjacent to transscutal

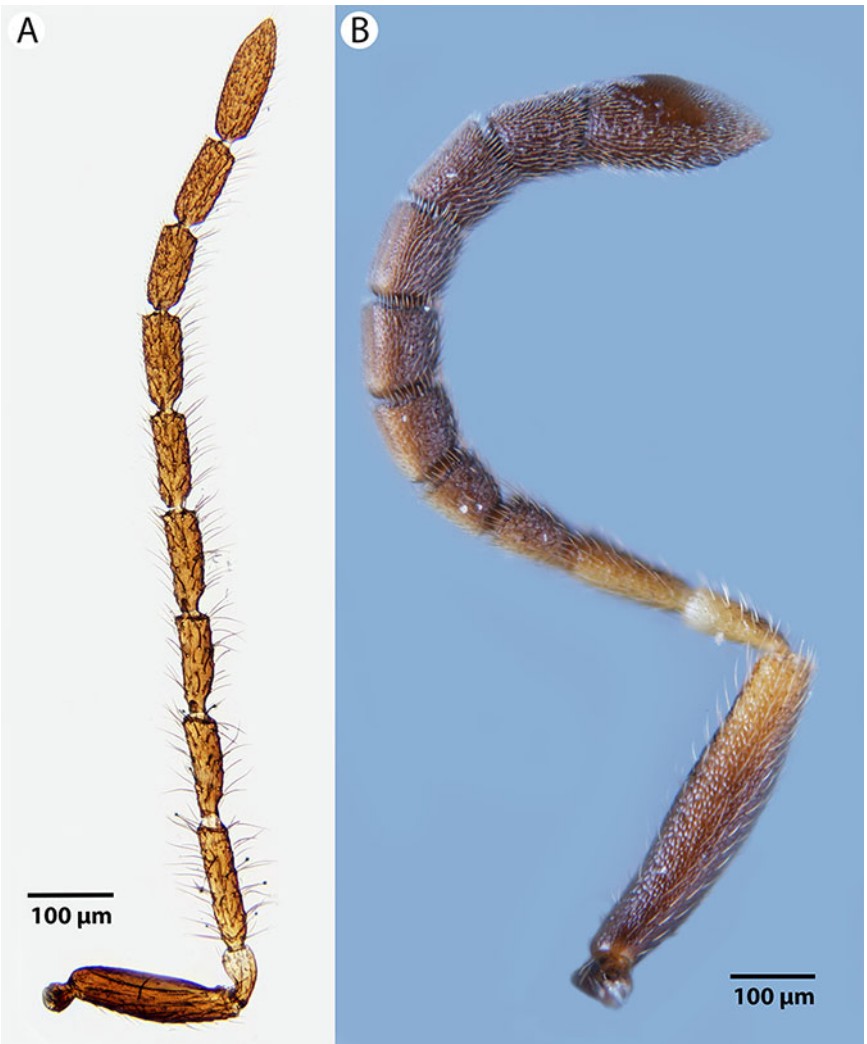

**Figure 20 Brightfield image showing the antenna of *Conostigmus madagascariensis* Mikó and Trietsch sp. nov.** (A) Male. (B) Female.

articulation. Median mesoscutal sulcus posterior end: adjacent to transscutal articulation. Scutoscutellar sulcus vs. transscutal articulation: adjacent; not adjacent. Axillular carina count: absent. Axillular carina shape: NOT CODED. Epicnemium posterior margin shape: anterior discrimenal pit present; epicnemial carina curved. Epicnemial carina count: interupted medially; complete. Sternaulus count: present. Sternaulus length: short, not reaching 1/2 of mesopleuron length at level of sternaulus. Speculum ventral limit: not extending ventrally of pleural pit line. Mesometapleural sulcus count: present. Metapleural carina count: present. Transverse line of the metanotum-propodeum vs. antecostal sulcus of the first abdominal tergum: adjacent sublaterally. Lateral propodeal carina count: present. Lateral propodeal carina shape: inverted "Y" (left and right lateral propodeal carinae are adjacent medially at their intersection with antecostal sulcus of the first abdominal tergum). Anteromedian projection of the metanoto-propodeo-metapecto-mesopectal complex count: absent. S1 length vs. shortest width: S1 wider than

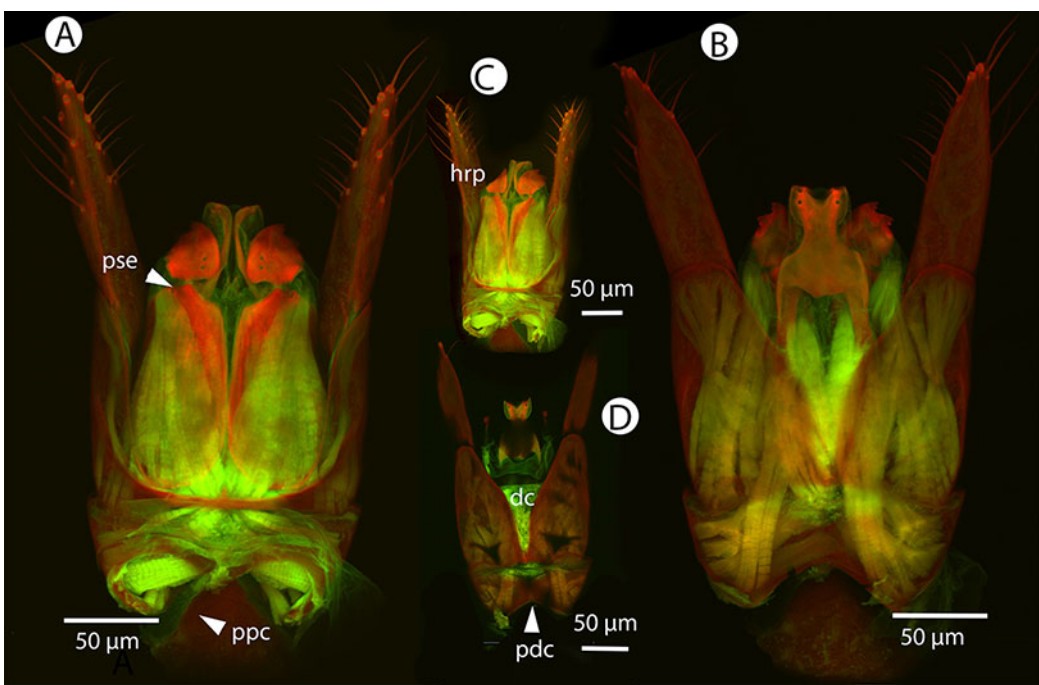

**Figure 21** CLSM volume rendered micrographs showing the male genitalia of *Conostigmus fianarantsoaensis* Mikó and Trietsch sp. nov. (A) Ventral view. (B) Dorsal view. (C) Lateroventral view. (D) Dorsal view partially rendered (ppc, proximolateral projection of cupula; pdc, proximodorsal notch of cupula; hrp, harpe; dc, Dorsomedian conjunctiva of the gonostyle/volsella complex; aps, apical parossicular seta).

long. Transverse carina on petiole shape: straight. Distal margin of male S9 shape: straight. Proximolateral corner of male S9 shape: blunt. Cupula length vs. gonostyle-volsella complex length: cupula less than 1/2 the length of gonostyle-volsella complex in lateral view. Proximodorsal notch of cupula count: present. Proximodorsal notch of cupula shape: notched. Proximolateral projection of the cupula shape: blunt. Proximodorsal notch of cupula width vs. length: wider than long. Distodorsal margin of cupula shape: straight. Dorsomedian conjunctiva of the gonostyle-volsella complex length relative to length of gonostyle-volsella complex: dorsomedian conjunctiva extending 2/3 of length of gonostyle-volsella complex in dorsal view. Dorsomedian conjunctiva of the gonostyle-volsella complex count: present. Distal end of dorsomedian conjunctiva of the gonostyle-volsella complex shape: blunt. Parossiculus count (parossiculus and gonostipes fusion): present (not fused with the gonostipes). Apical parossicular seta number: one. Distal projection of the parossiculus count: absent. Distal projection of the penisvalva count: absent. Dorsal apodeme of penisvalva count: absent. Harpe length: harpe shorter than gonostipes in lateral view. Distodorsal setae of sensillar ring of harpe length vs. harpe width in lateral view: setae as long or shorter than harpe width. Distodorsal setae of sensillar ring of harpe orientation: medially. Sensillar ring area of harpe orientation: medially. Lateral setae of harpe count: present. Lateral setae of harpe orientation: oriented distally. Distal margin of harpe in lateral view: shape: acute. Lateral margin of harpe shape: widest point of harpe is at its articulation site with gonostyle-volsella complex.

**Etymology:** The species epithet refers to the Fianarantsoa Province of Madagascar, where all specimens of this species were collected.

**Comments:** This species is very similar to *Conostigmus madagascariensis* sp. nov., and the two might possibly represent a single species.

**Material examined:** Holotype male: MADAGASCAR: Ranomafana JIRAMA water works, Malaise trap near river, 16.10–8.11.2001, R. Harin'Hala, CASENT 2053691 (deposited in CAS). Paratypes (17 males, one sex unknown, three females): MADAGASCAR: 17 males, one sex unknown, three females. CASENT 2022988, 2044151, 2045601, 2045741, 2045975, 2046177, 2046180, 2053303, 2053306, 2053641, 2053667; IM 2288; PSUC_FEM 79695, 79734, 79737–79738, 79740, 79749, 79756, 79760, 79762 (CAS, MRAC).

*Conostigmus longulus* Dessart, 1997
Figures 22, 23, 24, 25 and 26

**Diagnosis:** *Conostigmus longulus* Dessart, 1997 shares the presence of a prognathous head (dorsal-most point of occipital carina is dorsal to posterior ocellus in lateral view) and the presence of transverse scutes on the ventral region of the frons with *C. babaiax* Dessart, 1996, *C. toliaraensis* sp. nov. and *C. pseudobabaiax* sp. nov. *Conostigmus longulus* differs from *C. babaiax*, *C. toliaraensis* sp. nov. and *C. pseudobabaiax* in the presence of an impression surrounding the frontal pit, the absence of white setal patches on the frons, and the presence of the transverse frontal carina. *Conostigmus longulus* differs from other *Conostigmus* species in the distodorsal orientation of the sensillar ring of the harpe (the sensillar ring is oriented distomedially or distoventrally in other *Conostigmus* species).

**Description:** Body length: 1,750–2,450 μm. Color intensity pattern: ventral region of cranium is lighter than dorsal region of cranium. Color hue pattern: Legs except proximal region of metacoxa and distal region of metafemur, mouthparts yellow; rest of body ochre; Legs except proximal region of metacoxa and distal 2/3 of metafemur, mouthparts, scape and F1 orange; rest of body brown. Occipital carina sculpture: crenulate. Median flange of occipital carina count: absent. Submedial flange of occipital carina count: absent. Dorsal margin of occipital carina vs. dorsal margin of lateral ocellus in lateral view: occipital carina is dorsal to lateral ocellus in lateral view. Preoccipital lunula count: NOT CODED. Preoccipital carina count: absent. Preoccipital carina shape: NOT CODED. Preoccipital furrow count: present. Preoccipital furrow anterior end: Preoccipital furrow ends inside ocellar triangle. Postocellar carina count: absent. Male OOL: POL: LOL: 1.1–1.2:1:1. Female OOL: POL: LOL: 1.2–1.3:1.0:1.0. HW/IOS Male: 2.0–2.5. HW/IOS Female: 2.3–2.4. Setal pit on vertex size: smaller than diameter of scutes. Transverse frontal carina count: present. Transverse scutes on frons count: present. Rugose region on frons count: absent. Randomly sized areolae around setal pits on frons count: absent. Antennal scrobe count: absent. Ventromedian setiferous patch and ventrolateral setiferous patch count: absent. Facial pit count: facial pit present. Supraclypeal depression count: present. Supraclypeal depression structure: present medially, inverted U-shaped. Intertorular carina count: present.

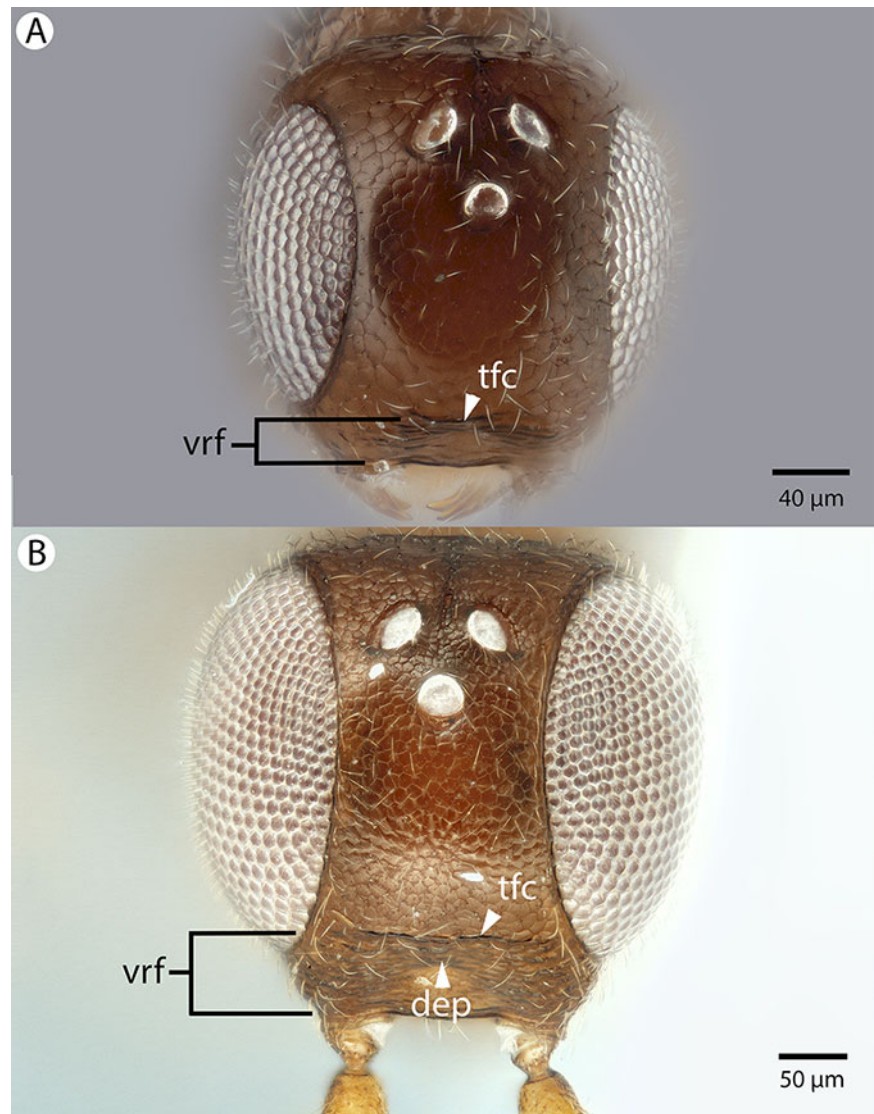

**Figure 22** Brightfield image showing the head of *Conostigmus longulus Dessart, 1997*, anterior view. (A) Smaller specimen. (B) Larger specimen (tfc, transverse frontal carina; vrf, ventral region of frons; dep, depression surrounding frontal pit).

Intertorular area count: present. Median region of intertorular area shape: flat. Ventral margin of antennal rim vs. dorsal margin of clypeus: not adjacent. Torulo-clypeal carina count: absent. Subtorular carina count: absent. Mandibular tooth count: 2. Female flagellomere one length vs. pedicel: F1 as long as pedicel (1.0–1.1). Female ninth flagellomere length: F9 less than F7 + F8. Sensillar patch of the male flagellomere pattern: F4–F9; F5–F9. Length of setae on male flagellomere vs. male flagellomere width: setae shorter than width of flagellomeres. Male flagellomere one length vs. male second flagellomere length: 1.2–1.4. Male flagellomere one length vs. pedicel length: 2.4–2.5. Ventrolateral invagination of the pronotum count: present. Scutes on posterior region of mesoscutum and dorsal region of mesoscutellum convexity: flat. Notaulus posterior end location: adjacent to transscutal articulation. Median mesoscutal sulcus

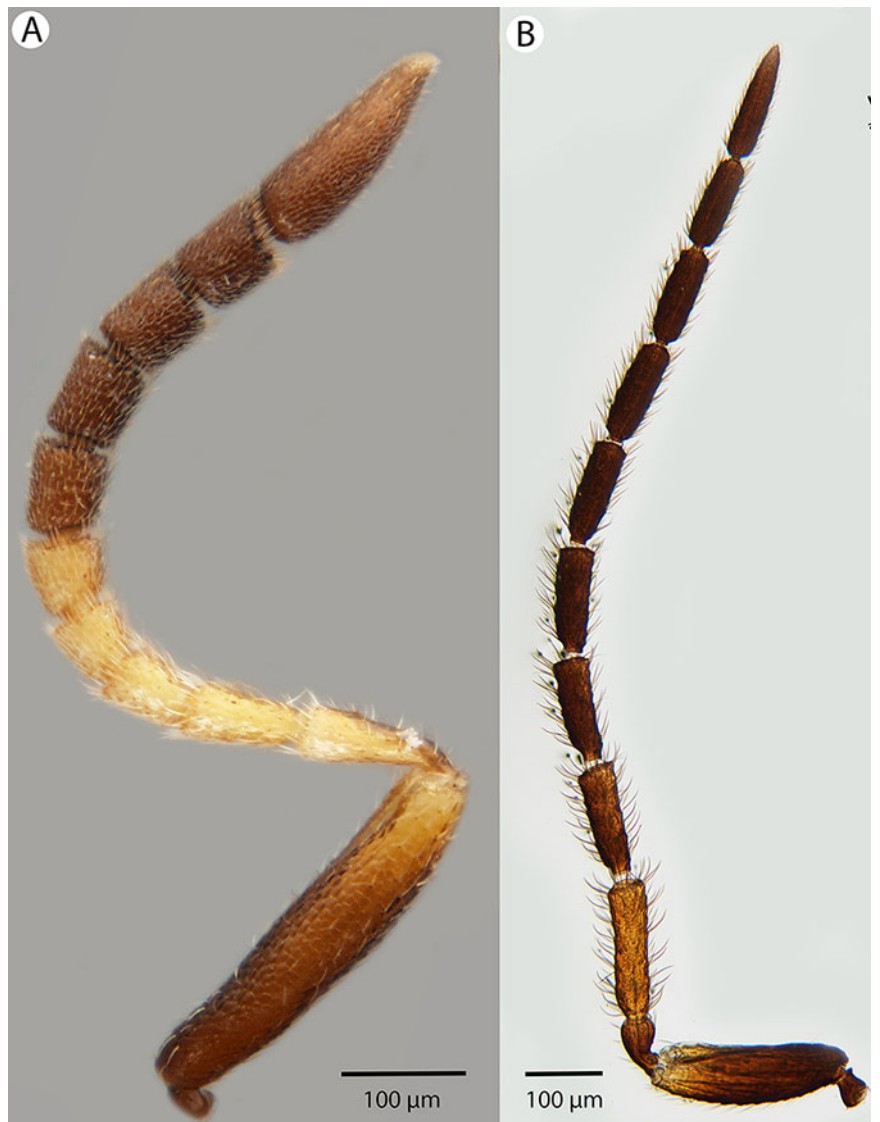

**Figure 23 Brightfield image showing the antenna of *Conostigmus longulus* *Dessart, 1997*, lateral view.** (A) Female. (B) Male.

posterior end: adjacent to transscutal articulation. Scutoscutellar sulcus vs. transscutal articulation: adjacent. Axillular carina count: absent. Axillular carina shape: NOT CODED. Epicnemium posterior margin shape: anterior discrimenal pit absent; epicnemial carina interrupted medially. Epicnemial carina count: present only laterally. Sternaulus count: absent; present. Sternaulus length: short, not reaching 1/2 of mesopleuron length at level of sternaulus. Speculum ventral limit: not extending ventrally of pleural pit line. Mesometapleural sulcus count: present. Metapleural carina count: present. Transverse line of the metanotum-propodeum vs. antecostal sulcus of the first abdominal tergum: adjacent sublaterally. Lateral propodeal carina count: present. Lateral propodeal carina shape: inverted "Y" (left and right lateral propodeal are adjacent medially posterior to antecostal sulcus of the first abdominal tergum, and

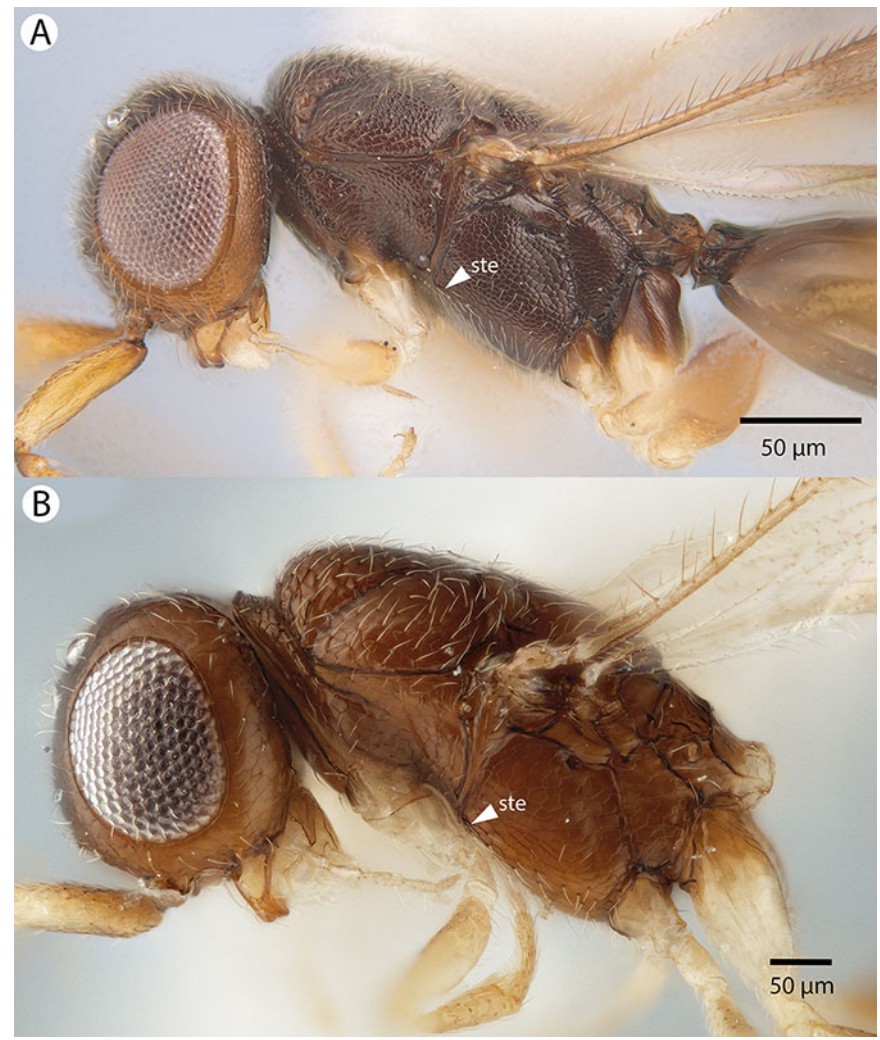

**Figure 24 Brightfield image showing the head and mesosoma of *Conostigmus longulus Dessart, 1997*, anterior view** (A) Larger specimen. (B) Smaller specimen.

connected to the antecostal sulcus by a median carina representing the median branch of the inverted "Y"); straight (left and right lateral propodeal carinae compose a carina that is not broken medially). Anteromedian projection of the metanoto-propodeo-metapecto-mesopectal complex count: absent. S1 length vs. shortest width: S1 wider than long. Transverse carina on petiole shape: concave. Distal margin of male S9 shape: convex. Proximolateral corner of male S9 shape: blunt. Cupula length vs. gonostyle-volsella complex length: cupula less than 1/2 the length of gonostyle-volsella complex in lateral view. Proximodorsal notch of cupula count: present. Proximodorsal notch of cupula shape: arched. Proximolateral projection of the cupula shape: acute. Proximodorsal notch of cupula width vs. length: wider than long. Distodorsal margin of cupula shape: straight. Dorsomedian conjunctiva of the gonostyle-volsella complex length relative to length of gonostyle-volsella complex: dorsomedian conjunctiva extending 2/3 of length of gonostyle-volsella complex in dorsal view. Dorsomedian conjunctiva of the gonostyle-volsella complex count: present. Distal end of

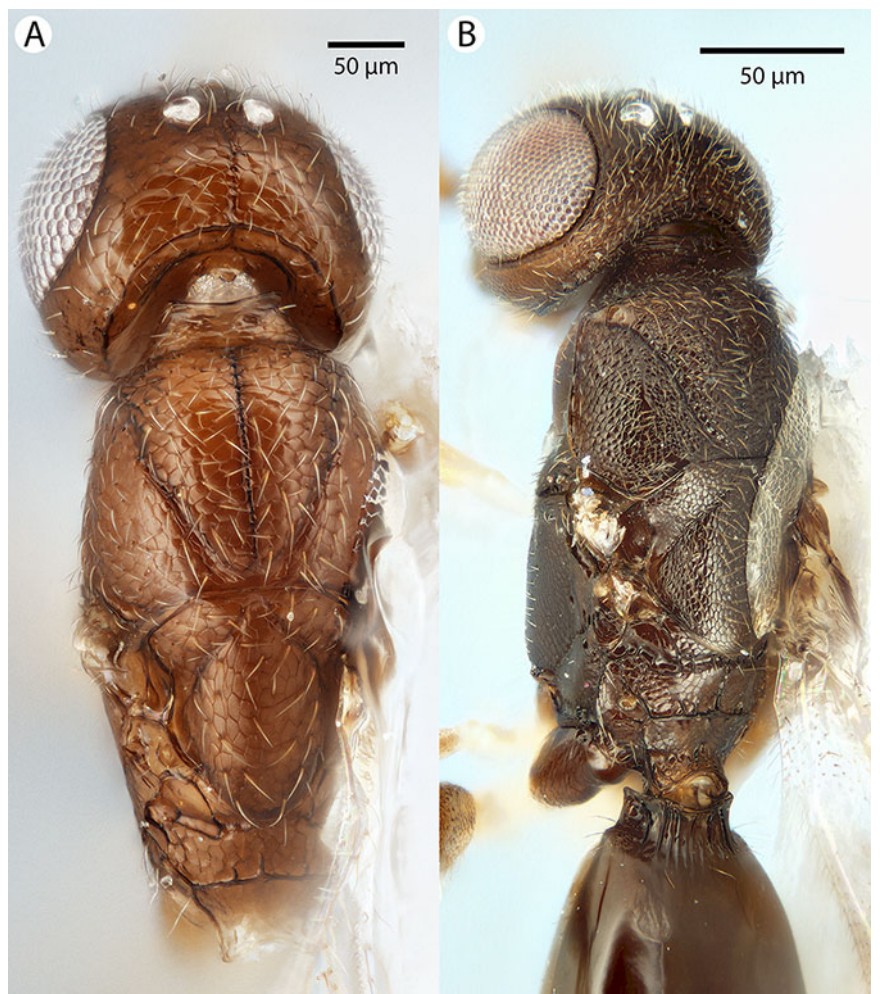

**Figure 25 Brightfield image showing the head and mesosoma of *Conostigmus longulus Dessart, 1997*, anterior view.** (A) Larger specimen. (B) Smaller specimen.

dorsomedian conjunctiva of the gonostyle-volsella complex shape: acute. Parossiculus count (parossiculus and gonostipes fusion): present (not fused with the gonostipes). Apical parossicular seta number: one. Distal projection of the parossiculus count: absent. Distal projection of the penisvalva count: absent. Dorsal apodeme of penisvalva count: absent. Harpe length: harpe shorter than gonostipes in lateral view. Distodorsal setae of sensillar ring of harpe length vs. harpe width in lateral view: setae two times as long as harpe width. Distodorsal setae of sensillar ring of harpe orientation: distomedially. Sensillar ring area of harpe orientation: dorsomedially. Lateral setae of harpe count: present. Lateral setae of harpe orientation: oriented distally. Distal margin of harpe in lateral view: shape: acute. Lateral margin of harpe shape: widest point of harpe is at its articulation site with gonostyle-volsella complex.

**Comments:** Males and females are variable in color pattern: in smaller males the coloration is lighter; the legs except the proximal region of hind coxa and distal 2/3 of hind femur, mouthparts, scape and F1 are yellow and the rest of the body is ochre, whereas in larger

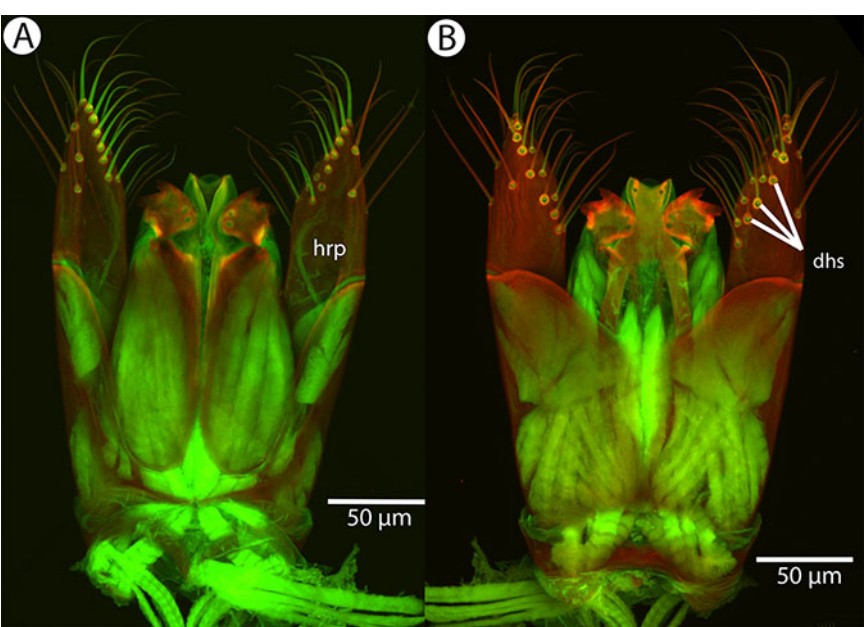

**Figure 26** **CLSM volume rendered micrographs showing the male genitalia of *Conostigmus longulus Dessart, 1997*.** (A) Ventral view. (B) Dorsal view (hrp, harpe; dhs, dorsomedial setae of harpal setal ring).

males the colors of these body parts are orange and brown. In most female specimens, the legs except the proximal region of the hind coxa and the distal 2/3 of hind femur, mouthparts, distal part of scape, pedicel and F1–F4 are yellow and the rest of the body is brown, whereas in one specimen (CAS2002193), only the distal 1/5 of the scape is yellow and the rest of the antenna is brown. The length of the preoccipital furrow is variable, from reaching the anterior 1/5 of the length of the ocellar triangle (CAS204825) to barely exceeding POL (CAS2053554). The sternaulus is present and short in larger specimens of *Conostigmus longulus* and absent from smaller specimens. The lateral propodeal carina of *Conostigmus longulus* is straight or Y-shaped and the frontal carina is distinct, sharply defined in larger and indistinct marked by a blunt edge in smaller specimens.

**Material examined:** Holotype male: MADAGASCAR: PSUC_FEM 8919 COLL. MUS. Congo Madagascar: Mandraka II-1944 A. Seyrig HOLOTYPUS Holotype Prep. microscopique n 9508/051 (deposited in MRAC). Other material (10 males, six females): MADAGASCAR: 10 males, six females. CASENT 2002193, 2009756, 2040771, 2040900, 2044193, 2046098, 2046100, 2053308, 2053554, 2053688; PSUC_FEM 79732, 79735, 79745, 79748, 79753, 79757 (deposited in CAS, MRAC).

*Conostigmus lucidus* Mikó and Trietsch sp. nov.

Figures 27, 28, 29 and 30

**Diagnosis:** *Conostigmus lucidus* sp. nov. differs from other Malagasy *Conostigmus* species in the presence of the long anterior neck of T1 (petiole neck and corresponding S1 are as long as wide in *C. lucidus* and at least about 2× as wide as long in other Malagasy *Conostigmus* species), absence of dorsomedian conjunctiva of the gonostyle/volsella complex and absence of the proximodorsal notch of cupula (both structures are present in

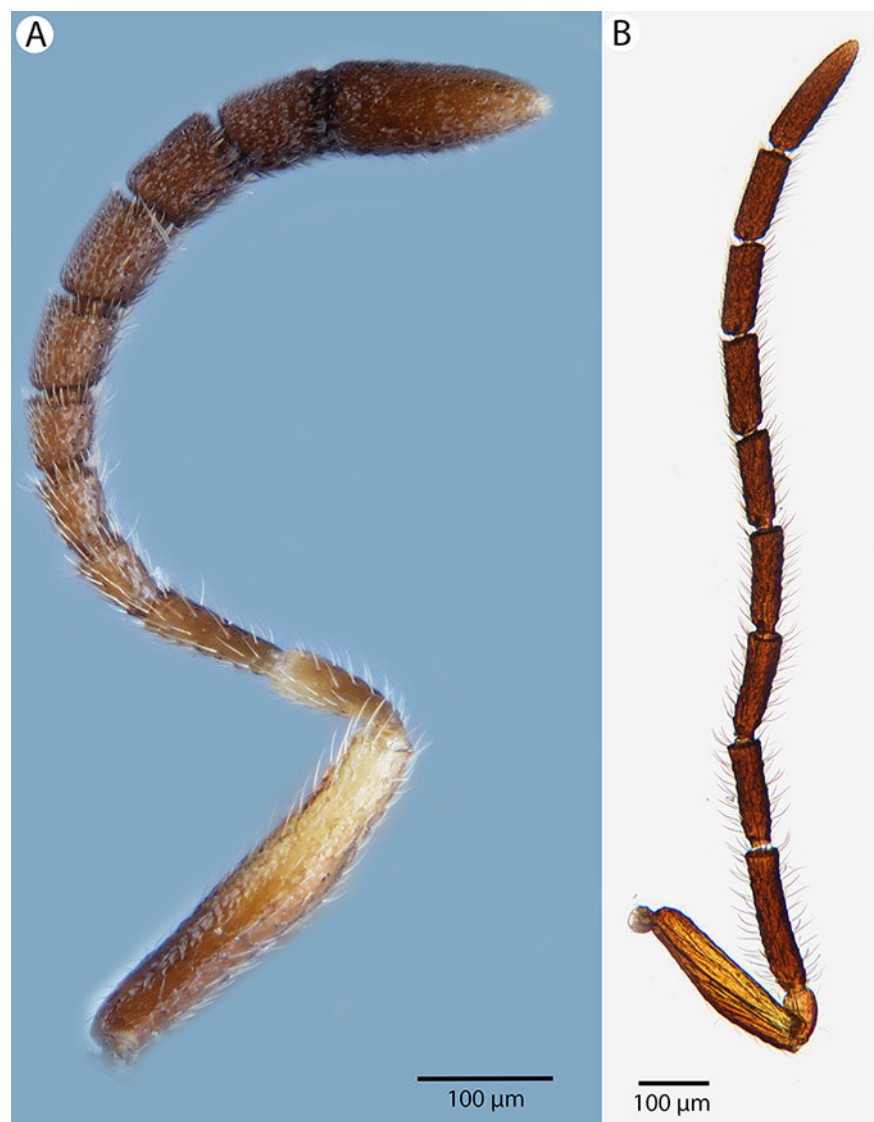

Figure 27 **Brightfield image showing the antenna of *Conostigmus lucidus* Mikó and Trietsch sp. nov., lateral view.** (A) Female. (B) Male.

other Malagasy *Conostigmus* species). The parossiculus as an independent sclerite is absent (parossiculus and gonostyle fused).

The petiole neck and corresponding first abdominal sternite is also elongated in the Oriental species *Conostigmus ampullaceus Dessart, 1997* where the petiole neck is even longer (sometimes 2× as long as wide) than in *C. lucidus*. The two species differ in numerous distinct characters such as the presence of color contrast between the black head and orange metasoma and the absence of the preoccipital lunula and preoccipital sulcus in *Conostigmus ampullaceus* (all tagmata are uniformly brown and both the preocipital lunula and preocipital suclus are present in *C. lucidus*).

**Description:** Body length: 2,100–2,600 μm. Color intensity pattern: front and middle leg lighter than distal half of scape, pedicel and tegula; cranium, distal region of

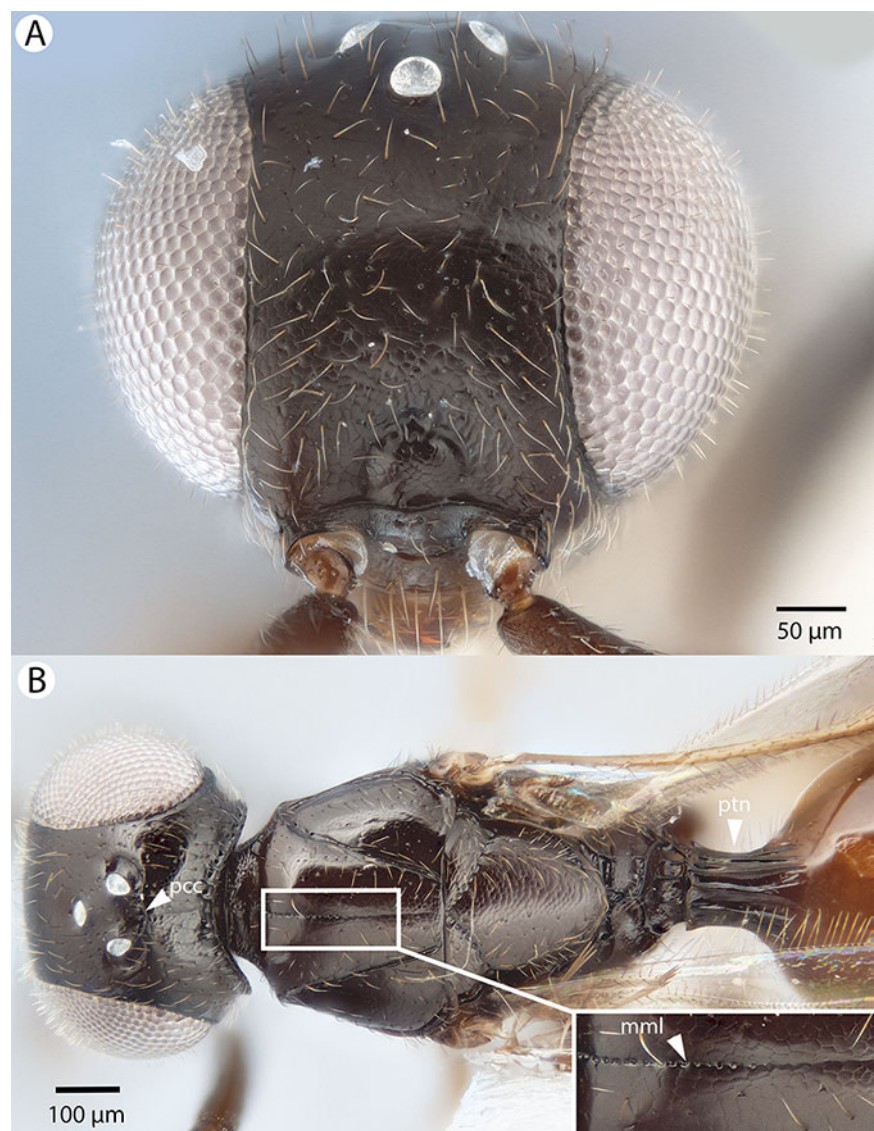

**Figure 28 Brightfield image showing the head and mesosoma of *Conostigmus lucidus* Mikó and Trietsch sp. nov.** (A) Head, anterior view. (B) Head and mesosoma, dorsal view (mml, median mesoscutal line).

flagellum, mesosoma except legs and petiole neck darker than proximal region of flagellum, hind leg and metasoma posterior to petiole neck. Color hue pattern: Distal half of scape, pedicel, fore leg and middle leg yellow; proximal part of scape, flagellum, mesosoma except front and middle leg, metasoma brown. Occipital carina sculpture: crenulate. Median flange of occipital carina count: absent. Submedial flange of occipital carina count: absent. Dorsal margin of occipital carina vs. dorsal margin of lateral ocellus in lateral view: occipital carina is ventral to lateral ocellus in lateral view. Preoccipital lunula count: present. Preoccipital carina count: absent. Preoccipital carina shape: NOT CODED. Preoccipital furrow count: present. Preoccipital furrow anterior end: Preoccipital furrow ends posterior to ocellar triangle. Postocellar carina count: present.

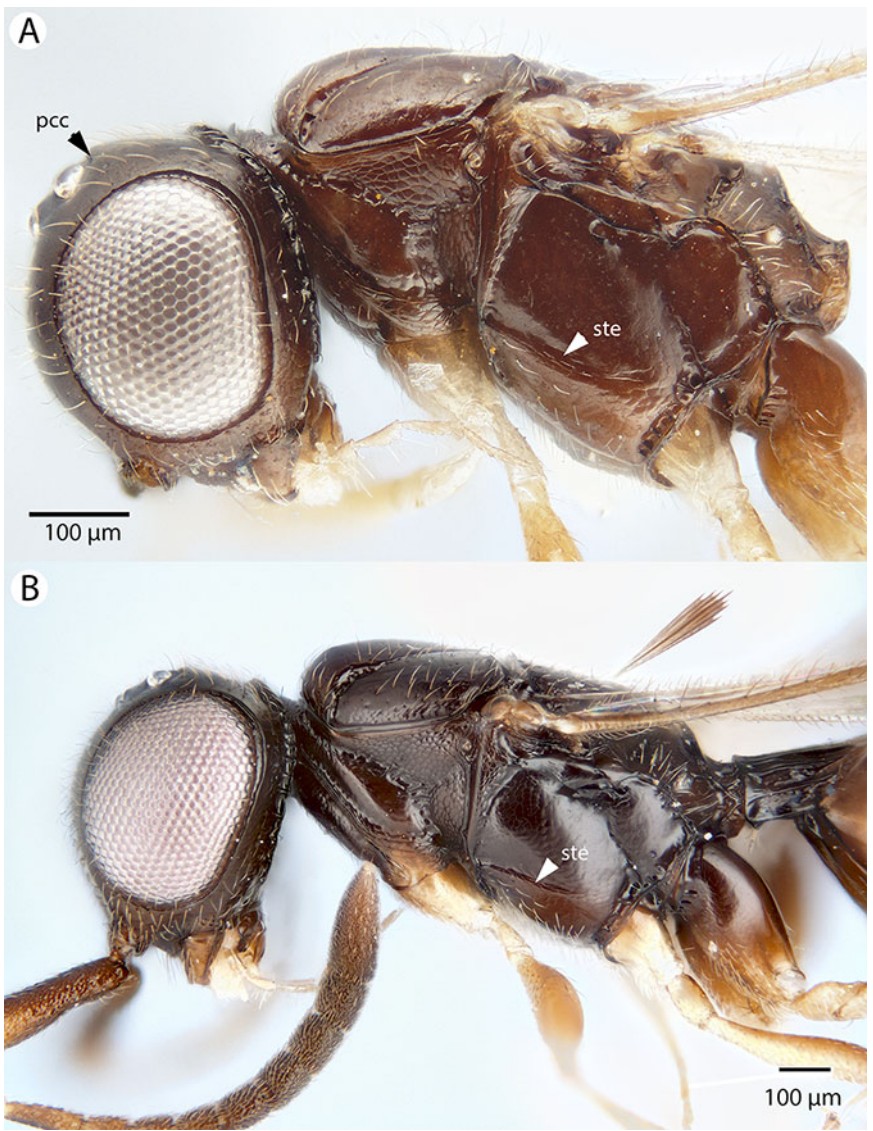

**Figure 29 Brightfield image showing the head and mesosoma of *Conostigmus lucidus* Mikó and Trietsch sp. nov., lateral view.** (A) Smaller specimen. (B) Larger specimen (ste, sternaulus).

Male OOL: POL: LOL: 2.2:1.1–1.4:1. Female OOL: POL: LOL: 1.5–2.1:1.2–1.4:1.0. HW/IOS Male: 2.1. HW/IOS Female: 2.0–2.1. Setal pit on vertex size: smaller than diameter of scutes. Transverse frontal carina count: absent. Transverse scutes on frons count: absent. Rugose region on frons count: absent. Randomly sized areolae around setal pits on frons count: absent. Antennal scrobe count: absent. Ventromedian setiferous patch and ventrolateral setiferous patch count: absent. Facial pit count: facial pit present. Supraclypeal depression count: present. Supraclypeal depression structure: present medially, inverted U-shaped. Intertorular carina count: present. Intertorular area count: present. Median region of intertorular area shape: convex. Ventral margin of antennal rim vs. dorsal margin of clypeus: not adjacent. Torulo-clypeal carina count: present. Subtorular carina count: absent. Mandibular tooth count: 2. Female flagellomere one

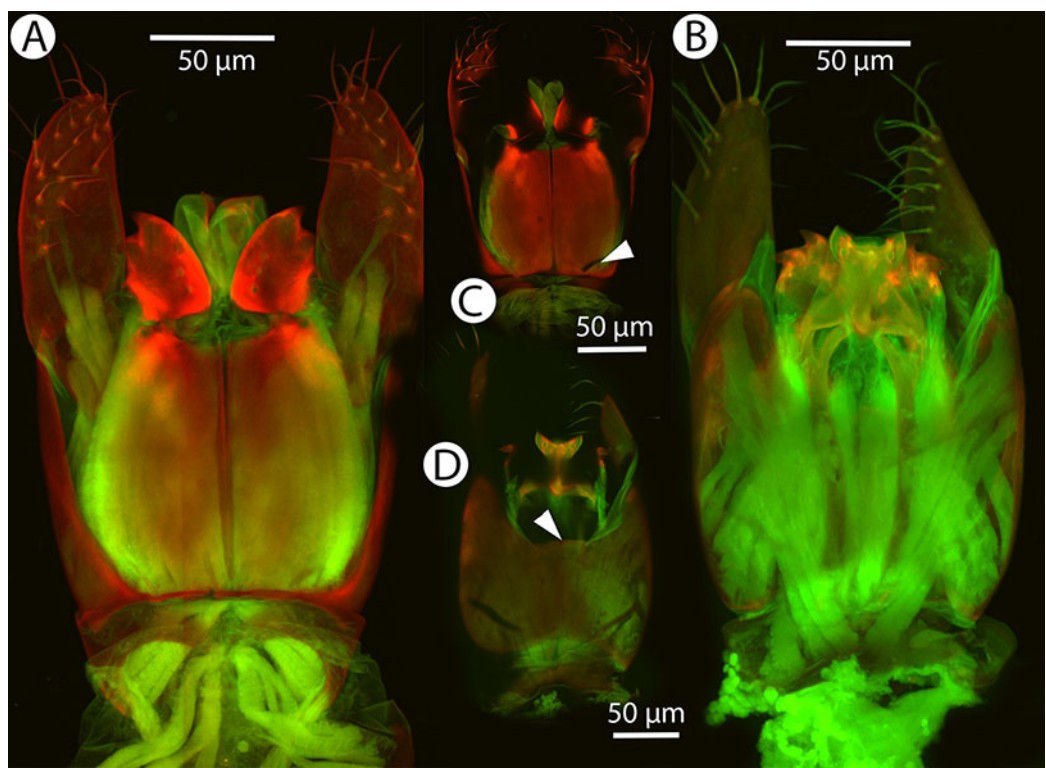

**Figure 30** CLSM volume rendered micrographs showing the male genitalia of *Conostigmus lucidus* **Mikó and Trietsch sp. nov.** (A) Ventral view. (B) Dorsal view. (C) Ventral view, partially rendered. (D) Dorsal view, partially rendered. (Arrow on (C) showing bridge connecting parossiculus with gonostyle, on (D) showing lack of dorsomedian conjectiva of gonostyle/volsella complex).

length vs. pedicel: 1.0. Female ninth flagellomere length: F9 less than F7 + F8. Sensillar patch of the male flagellomere pattern: F5–F9. Length of setae on male flagellomere vs. male flagellomere width: setae shorter than width of flagellomeres. Male flagellomere one length vs. male second flagellomere length: 1.4. Male flagellomere one length vs. pedicel length: 2.5. Ventrolateral invagination of the pronotum count: present. Scutes on posterior region of mesoscutum and dorsal region of mesoscutellum convexity: flat. Notaulus posterior end location: adjacent to transscutal articulation. Median mesoscutal sulcus posterior end: not adjacent to transscutal articulation (ends anterior to transscutal articulation). Scutoscutellar sulcus vs. transscutal articulation: adjacent. Axillular carina count: absent. Axillular carina shape: NOT CODED. Epicnemium posterior margin shape: anterior discrimenal pit present; epicnemial carina curved. Epicnemial carina count: complete. Sternaulus count: present. Sternaulus length: elongate, exceeding 3/4 of mesopleuron length at level of sternaulus. Speculum ventral limit: not extending ventrally of pleural pit line. Mesometapleural sulcus count: present. Metapleural carina count: present. Transverse line of the metanotum-propodeum vs. antecostal sulcus of the first abdominal tergum: adjacent sublaterally. Lateral propodeal carina count: present. Lateral propodeal carina shape: straight (left and right lateral propodeal carinae compose a carina that is not broken medially). Anteromedian projection of the metanoto-propodeo-metapecto-mesopectal complex count: absent. S1 length vs. shortest width: S1 longer than

wide. Transverse carina on petiole shape: straight. Distal margin of male S9 shape: convex. Proximolateral corner of male S9 shape: blunt. Cupula length vs. gonostyle-volsella complex length: cupula less than 1/2 the length of gonostyle-volsella complex in lateral view. Proximodorsal notch of cupula count: absent. Proximodorsal notch of cupula shape: NOT CODED. Proximolateral projection of the cupula shape: NOT CODED. Proximodorsal notch of cupula width vs. length: NOT CODED. Distodorsal margin of cupula shape: straight. Dorsomedian conjunctiva of the gonostyle-volsella complex length relative to length of gonostyle-volsella complex: NOT CODED. Dorsomedian conjunctiva of the gonostyle-volsella complex count: absent. Distal end of dorsomedian conjunctiva of the gonostyle-volsella complex shape: NOT CODED. Parossiculus count (parossiculus and gonostipes fusion): absent (fused with the gonostipes). Apical parossiculal seta number: one. Distal projection of the parossiculus count: absent. Distal projection of the penisvalva count: absent. Dorsal apodeme of penisvalva count: absent. Harpe length: harpe shorter than gonostipes in lateral view. Distodorsal setae of sensillar ring of harpe length vs. harpe width in lateral view: setae as long or shorter than harpe width. Distodorsal setae of sensillar ring of harpe orientation: distomedially. Sensillar ring area of harpe orientation: dorsomedially. Lateral setae of harpe count: absent. Lateral setae of harpe orientation: oriented distally. Distal margin of harpe in lateral view: shape: blunt. Lateral margin of harpe shape: widest point of harpe is at its articulation site with gonostyle-volsella complex.

**Etymology:** The species epithet is derived from the Latin *lucidus* which means "shining," in reference to the shining appearance of the cuticle due to the weak microsculpture of a large portion of the body.

**Material examined:** Holotype male: MADAGASCAR: 3 km 41°NE Andranomay, 11.5 km 147°SSE Anjozobe, sifted litter in montane rainforest, 3–13.12.2000, Fisher, Griswold et al., CASENT 2001309 (deposited in CAS). Paratypes (one male, six females): MADAGASCAR: one male, six females. CASENT 2002181, 2004743, 2004751, 2040895, 2045754, 2046026, 2046176 (deposited in CAS, MRAC).

*Conostigmus macrocupula* Mikó and Trietsch sp. nov.

Figures 31, 32, 33, 34, 35 and 36

**Diagnosis (male):** *Conostigmus macrocupula* sp. nov. differs from other *Conostigmus* species in the elongate cupula, which is as long as the gonostyle volsella complex (the cupula is less than half as long as the gonostyle volsella complex in other *Conostigmus* species).

The only other Ceraphronoidea species with an unusually long cupula is *Dendrocerus phallocrates* Dessart, 1987.

**Description:** Body length: 1,270–1,300 μm. Color intensity pattern: flagellum, tibiae and tarsi lighter than scape, pedicel, mandible, tegula, coxae and femora. Color hue pattern: Cranium, mesosoma except legs and metasoma except gonostipes and volsella ochre; antenna, legs, mandible, gonostipes and volsella yellow. Occipital carina sculpture: crenulate. Median flange of occipital carina count: absent. Submedial flange of occipital carina count: absent. Dorsal margin of occipital carina vs. dorsal margin of lateral ocellus in

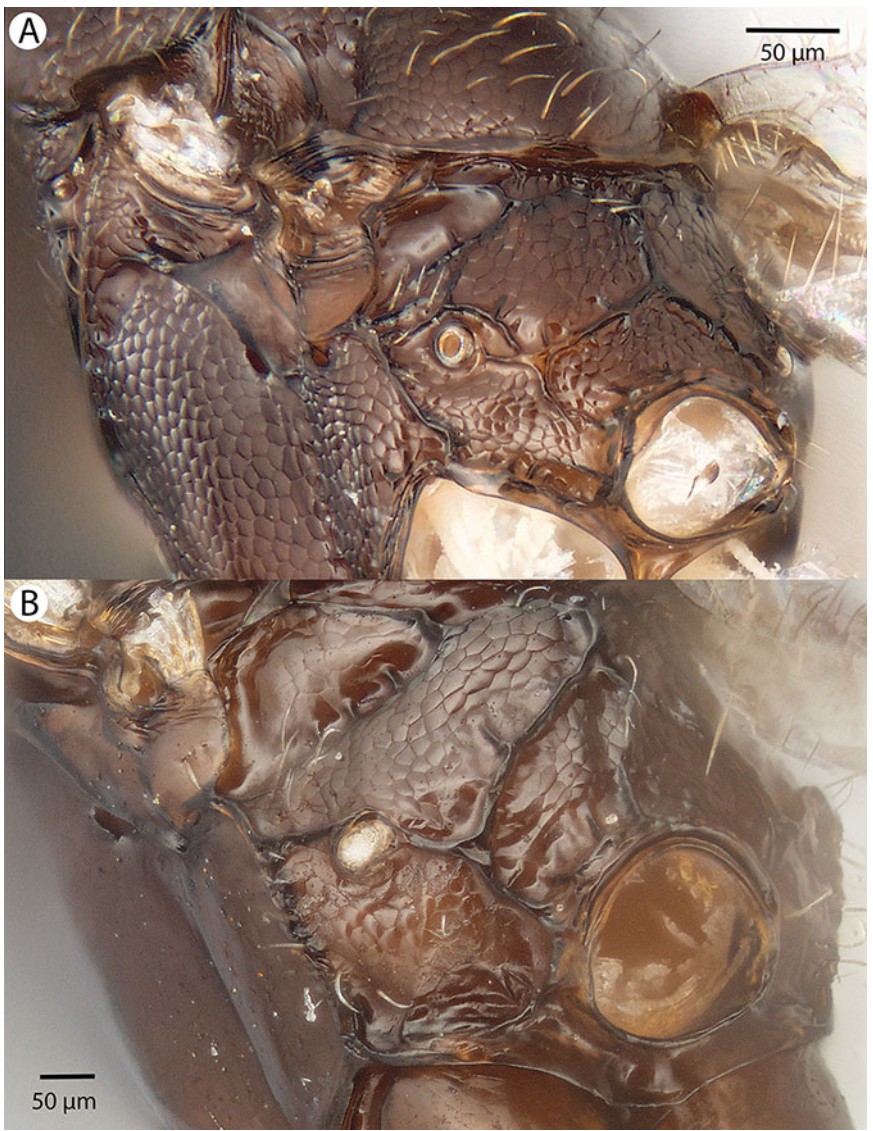

**Figure 31 Brightfield image showing the mesosoma of *Conostigmus* species, posterolateral view.** (A) *Conostigmus missyhazenae* Mikó and Trietsch sp. nov. (B) *Conostigmus lucidus* Mikó and Trietsch sp. nov.

lateral view: occipital carina is ventral to lateral ocellus in lateral view. Preoccipital lunula count: present. Preoccipital carina count: absent. Preoccipital carina shape: NOT CODED. Preoccipital furrow count: present. Preoccipital furrow anterior end: Preoccipital furrow ends posterior to ocellar triangle. Postocellar carina count: absent. Male OOL: POL: LOL: 2.0–2.1:1.7–1.8:1. Female OOL: POL: LOL: NOT CODED. HW/IOS Male: 1.8–2.0. HW/IOS Female: NOT CODED. Setal pit on vertex size: smaller than diameter of scutes. Transverse frontal carina count: absent. Transverse scutes on frons count: absent. Rugose region on frons count: absent. Randomly sized areolae around setal pits on frons count: absent. Antennal scrobe count: absent. Ventromedian setiferous patch and ventrolateral setiferous patch count: absent. Facial pit count: facial pit present. Supraclypeal depression count: present. Supraclypeal depression structure: absent

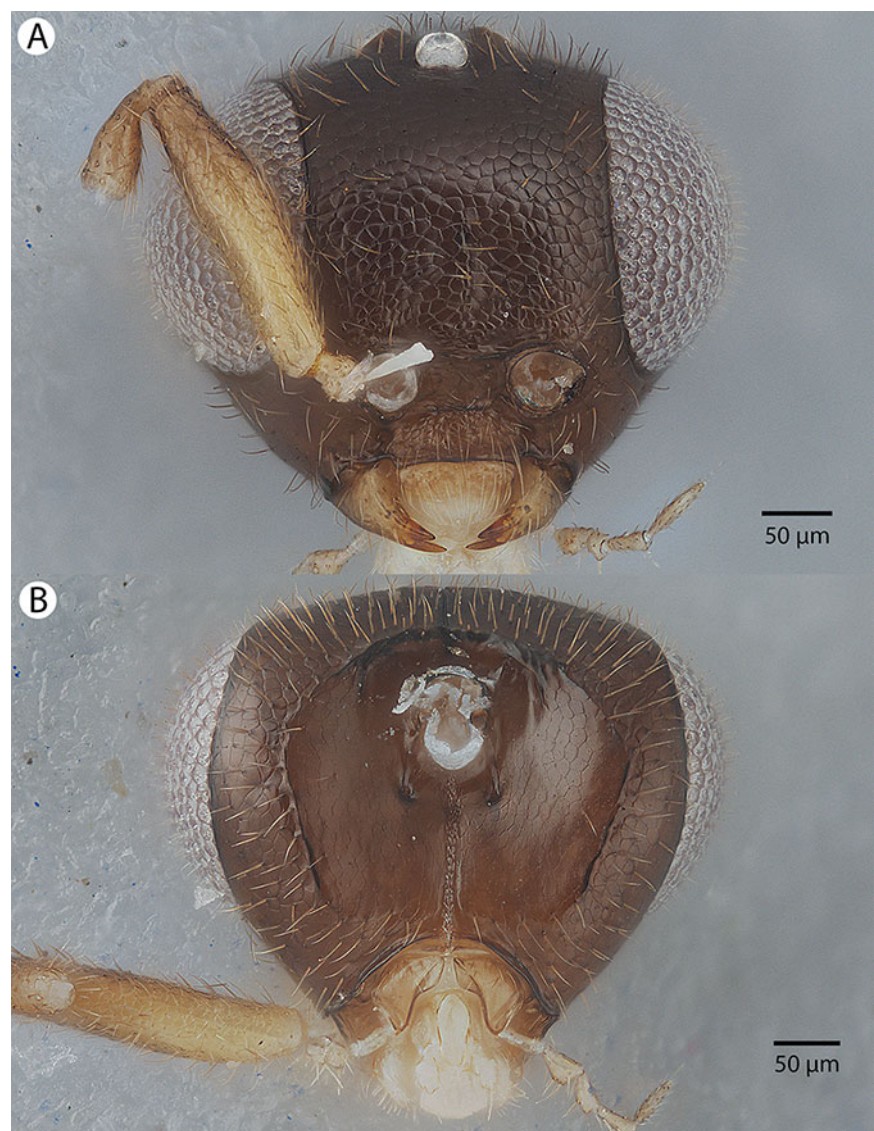

**Figure 32 Brightfield image showing the head of *Conostigmus macrocupula* Mikó and Trietsch sp. nov.** (A) Anterior view. (B) Posterior view.

medially, represented by two grooves laterally of facial pit. Intertorular carina count: present. Intertorular area count: present. Median region of intertorular area shape: flat. Ventral margin of antennal rim vs. dorsal margin of clypeus: not adjacent. Torulo-clypeal carina count: present; absent. Subtorular carina count: absent. Mandibular tooth count: 2. Female flagellomere one length vs. pedicel: NOT CODED. Female ninth flagellomere length: F9 less than F7 + F8. Sensillar patch of the male flagellomere pattern: F5–F9. Length of setae on male flagellomere vs. male flagellomere width: setae shorter than width of flagellomeres. Male flagellomere one length vs. male second flagellomere length: 1.2–1.3. Male flagellomere one length vs. pedicel length: 1.2–1.3. Ventrolateral invagination of the pronotum count: present. Scutes on posterior region of mesoscutum and dorsal region of mesoscutellum convexity: flat. Notaulus posterior end location: adjacent to transscutal

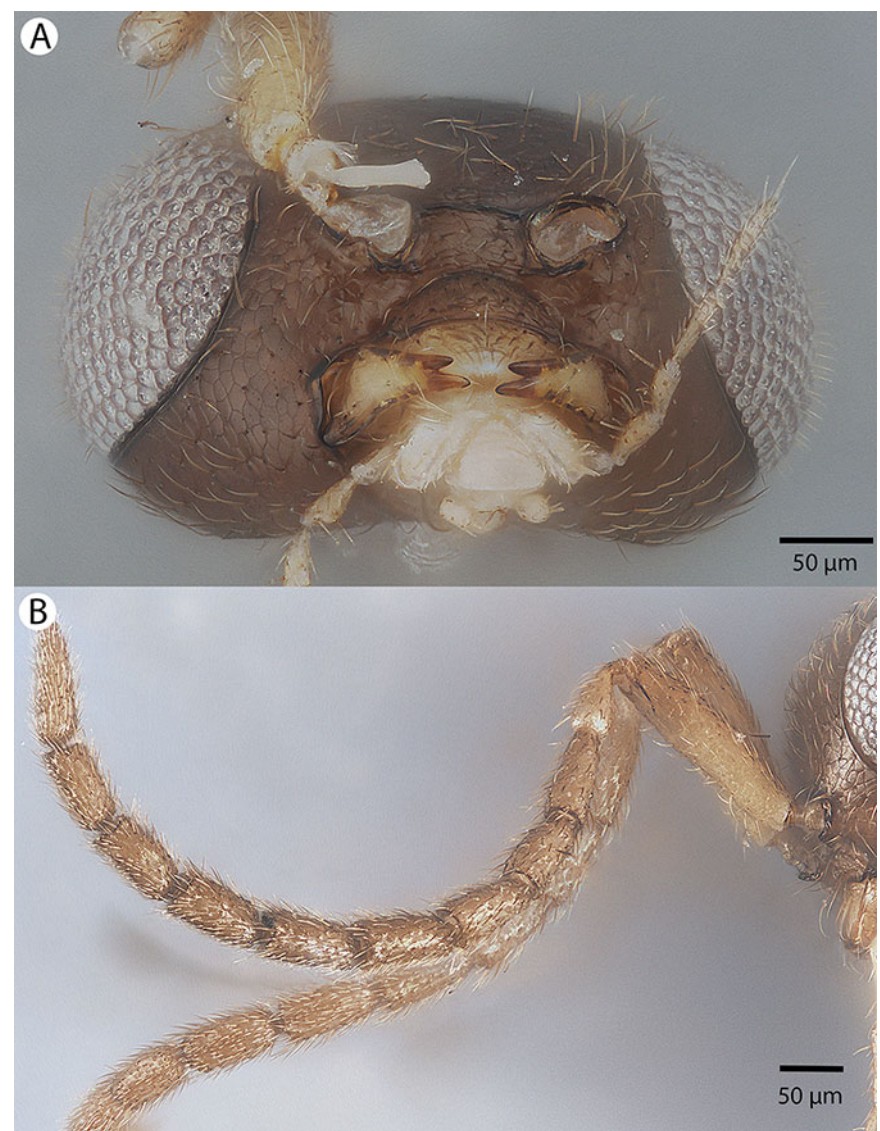

**Figure 33** Brightfield image showing the head and antenna of *Conostigmus macrocupula* Mikó and Trietsch sp. nov. (A) Head, ventral view. (B) Antenna, lateral view.

articulation. Median mesoscutal sulcus posterior end: adjacent to transscutal articulation. Scutoscutellar sulcus vs. transscutal articulation: adjacent. Axillular carina count: absent. Axillular carina shape: NOT CODED. Epicnemium posterior margin shape: anterior discrimenal pit present; epicnemial carina curved. Epicnemial carina count: complete. Sternaulus count: absent. Sternaulus length: NOT CODED. Speculum ventral limit: not extending ventrally of pleural pit line. Mesometapleural sulcus count: absent. Metapleural carina count: present. Transverse line of the metanotum-propodeum vs. antecostal sulcus of the first abdominal tergum: adjacent sublaterally. Lateral propodeal carina count: present. Lateral propodeal carina shape: inverted "Y" (left and right lateral propodeal are adjacent medially posterior to antecostal sulcus of the first abdominal tergum, and connected to the antecostal sulcus by a median carina representing the median branch of

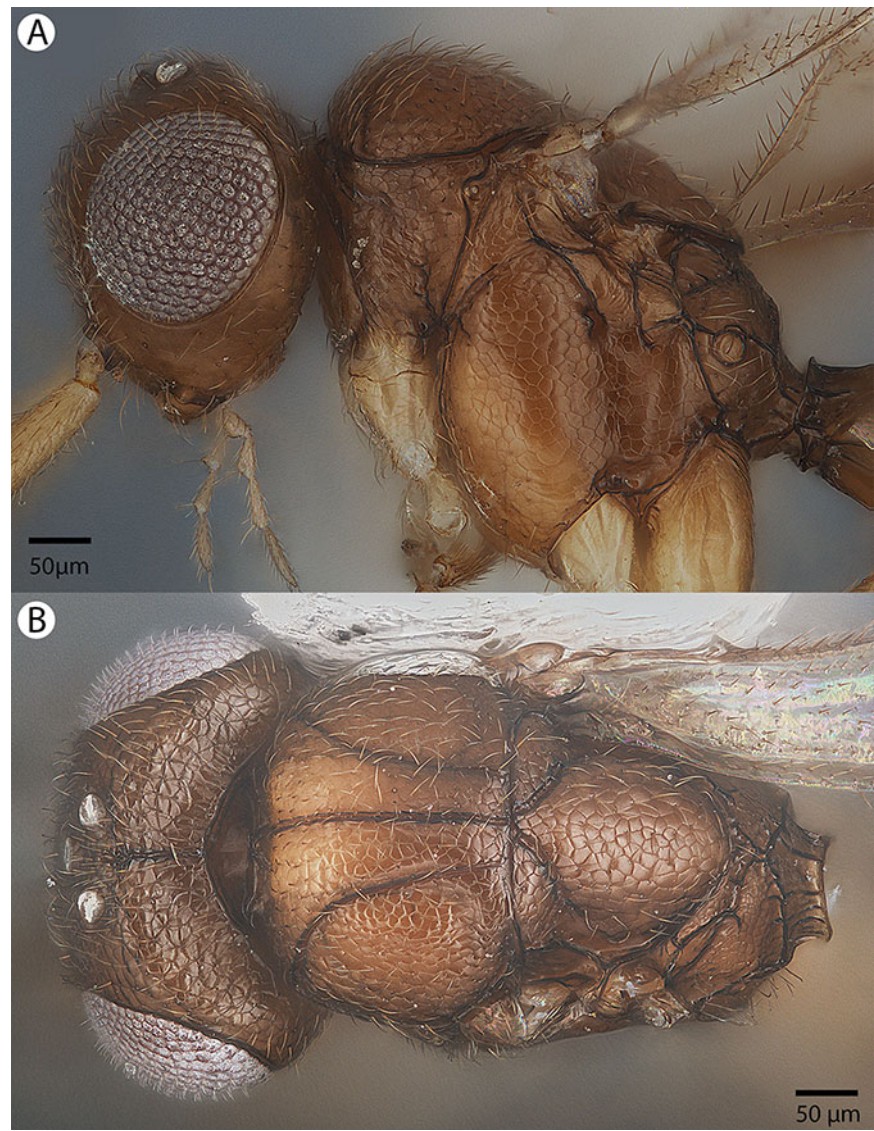

**Figure 34 Brightfield image showing the head and mesosoma of *Conostigmus macrocupula* Mikó and Trietsch sp. nov.** (A) Lateral ventral view. (B) Dorsal view.

the inverted "Y"). Anteromedian projection of the metanoto-propodeo-metapecto-mesopectal complex count: absent. S1 length vs. shortest width: S1 wider than long. Transverse carina on petiole shape: straight. Distal margin of male S9 shape: concave. Proximolateral corner of male S9 shape: blunt. Cupula length vs. gonostyle-volsella complex length: cupula as long as gonostyle-volsella complex in lateral view. Proximodorsal notch of cupula count: present. Proximodorsal notch of cupula shape: arched. Proximolateral projection of the cupula shape: blunt. Proximodorsal notch of cupula width vs. length: at least two times as long as wide. Distodorsal margin of cupula shape: straight. Dorsomedian conjunctiva of the gonostyle-volsella complex length relative to length of gonostyle-volsella complex: dorsomedian conjunctiva not extending 2/3 of length of gonostyle-volsella complex in dorsal view. Dorsomedian conjunctiva of the

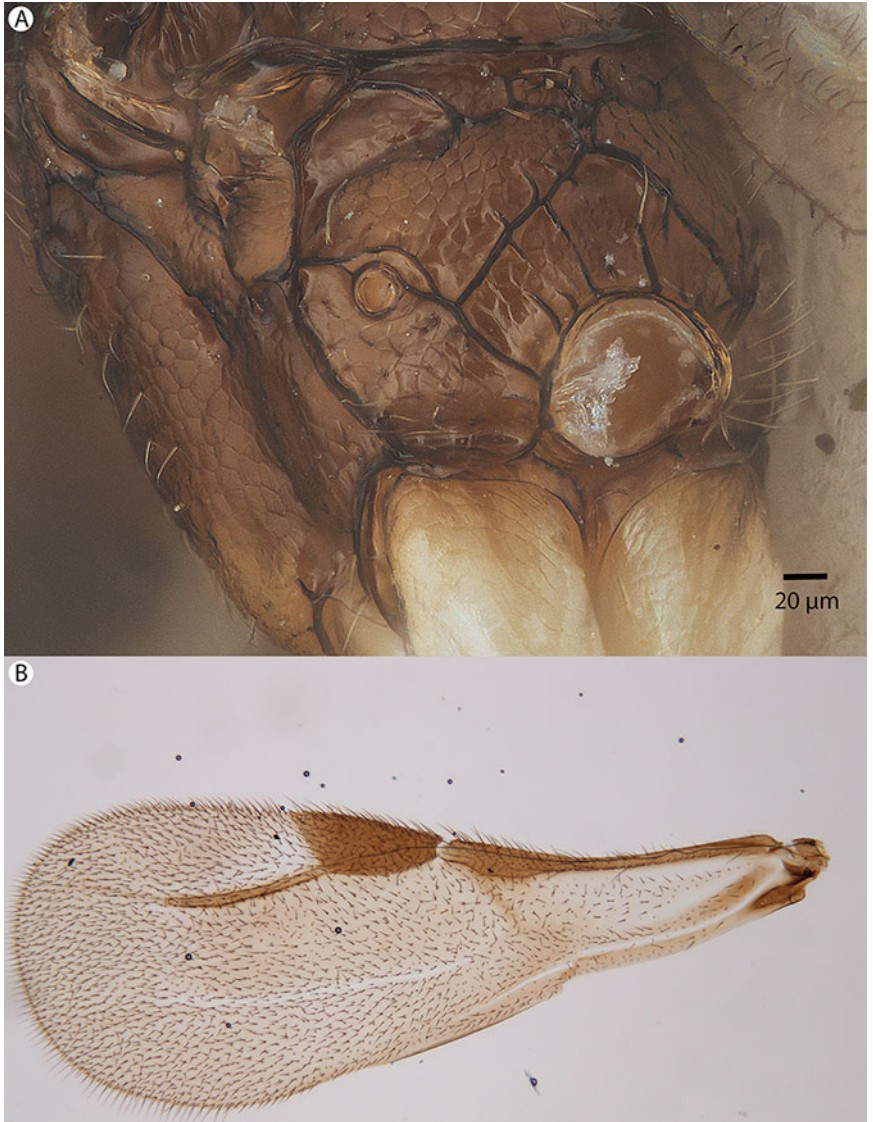

**Figure 35 Brightfield image showing the mesosoma and wing of *Conostigmus macrocupula* Mikó and Trietsch sp. nov.** (A) Mesosoma, posterolateral view. (B) Fore wing.

gonostyle-volsella complex count: present. Distal end of dorsomedian conjunctiva of the gonostyle-volsella complex shape: acute. Parossiculus count (parossiculus and gonostipes fusion): present (not fused with the gonostipes). Apical parossicular seta number: two. Distal projection of the parossiculus count: absent. Distal projection of the penisvalva count: absent. Dorsal apodeme of penisvalva count: absent. Harpe length: harpe as long as gonostipes in lateral view. Distodorsal setae of sensillar ring of harpe length vs. harpe width in lateral view: setae two times as long as harpe width. Distodorsal setae of sensillar ring of harpe orientation: dorsally. Sensillar ring area of harpe orientation: medially. Lateral setae of harpe count: present. Lateral setae of harpe orientation: oriented distally. Distal margin of harpe in lateral view: shape: blunt. Lateral margin of harpe shape: widest point of harpe is at its articulation site with gonostyle-volsella complex.

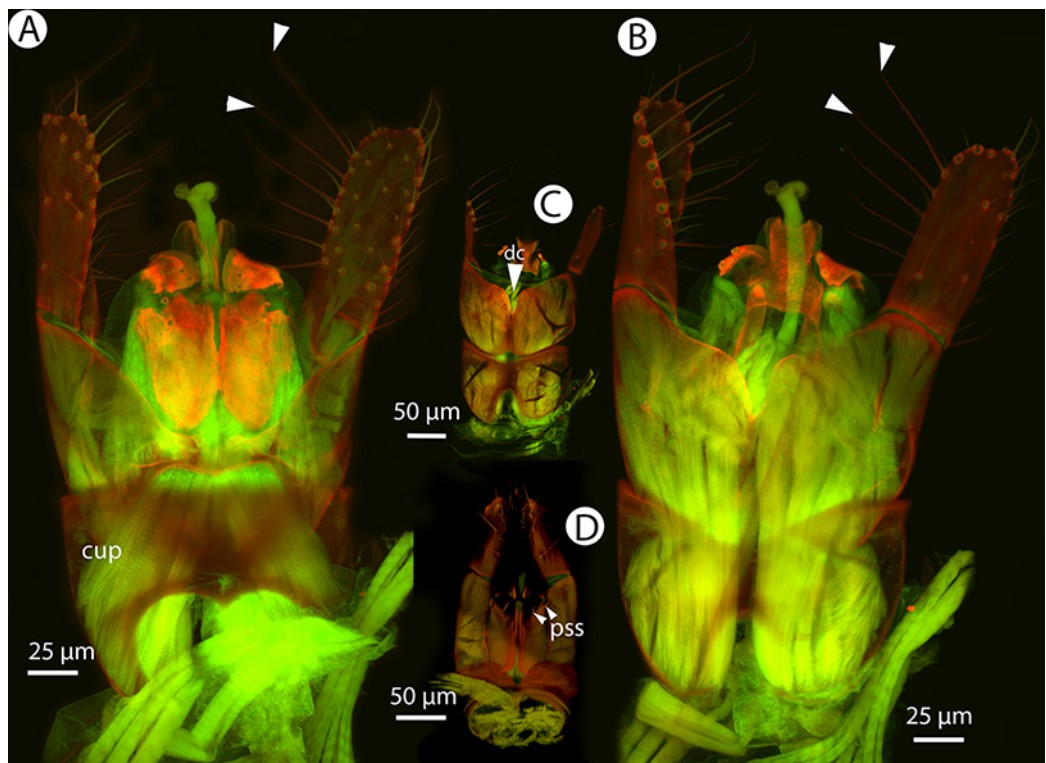

**Figure 36 CLSM volume rendered micrographs showing the male genitalia of *Conostigmus macrocupula* Mikó and Trietsch sp. nov.** (A) Ventral view. (B) Dorsal view (cup, cupula; pss, parosiculal setae; dc, dorsomedial conjunctiva of gonostyle/volsella complex).

**Etymology:** The species epithet is derived from the Greek macro (large) and the Latin noun cupula (small, inverted cup). The latin name of the species refers to the large cupula that is as long as the gonostyle/volsella complex.

**Material examined:** Holotype male: MADAGASCAR: Parc National Ranomafana, Belle Vue at Talatakely, Malaise, secondary tropical forest, 12–19.2.2002, R. Harin'Hala, CASENT 2046023 (deposited in CAS). Paratypes (seven males): MADAGASCAR: seven males. CASENT 2046022, 2046025, 2046181, 2053451; PSUC_FEM 79741–79742, 79750 (CAS, MRAC).

*Conostigmus madagascariensis* Mikó and Trietsch sp. nov.

Figures 37, 38, 39, 40, 41, 42 and 43

**Diagnosis:** *Conostigmus madagascariensis* sp. nov. is most similar to *C. fianarantsoaensis* sp. nov. among Malagasy *Conostigmus*. *Conostigmus madagascariensis* differs from *C. fianarantsoaensis* in the presence of two teeth on the mandibles, flagellar setae longer than the flagellomere width (in *C. fianarantsoaensis*, flagellar setae are shorter than flagellomere width), acute proximolateral projection of cupula (blunt in *C. fianarantsoaensis*), arched proximodorsal notch of cupula (v-shaped (notched) in *C. fianarantsoaensis*), acute distal end of dorsomedial conjunctiva of gonostyle/volsella

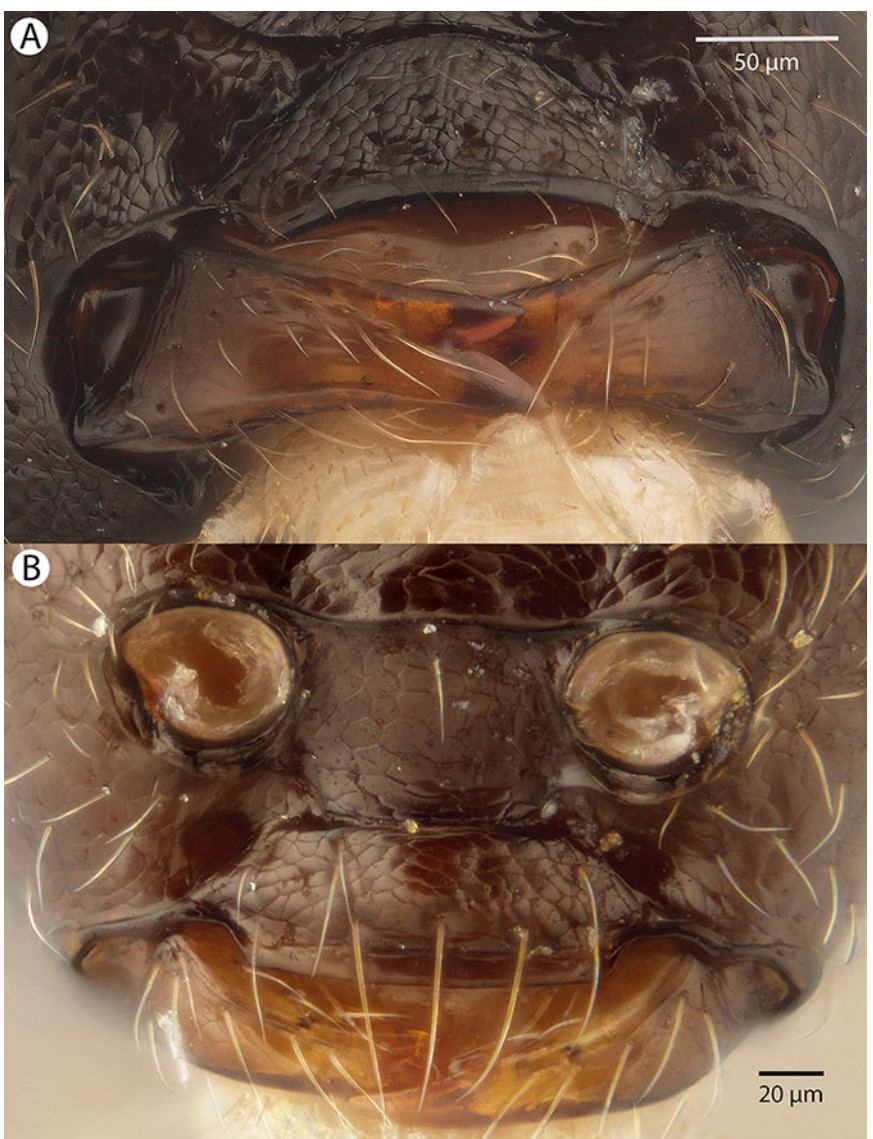

**Figure 37 Brightfield image showing the intraspecific variability in mandible structure of *Conostigmus madagascariensis* Mikó and Trietsch sp. nov.**

complex (blunt in *C. fianarantsoaensis*), and blunt distal margin of harpe in lateral view (acute in *C. fianarantsoaensis*).

**Description:** Body length: 1,500–2,700 µm. Color intensity pattern: metasoma and mandible lighter than mesosoma. Color hue pattern: Antenna except pedicel, cranium, mesosoma except fore and middle legs and metasoma brown; fore and middle legs, tegula, pedicel, maxillary palp and labial palp yellow; F3–F8, cranium, mandible, metasoma, tegula brown; legs, except brown proximal region of metacoxa and distal region of metafemur, scape, pedicel, F1–F4 yellow; Antenna except pedicel and scape, cranium, mesosoma except fore and middle legs and distal region of metacoxa, and metasoma brown; fore and middle legs, tegula, pedicel, scape, proximal part of metacoxa, palpus maxillaris, and palpus labialis yellow; Antenna except pedicel and scape, cranium,

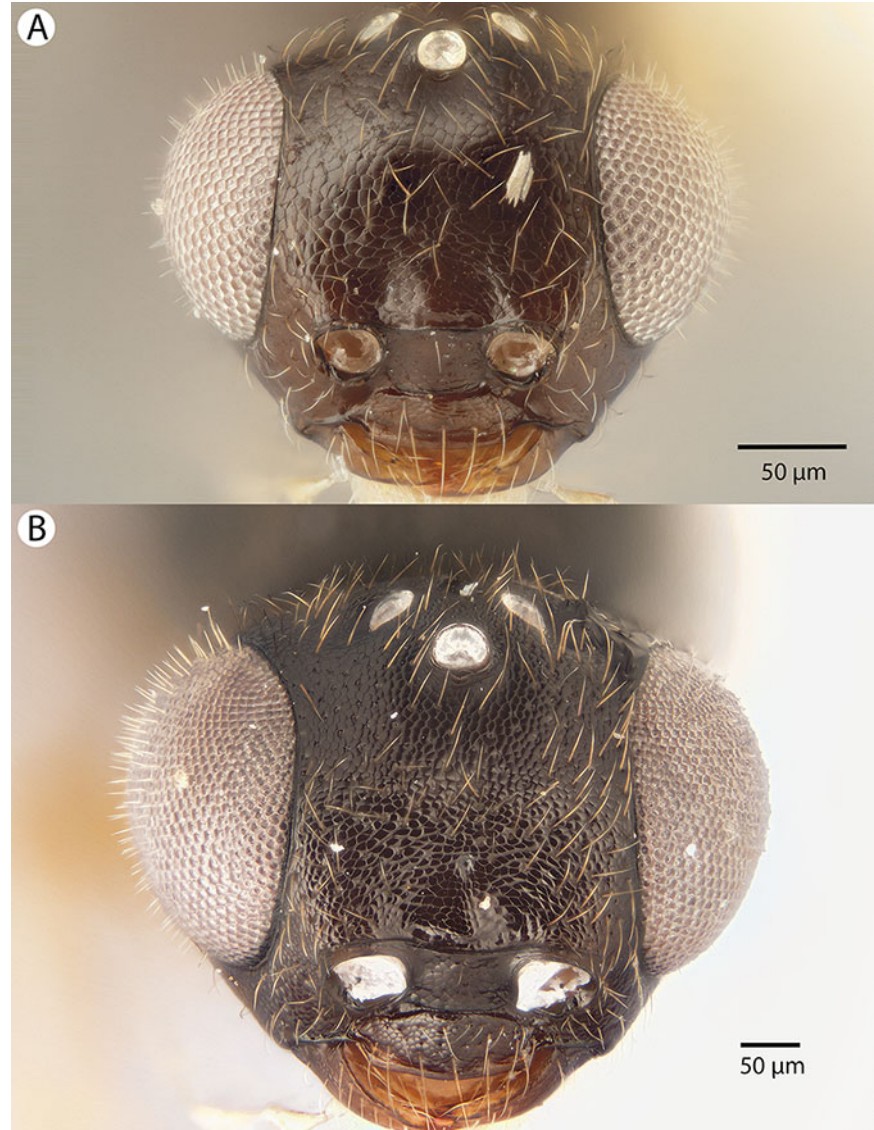

**Figure 38 Brightfield image showing the intraspecific variability in anterior head morphology of** *Conostigmus madagascariensis* **Mikó and Trietsch sp. nov.**

mesosoma except fore and middle legs and distal region of metacoxa, and metasoma brown; fore and middle legs, tegula, pedicel, scape, maxillary palp, and labial palp yellow; Antenna except pedicel, cranium, mesosoma except fore and middle legs and distal region of metacoxa, and metasoma brown; fore and middle legs, tegula, pedicel, proximal region of metacoxa, maxillary palp, and labial palp yellow; Scape, F4–F8, cranium, mandible, metasoma, tegula brown; legs, except brown proximal region of metacoxa and distal region of metafemur, pedicel, F1–F3 yellow; F1–F8, cranium, mandible, metasoma, tegula brown; legs, scape, pedicel yellow. Occipital carina sculpture: crenulate. Median flange of occipital carina count: absent. Submedial flange of occipital carina count: absent. Dorsal margin of occipital carina vs. dorsal margin of lateral ocellus in lateral view: occipital carina is ventral to lateral ocellus in lateral view. Preoccipital lunula count:

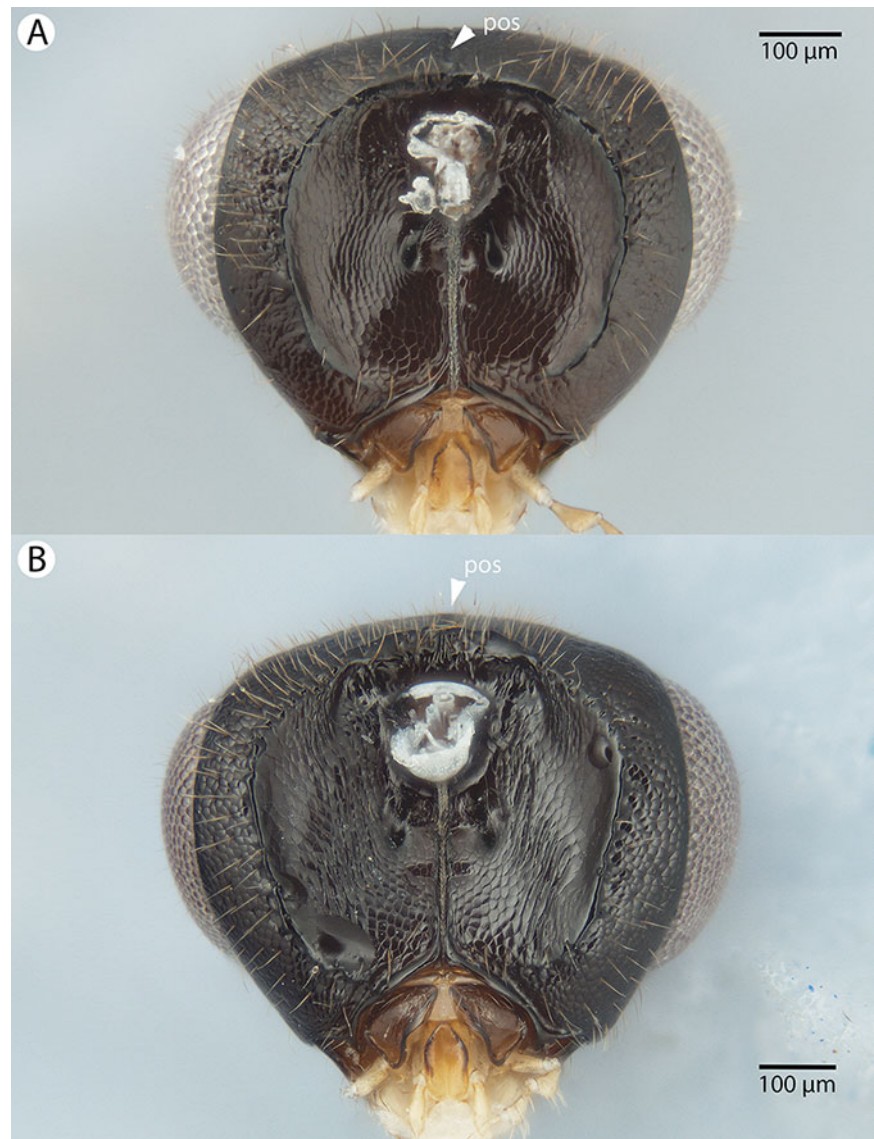

**Figure 39 Brightfield image showing the head of Conostigmus species, posterior view.** (A) *Conostigmus madagascariensis* Mikó and Trietsch sp. nov. (B) *Conostigmus fianarantsoaensis* Mikó and Trietsch sp. nov. (pos, preoccipital furrow).

present. Preoccipital carina count: absent. Preoccipital carina shape: NOT CODED. Preoccipital furrow count: present. Preoccipital furrow anterior end: Preoccipital furrow ends posterior to ocellar triangle. Postocellar carina count: absent. Male OOL: POL: LOL: 1.8–2:1.7–1.8:1. Female OOL: POL: LOL: 1.4:1.6–1.7:1.0. HW/IOS Male: 1.6–1.9. HW/IOS Female: 2.3. Setal pit on vertex size: smaller than diameter of scutes. Transverse frontal carina count: absent. Transverse scutes on frons count: absent. Rugose region on frons count: absent. Randomly sized areolae around setal pits on frons count: absent. Antennal scrobe count: absent. Ventromedian setiferous patch and ventrolateral setiferous patch count: absent. Facial pit count: facial pit present. Supraclypeal depression count: present. Supraclypeal depression structure: present medially, inverted U-shaped; absent
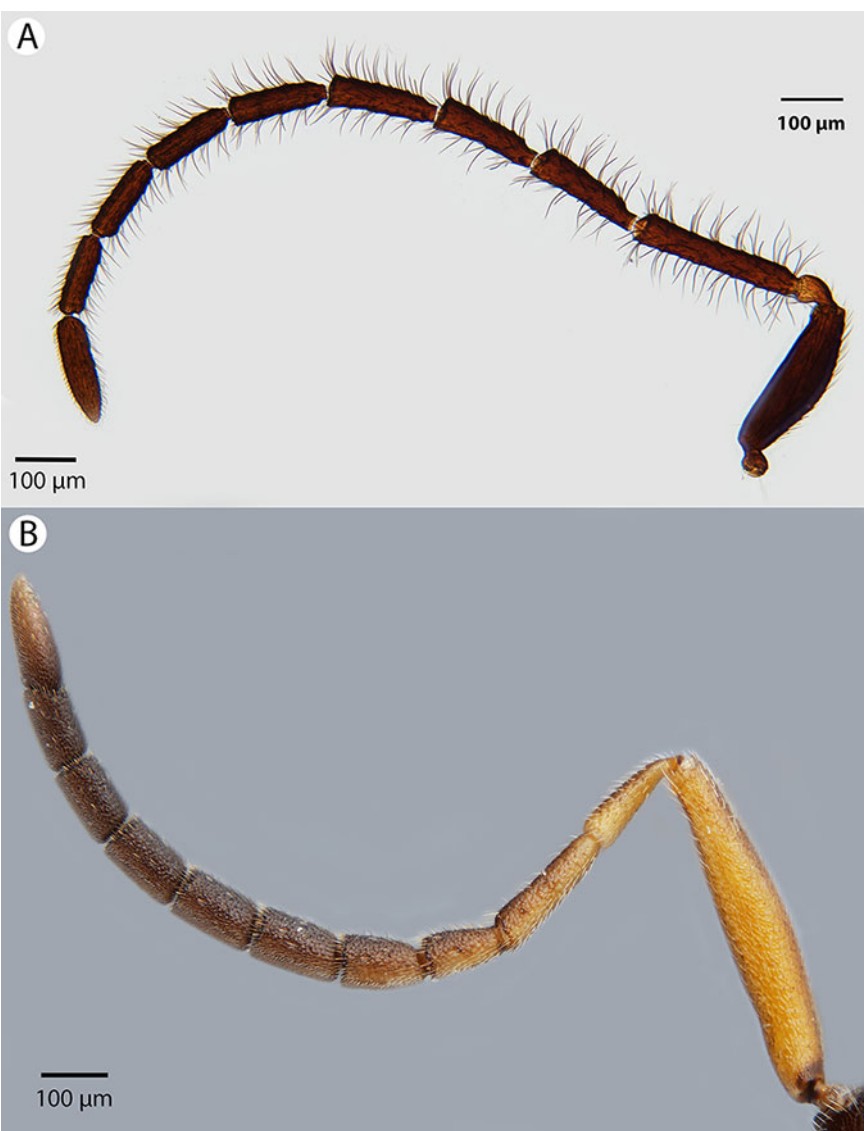

**Figure 40 Brightfield image showing the antenna of *Conostigmus madagascariensis* Mikó and Trietsch sp. nov.** (A) Male. (B) Female.

medially, represented by two grooves laterally of facial pit. Intertorular carina count: present. Intertorular area count: present. Median region of intertorular area shape: flat. Ventral margin of antennal rim vs. dorsal margin of clypeus: not adjacent. Torulo-clypeal carina count: present. Subtorular carina count: absent. Mandibular tooth count: 2. Female flagellomere one length vs. pedicel: 0.8–1.2. Female ninth flagellomere length: F9 less than F7 + F8. Sensillar patch of the male flagellomere pattern: F5–F9. Length of setae on male flagellomere vs. male flagellomere width: setae longer than width of flagellomeres. Male flagellomere one length vs. male second flagellomere length: 1.2–1.5. Male flagellomere one length vs. pedicel length: 4–4.2. Ventrolateral invagination of the pronotum count: present. Scutes on posterior region of mesoscutum and dorsal region of mesoscutellum convexity: flat. Notaulus posterior end location: adjacent to transscutal

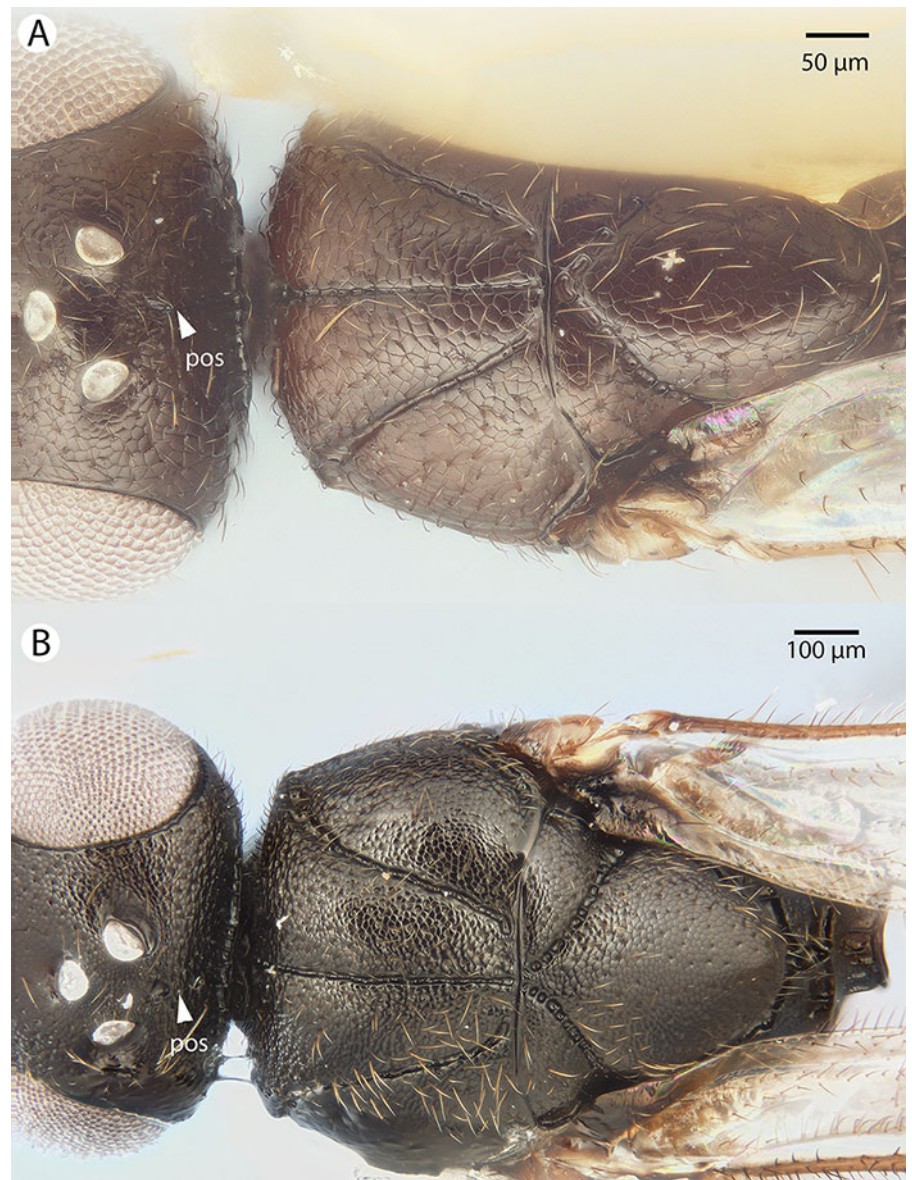

**Figure 41** **Brightfield image showing the inraspecific variability of the mesosoma of *Conostigmus madagascariensis* Mikó and Trietsch sp. nov., dorsal view.** (A) Smaller specimen. (B) Larger specimen (pos, postocellar sulcus).

articulation. Median mesoscutal sulcus posterior end: adjacent to transscutal articulation. Scutoscutellar sulcus vs. transscutal articulation: adjacent. Axillular carina count: absent. Axillular carina shape: NOT CODED. Epicnemium posterior margin shape: anterior discrimenal pit present; epicnemial carina curved. Epicnemial carina count: interupted medially; complete. Sternaulus count: present. Sternaulus length: short, not reaching 1/2 of mesopleuron length at level of sternaulus. Speculum ventral limit: not extending ventrally of pleural pit line. Mesometapleural sulcus count: present. Metapleural carina count: present. Transverse line of the metanotum-propodeum vs. antecostal sulcus of the first abdominal tergum: adjacent sublaterally. Lateral propodeal carina count: present. Lateral

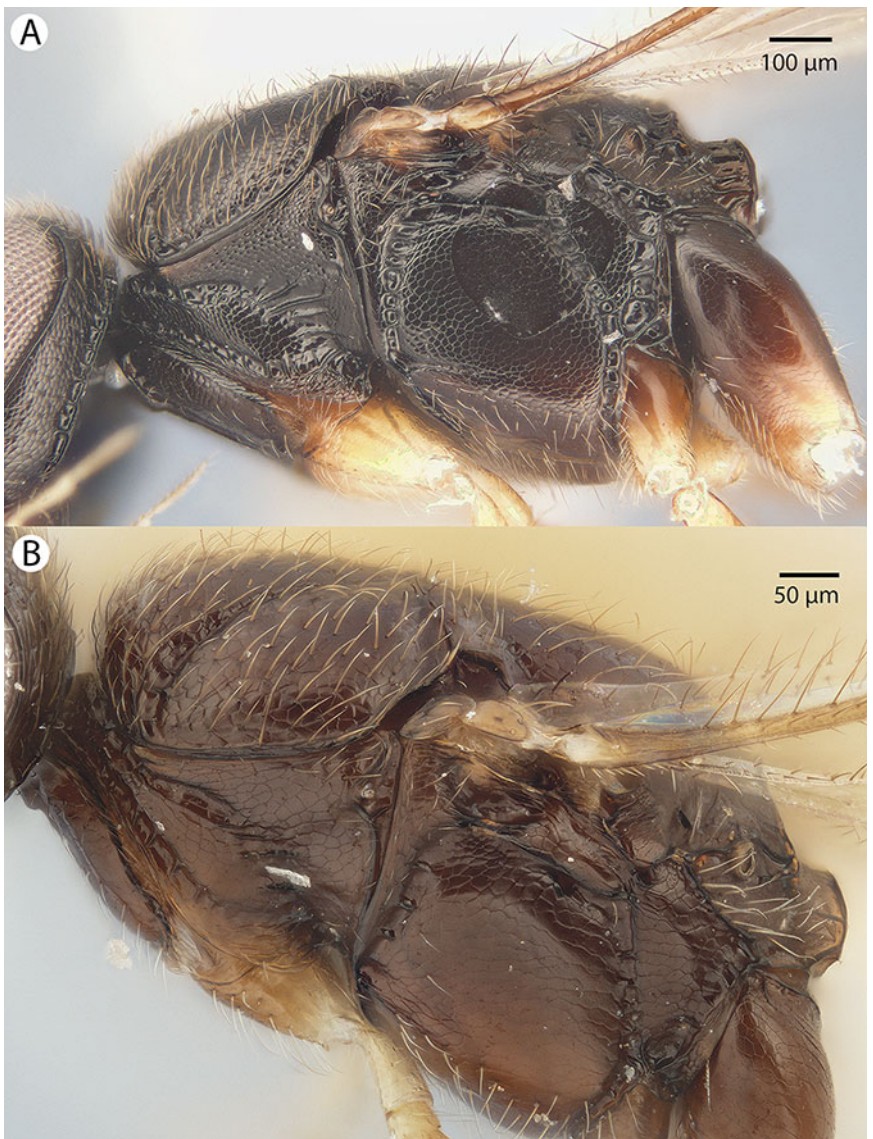

**Figure 42 Brightfield image showing the inraspecific variability of the mesosoma of *Conostigmus madagascariensis* Mikó and Trietsch sp. nov., lateral view.** (A) Larger specimen. (B) Smaller specimen.

propodeal carina shape: inverted "V" (left and right lateral propodeal carinae are adjacent medially at their intersection with antecostal sulcus of the first abdominal tergum); inverted "Y" (left and right lateral propodeal are adjacent medially posterior to antecostal sulcus of the first abdominal tergum, and connected to the antecostal sulcus by a median carina representing the median branch of the inverted "Y"). Anteromedian projection of the metanoto-propodeo-metapecto-mesopectal complex count: absent. S1 length vs. shortest width: S1 wider than long. Transverse carina on petiole shape: straight. Distal margin of male S9 shape: straight. Proximolateral corner of male S9 shape: blunt. Cupula length vs. gonostyle-volsella complex length: cupula less than 1/2 the length of gonostyle-volsella complex in lateral view. Proximodorsal notch of cupula count: present. Proximodorsal notch of cupula shape: arched. Proximolateral projection of the cupula

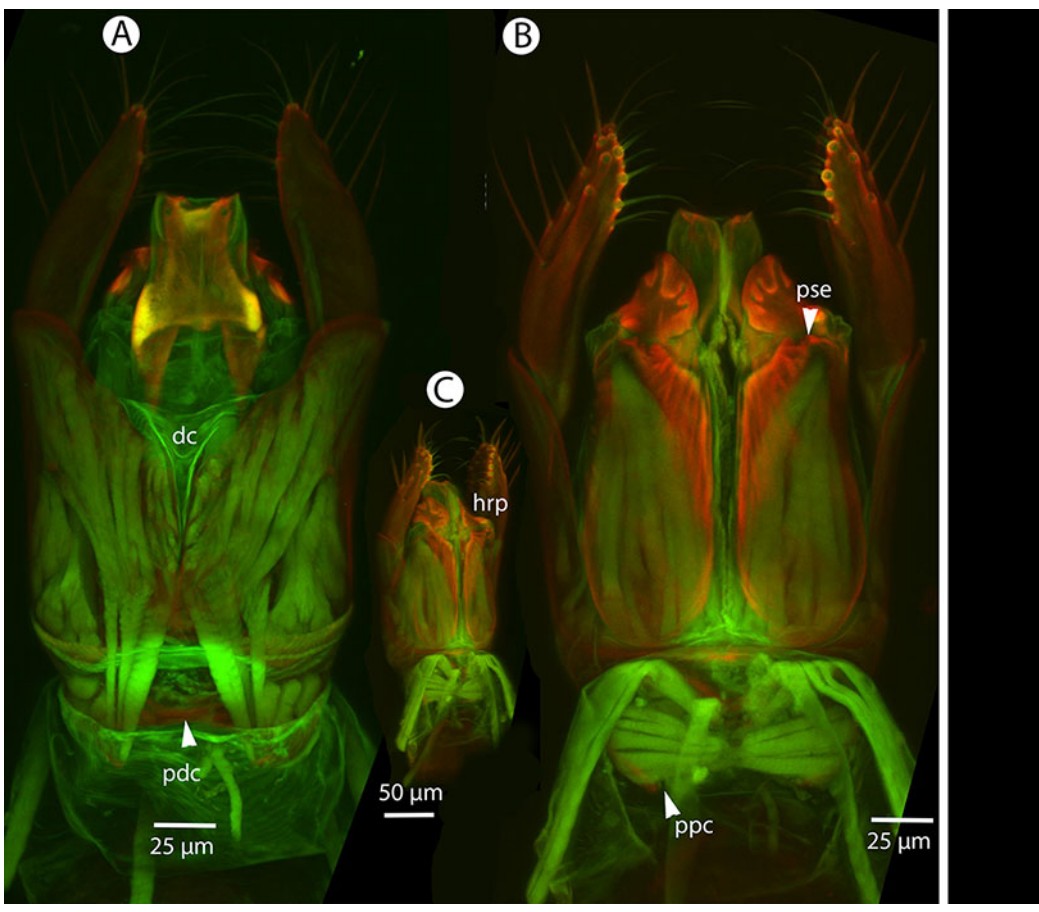

**Figure 43 CLSM volume rendered micrographs showing the male genitalia of *Conostigmus madagascariensis* Mikó and Trietsch sp. nov.** (A) Ventral view. (B) Dorsal view (dc, dorsomedian conjunctiva of gonostyle/volsella complex; hrp, harpe; pdc, proximodorsal notch of cupula; ppc, proximlateral projection of cupula; pss, parossicular seta).

shape: acute. Proximodorsal notch of cupula width vs. length: wider than long. Distodorsal margin of cupula shape: straight. Dorsomedian conjunctiva of the gonostyle-volsella complex length relative to length of gonostyle-volsella complex: dorsomedian conjunctiva extending 2/3 of length of gonostyle-volsella complex in dorsal view. Dorsomedian conjunctiva of the gonostyle-volsella complex count: present. Distal end of dorsomedian conjunctiva of the gonostyle-volsella complex shape: acute. Parossiculus count (parossiculus and gonostipes fusion): present (not fused with the gonostipes).
Apical parossicular seta number: one. Distal projection of the parossiculus count: absent. Distal projection of the penisvalva count: absent. Dorsal apodeme of penisvalva count: absent. Harpe length: harpe shorter than gonostipes in lateral view. Distodorsal setae of sensillar ring of harpe length vs. harpe width in lateral view: setae longer than harpe width. Distodorsal setae of sensillar ring of harpe orientation: medially. Sensillar ring area of harpe orientation: medially. Lateral setae of harpe count: present. Lateral setae of harpe orientation: oriented distally. Distal margin of harpe in lateral view: shape: blunt. Lateral margin of harpe shape: widest point of harpe is at its articulation site with gonostyle-volsella complex.

**Etymology:** The species epithet refers to Madagascar where *Conostigmus madagascariensis* is the most commonly collected among *Conostigmus* species.

**Comments:** The coloration of *Conostigmus madagascariensis* males is variable: specimens CASENT 2040905 and CASENT 2046020 have distally yellow hind coxa, and specimens CASENT 2040905 and CASENT 2022986 have yellow scapes. The coloration of *Conostigmus madagascariensis* females is also variable: F1–F8, cranium, mandible, metasoma, tegula brown, legs, scape, pedicel yellow in specimens CASENT 2053365, CASENT 2053573, CASENT 2053574; scape, F4–F8, cranium, mandible, metasoma, tegula brown, legs, except brown proximal region of hind coxa and distal region of hind femur, pedicel, F1–F4 yellow in specimens CASENT 2041648, CASENT 2044995.

Most specimens of *Conostigmus madagascariensis* lack the postocellar carina. In larger specimens, a very shallow sulcus connecting the posterior margins of the lateral ocelli present. In one specimen (CASENT 2044509) the postocellar carina is similar to *Conostigmus lucidus* sp. nov. Other charactersics of *Conostigmus lucidus* are absent from this specimen (*e.g.* petiole neck as long as wide, very weak microsculpture allover the body, sternaulus longer than half of mesopleuron length at the level of sternaulus, presence of straight lateral propodeal carinae).

**Material examined:** Holotype male: MADAGASCAR: Province Fianarantsoa, Parc National Ranomafana, radio tower at forest edge, Malaise mixed tropical forest, 12–19.2.2002, R. Harin'Hala, CASENT 2044913 (deposited in CAS).

Paratypes (44 males, 15 females): MADAGASCAR: 44 males, 15 females. CASENT 2000886, 2002178, 2002180, 2002187–2002191, 2004742, 2004744, 2004746–2004750, 2004753–2004754, 2009143–2009144, 2022986–2022987, 2040889–2040894, 2040896–2040899, 2040901, 2040905–2040908, 2041648, 2041940, 2041942, 2041945, 2044507, 2044509, 2044824, 2044895, 2044912, 2044995, 2045756, 2046020, 2053365, 2053393, 2053503, 2053573–2053574; IM 2289; PSUC_FEM 79702, 79759, 79761, 79763, PSUC_79714 (deposited in CAS, MRAC).

*Conostigmus missyhazenae* Mikó and Trietsch sp. nov.

Figures 44, 45, 46 and 47

**Diagnosis:** *Conostigmus missyhazenae* sp. nov. differs from other Malagasy *Conostigmus* species in the globular head (almost as long as wide in dorsal view and as high as long in lateral view) and the absence of the preoccipital sulcus.

**Description:** Body length: 1,750–2,000 μm. Color intensity pattern: NOT CODED. Color hue pattern: Cranium, mandible, mesosoma excluding front and proximal middle tibia, metasoma, antenna excluding distal scape and pedicel brown; distal scape, pedicel, protibia and proximal mesotibia ochre. Occipital carina sculpture: smooth. Median flange of occipital carina count: absent. Submedial flange of occipital carina count: absent. Dorsal margin of occipital carina vs. dorsal margin of lateral ocellus in lateral view: occipital carina is ventral to lateral ocellus in lateral view. Preoccipital lunula count: absent. Preoccipital carina count: absent. Preoccipital carina shape: NOT CODED. Preoccipital furrow count: absent. Preoccipital furrow anterior end: NOT CODED.

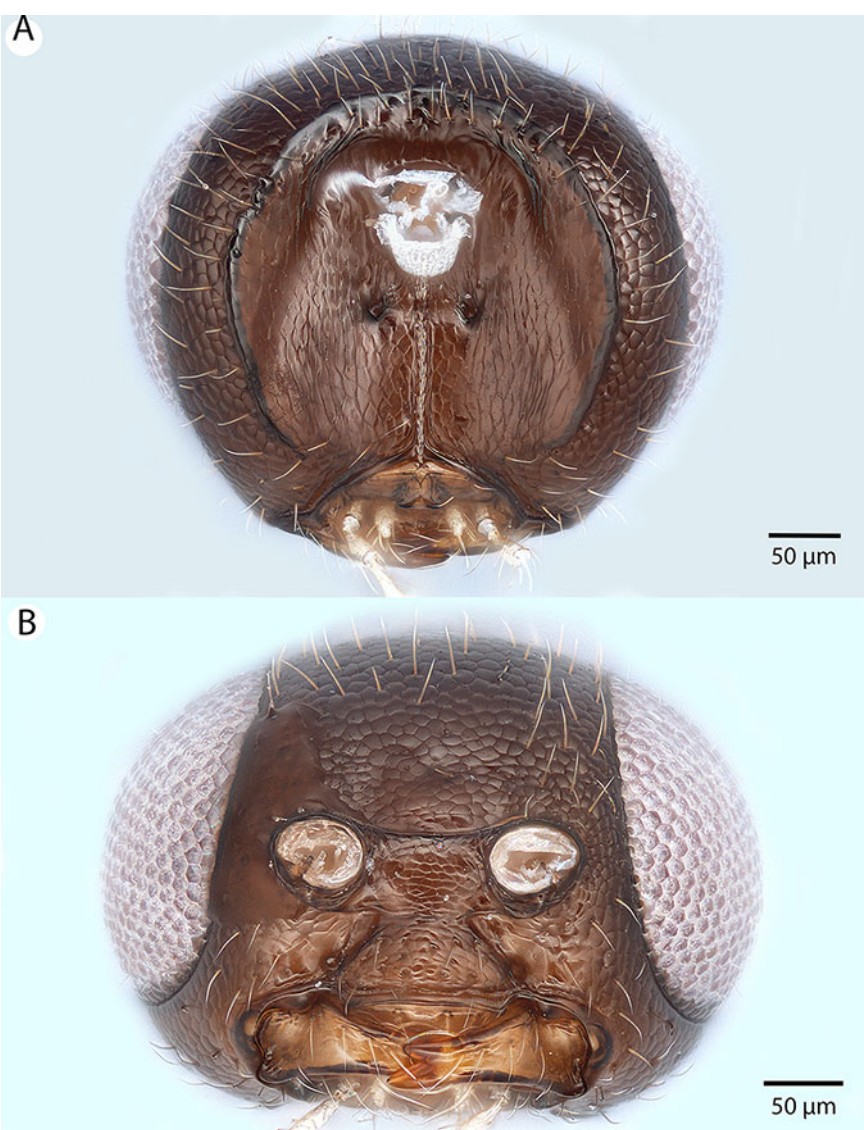

**Figure 44 Brightfield image showing the head of *Conostigmus missyhazenae* Mikó and Trietsch sp. nov.** (A) Posterior view. (B) Ventral view.

Postocellar carina count: absent. Male OOL: POL: LOL: 1.1–1.2:1.6–1.8:1. Female OOL: POL: LOL: 1.0–1.1:1.4:1.0. HW/IOS Male: 1.8–1.9. HW/IOS Female: 2.4. Setal pit on vertex size: smaller than diameter of scutes. Transverse frontal carina count: absent. Transverse scutes on frons count: absent. Rugose region on frons count: absent. Randomly sized areolae around setal pits on frons count: absent. Antennal scrobe count: absent. Ventromedian setiferous patch and ventrolateral setiferous patch count: absent. Facial pit count: facial pit present. Supraclypeal depression count: present. Supraclypeal depression structure: present medially, inverted U-shaped. Intertorular carina count: present. Intertorular area count: present. Median region of intertorular area shape: flat. Ventral margin of antennal rim vs. dorsal margin of clypeus: not adjacent. Torulo-clypeal carina count: present. Subtorular carina count: absent. Mandibular tooth count: 2. Female

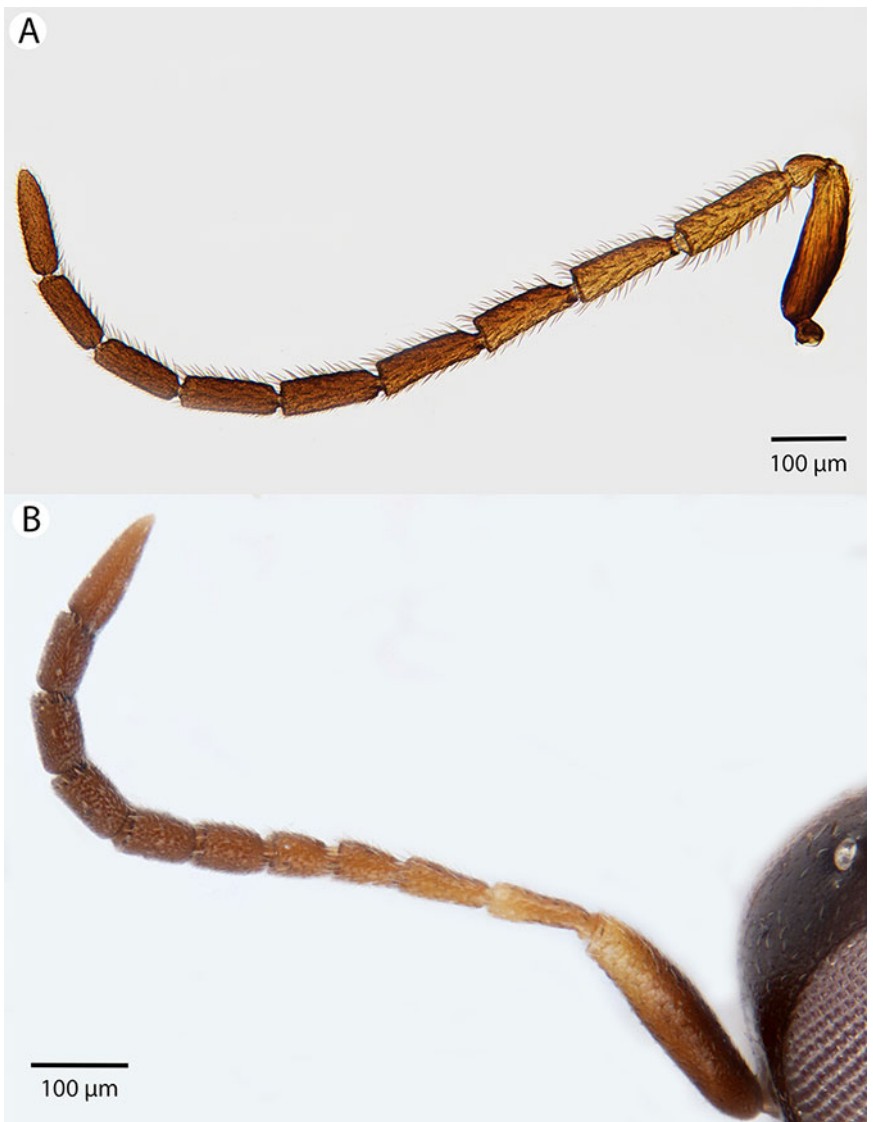

**Figure 45 Brightfield image showing the antenna of *Conostigmus missyhazenae* Mikó and Trietsch sp. nov.** (A) Male. (B) Female.

flagellomere one length vs. pedicel: 0.9–1.0. Female ninth flagellomere length: F9 less than F7 + F8. Sensillar patch of the male flagellomere pattern: F5–F9. Length of setae on male flagellomere vs. male flagellomere width: setae shorter than width of flagellomeres. Male flagellomere one length vs. male second flagellomere length: 1.1–1.2. Male flagellomere one length vs. pedicel length: 3.2–4.0. Ventrolateral invagination of the pronotum count: present. Scutes on posterior region of mesoscutum and dorsal region of mesoscutellum convexity: flat. Notaulus posterior end location: adjacent to transscutal articulation. Median mesoscutal sulcus posterior end: adjacent to transscutal articulation. Scutoscutellar sulcus vs. transscutal articulation: adjacent. Axillular carina count: absent. Axillular carina shape: NOT CODED. Epicnemium posterior margin shape: anterior discrimenal pit present; epicnemial carina curved. Epicnemial carina count:

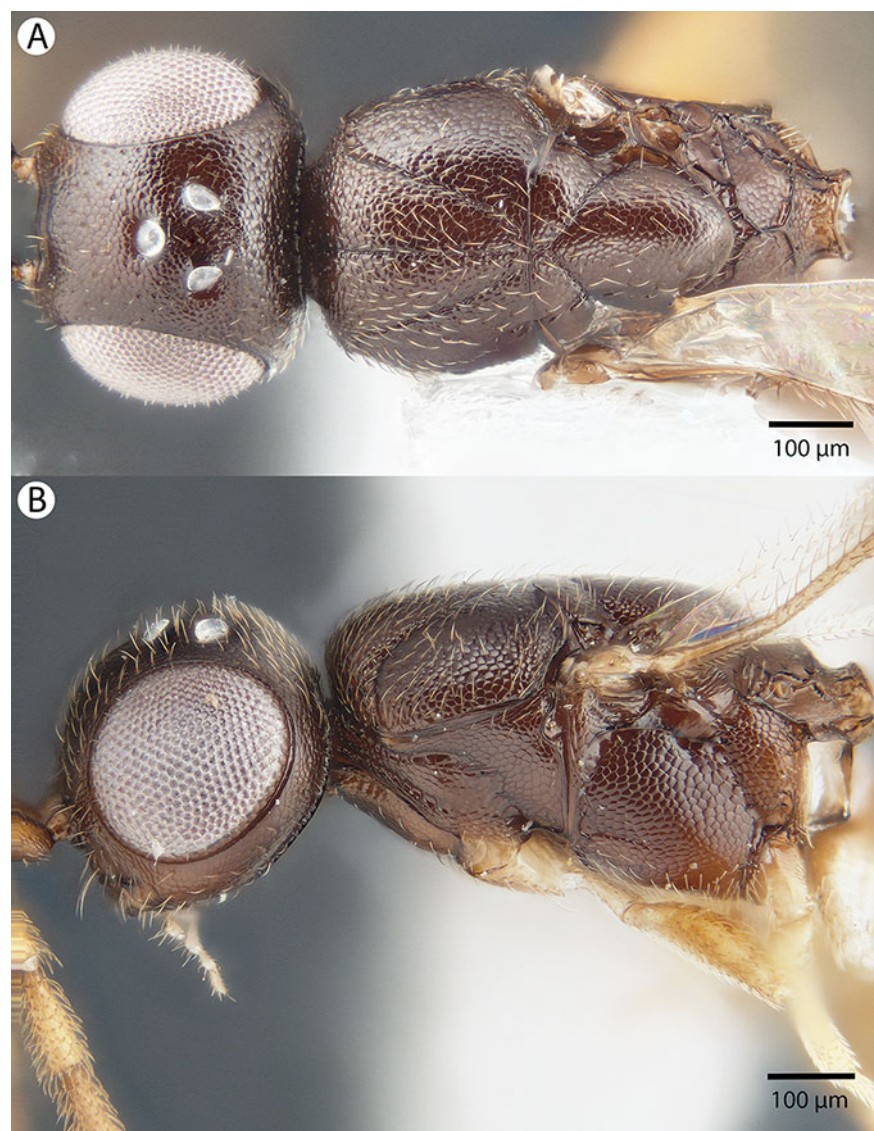

**Figure 46 Brightfield image showing the head and mesosoma of *Conostigmus missyhazenae* Mikó and Trietsch sp. nov.** (A) Dorsal view. (B) Lateral view.

complete. Sternaulus count: present. Sternaulus length: short, not reaching 1/2 of mesopleuron length at level of sternaulus. Speculum ventral limit: not extending ventrally of pleural pit line. Mesometapleural sulcus count: present. Metapleural carina count: present. Transverse line of the metanotum-propodeum vs. antecostal sulcus of the first abdominal tergum: adjacent sublaterally. Lateral propodeal carina count: present. Lateral propodeal carina shape: inverted "Y" (left and right lateral propodeal are adjacent medially posterior to antecostal sulcus of the first abdominal tergum, and connected to the antecostal sulcus by a median carina representing the median branch of the inverted "Y"). Anteromedian projection of the metanoto-propodeo-metapecto-mesopectal complex count: absent. S1 length vs. shortest width: S1 wider than long. Transverse carina on petiole shape: concave. Distal margin of male S9 shape: convex. Proximolateral corner

Mikó et al. (2016), *PeerJ*, DOI 10.7717/peerj.2682 58/87

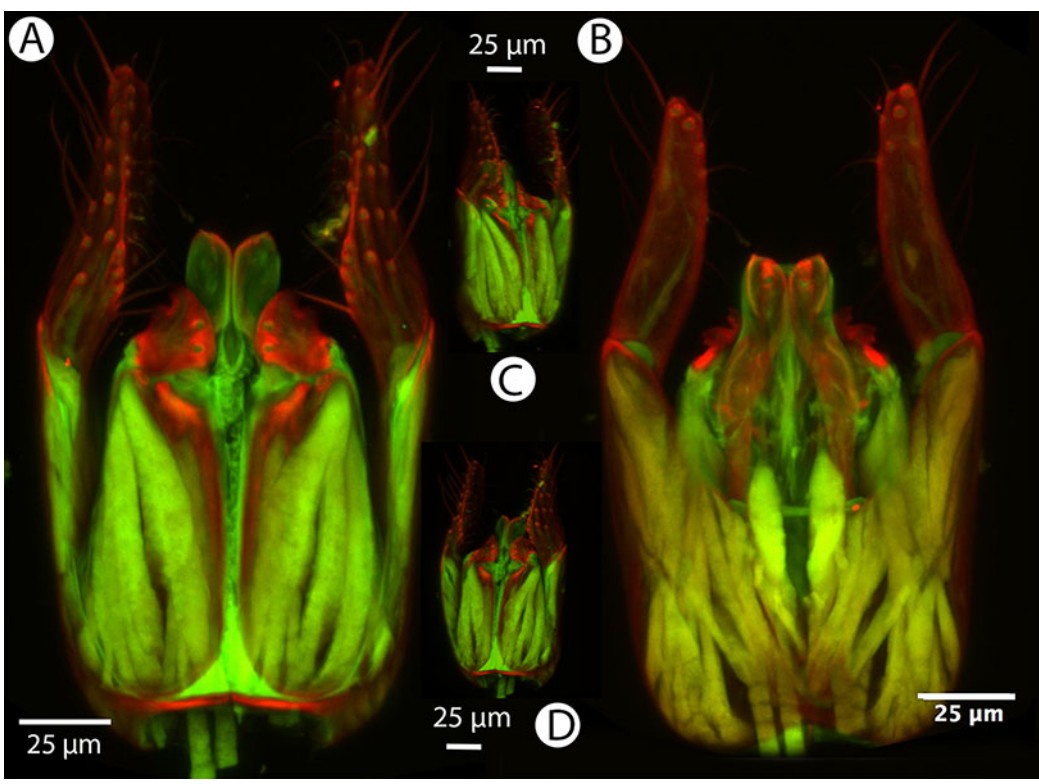

**Figure 47 CLSM volume rendered micrographs showing the male genitalia of *Conostigmus missyhazenae* Mikó and Trietsch sp. nov.** (A) Ventral view. (B) Dorsal view (hrp, harpe; dhs, dorsomedial setae of harpal setal ring).

of male S9 shape: blunt. Cupula length vs. gonostyle-volsella complex length: cupula less than 1/2 the length of gonostyle-volsella complex in lateral view. Proximodorsal notch of cupula count: present. Proximodorsal notch of cupula shape: arched. Proximolateral projection of the cupula shape: acute. Proximodorsal notch of cupula width vs. length: wider than long. Distodorsal margin of cupula shape: straight. Dorsomedian conjunctiva of the gonostyle-volsella complex length relative to length of gonostyle-volsella complex: dorsomedian conjunctiva extending 2/3 of length of gonostyle-volsella complex in dorsal view. Dorsomedian conjunctiva of the gonostyle-volsella complex count: present. Distal end of dorsomedian conjunctiva of the gonostyle-volsella complex shape: acute. Parossiculus count (parossiculus and gonostipes fusion): present (not fused with the gonostipes). Apical parossicular seta number: one. Distal projection of the parossiculus count: absent. Distal projection of the penisvalva count: absent. Dorsal apodeme of penisvalva count: absent. Harpe length: harpe shorter than gonostipes in lateral view. Distodorsal setae of sensillar ring of harpe length vs. harpe width in lateral view: setae as long or shorter than harpe width. Distodorsal setae of sensillar ring of harpe orientation: distomedially. Sensillar ring area of harpe orientation: medially. Lateral setae of harpe count: present. Lateral setae of harpe orientation: oriented distoventrally. Distal margin of harpe in lateral view: shape: blunt. Lateral margin of harpe shape: widest point of harpe is in its proximal 1/3rd.

**Etymology:** The species epithet honors Missy Hazen, research technologist at The Huck Institute of the Life Sciences, Pennsylvania State University, who facilitated the microscopy of these and other specimens.

**Material examined:** Holotype male: MADAGASCAR: Parc National Ranomafana, Belle Vue at Talatakely, Malaise, secondary tropical forest, 12–19.2.2002, R. Harin'Hala CASENT 2046019 (deposited in CAS). Paratypes (two males, two females): MADAGASCAR: two males, two females. CASENT 2002183, 2004752; PSUC_FEM 79731, 79747 (CAS).

*Conostigmus pseudobabaiax* Mikó and Trietsch sp. nov.

Figures 48, 49, 50, 51 and 52

**Diagnosis:** *Conostigmus pseudobabaiax* sp. nov. shares the presence of a prognathous head dorsal-most point of occipital carina is dorsal to posterior ocellus in lateral view) and the presence of transverse scutes on the ventral region of the frons with *C. babaiax* Dessart, 1997, *C. toliaraensis* sp. nov. and *Conostigmus longulus* Dessart, 1996. *Conostigmus pseudobabaix, C. babaiax,* and *C. toliaraensis* sp. nov. differ from other *Conostigmus* species by the presence of ventromedian and ventrolateral white, setiferous patches on the frons. *Conostigmus pseudobabaiax* and *C. toliaraensis* differ from *Conostigmus babaiax* in OOL longer than LOL (in *Conostigmus babaiax* OOL is shorter than LOL). *Conostigmus toliaraensis* can be readily differentiated from *C. pseudobabaiax* by the following phenotypes: first female flagellomere $0.9\times$ the length of pedicel ($1.4\times$ as long in *C. pseudobabaiax*); male flagellomere 1 $1.1\times$ as long as second male flagellomere ($1.3$–$1.4\times$ as long in *C. pseudobabaiax*); scutes are strongly convex (flat in *C. pseudobabaiax*); proximodorsal notch of cupula as long as wide and harpe as long as gonostyle/volsella complex in lateral view (proximodorsal notch of cupula almost $2\times$ as wide as long; harpe $0.7\times$ length of gonostyle/volsella complex in *C. pseudobabaiax*).

**Description:** Body length: 2,450–3,125 μm. Color intensity pattern: ventral region of cranium is lighter than dorsal region of cranium. Color hue pattern: Distal part of scape, pedicel, F1–F3 ochre; legs except proximal metacoxa yellow; rest of body brown. Occipital carina sculpture: crenulate. Median flange of occipital carina count: absent. Submedial flange of occipital carina count: absent. Dorsal margin of occipital carina vs. dorsal margin of lateral ocellus in lateral view: occipital carina is dorsal to lateral ocellus in lateral view. Preoccipital lunula count: NOT CODED. Preoccipital carina count: absent. Preoccipital carina shape: NOT CODED. Preoccipital furrow count: present. Preoccipital furrow anterior end: Preoccipital furrow ends inside ocellar triangle. Postocellar carina count: absent. Male OOL: POL: LOL: 1.2–1.3:1:1. Female OOL: POL: LOL: 1.4:1.0–1.2:1.0. HW/IOS Male: 2.0–2.2. HW/IOS Female: 2.3–2.6. Setal pit on vertex size: smaller than diameter of scutes. Transverse frontal carina count: absent. Transverse scutes on frons count: present. Rugose region on frons count: absent. Randomly sized areolae around setal pits on frons count: absent. Antennal scrobe count: absent. Ventromedian setiferous patch and ventrolateral setiferous patch count: present. Facial pit count: no external corresponding structure present. Supraclypeal depression count:

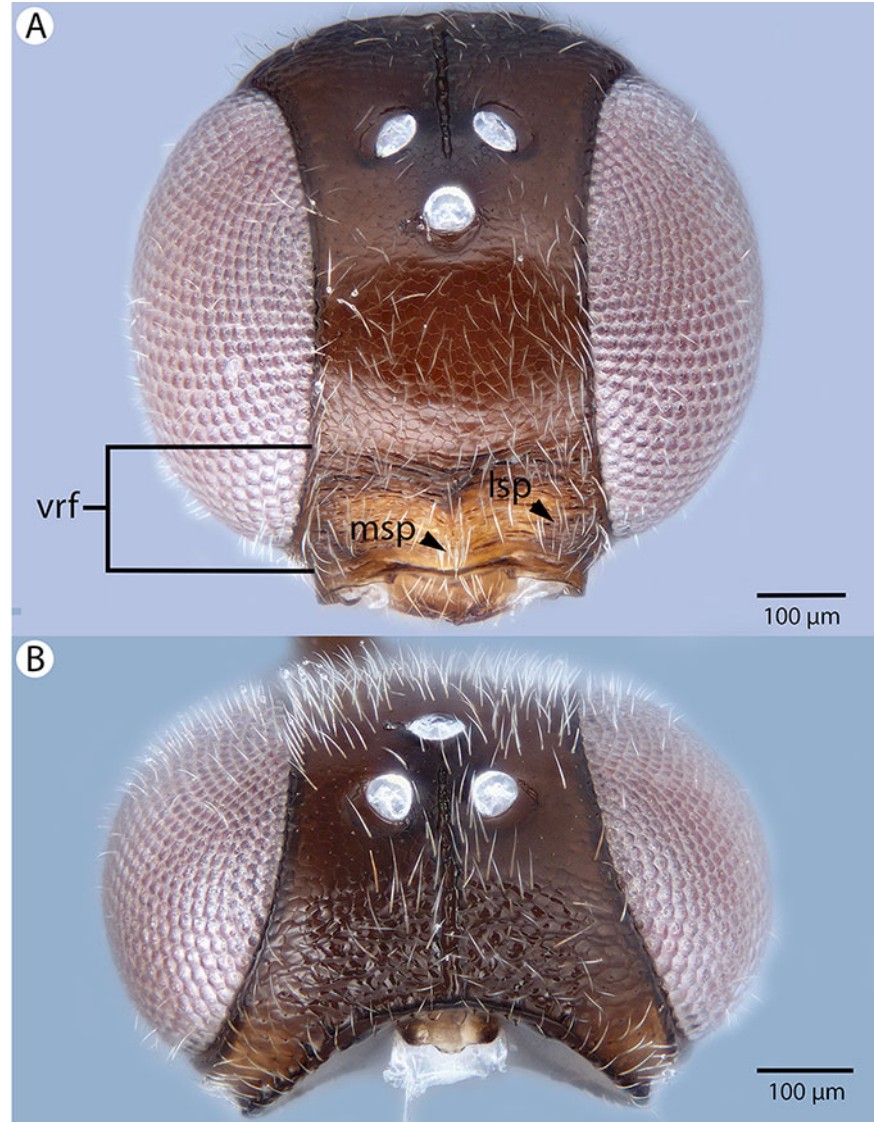

**Figure 48 Brightfield image showing the head of *Conostigmus pseudobabaiax* Mikó and Trietsch sp. nov.** (A) Anterior view. (B) Dorsal view.

absent. Supraclypeal depression structure: NOT CODED. Intertorular carina count: present. Intertorular area count: present. Median region of intertorular area shape: flat. Ventral margin of antennal rim vs. dorsal margin of clypeus: not adjacent. Torulo-clypeal carina count: present. Subtorular carina count: absent. Mandibular tooth count: 2. Female flagellomere one length vs. pedicel: 1.4. Female ninth flagellomere length: F9 less than F7 + F8. Sensillar patch of the male flagellomere pattern: F4–F9. Length of setae on male flagellomere vs. male flagellomere width: setae shorter than width of flagellomeres. Male flagellomere one length vs. male second flagellomere length: 1.3–1.4. Male flagellomere one length vs. pedicel length: 3.0–3.2. Ventrolateral invagination of the pronotum count: present. Scutes on posterior region of mesoscutum and dorsal region of mesoscutellum convexity: flat. Notaulus posterior end location: adjacent to transscutal

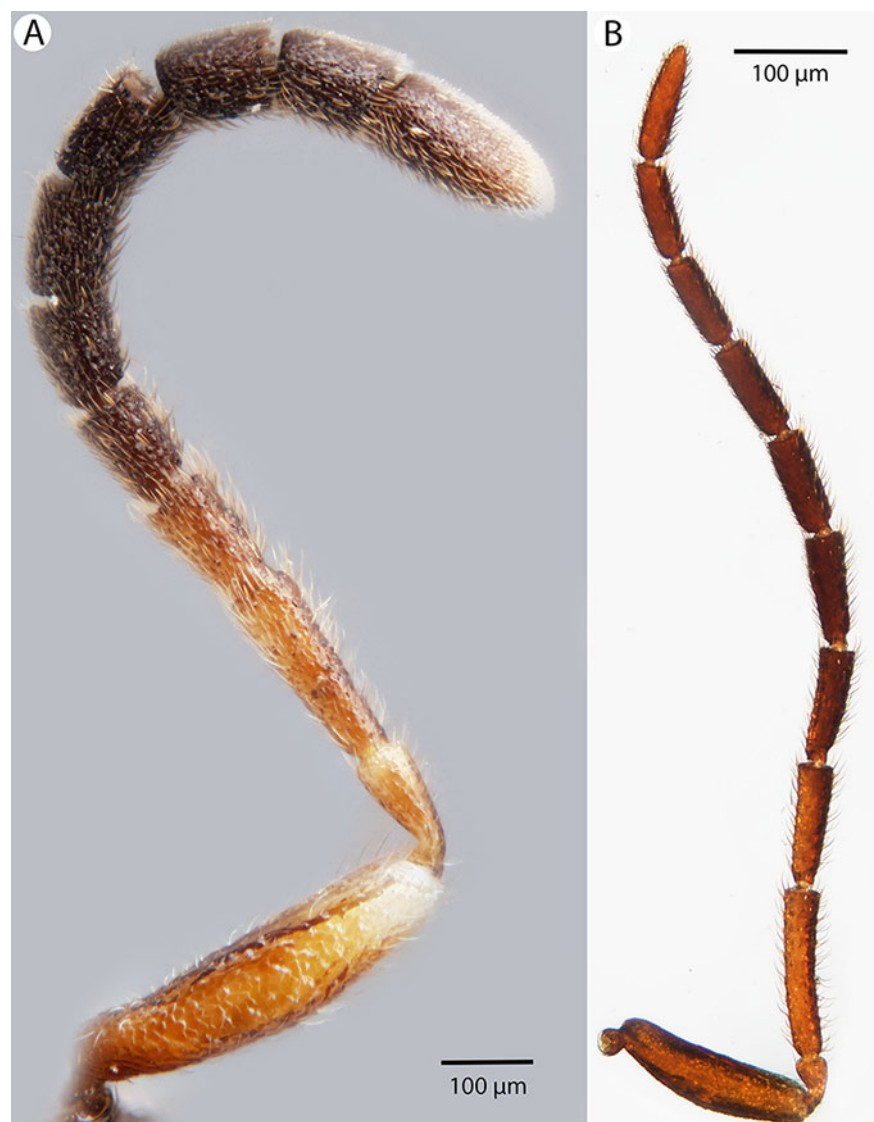

**Figure 49 Brightfield image showing the antenna of *Conostigmus pseudobabaiax* Mikó and Trietsch sp. nov.** (A) Female. (B) Male.

articulation. Median mesoscutal sulcus posterior end: not adjacent to transscutal articulation (ends anterior to transscutal articulation). Scutoscutellar sulcus vs. transscutal articulation: adjacent. Axillular carina count: absent. Axillular carina shape: NOT CODED. Epicnemium posterior margin shape: anterior discrimenal pit absent; epicnemial carina interrupted medially. Epicnemial carina count: present only laterally. Sternaulus count: present. Sternaulus length: short, not reaching 1/2 of mesopleuron length at level of sternaulus. Speculum ventral limit: not extending ventrally of pleural pit line. Mesometapleural sulcus count: present. Metapleural carina count: present. Transverse line of the metanotum-propodeum vs. antecostal sulcus of the first abdominal tergum: adjacent sublaterally. Lateral propodeal carina count: present. Lateral propodeal carina shape: straight (left and right lateral propodeal carinae compose a carina that is not broken

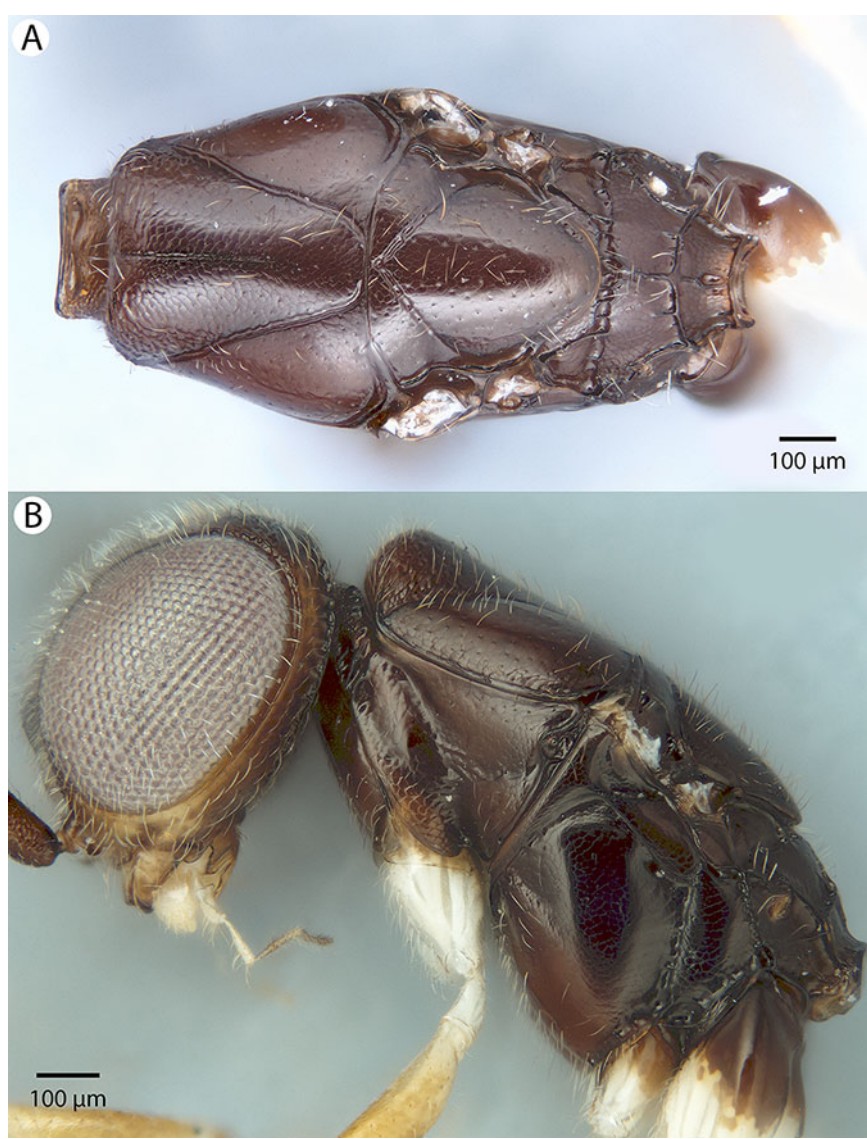

**Figure 50** Brightfield image showing the head and mesosoma of *Conostigmus pseudobabaiax* Mikó and Trietsch sp. nov. (A) Mesosoma, dorsal view. (B) Head and mesosoma, lateral view.

medially). Anteromedian projection of the metanoto-propodeo-metapecto-mesopectal complex count: absent. S1 length vs. shortest width: S1 wider than long. Transverse carina on petiole shape: concave. Distal margin of male S9 shape: convex. Proximolateral corner of male S9 shape: blunt. Cupula length vs. gonostyle-volsella complex length: cupula less than 1/2 the length of gonostyle-volsella complex in lateral view. Proximodorsal notch of cupula count: present. Proximodorsal notch of cupula shape: arched. Proximolateral projection of the cupula shape: blunt. Proximodorsal notch of cupula width vs. length: wider than long. Distodorsal margin of cupula shape: straight. Dorsomedian conjunctiva of the gonostyle-volsella complex length relative to length of gonostyle-volsella complex: dorsomedian conjunctiva extending 2/3 of length of gonostyle-volsella complex in dorsal view. Dorsomedian conjunctiva of the gonostyle-volsella complex count: present.

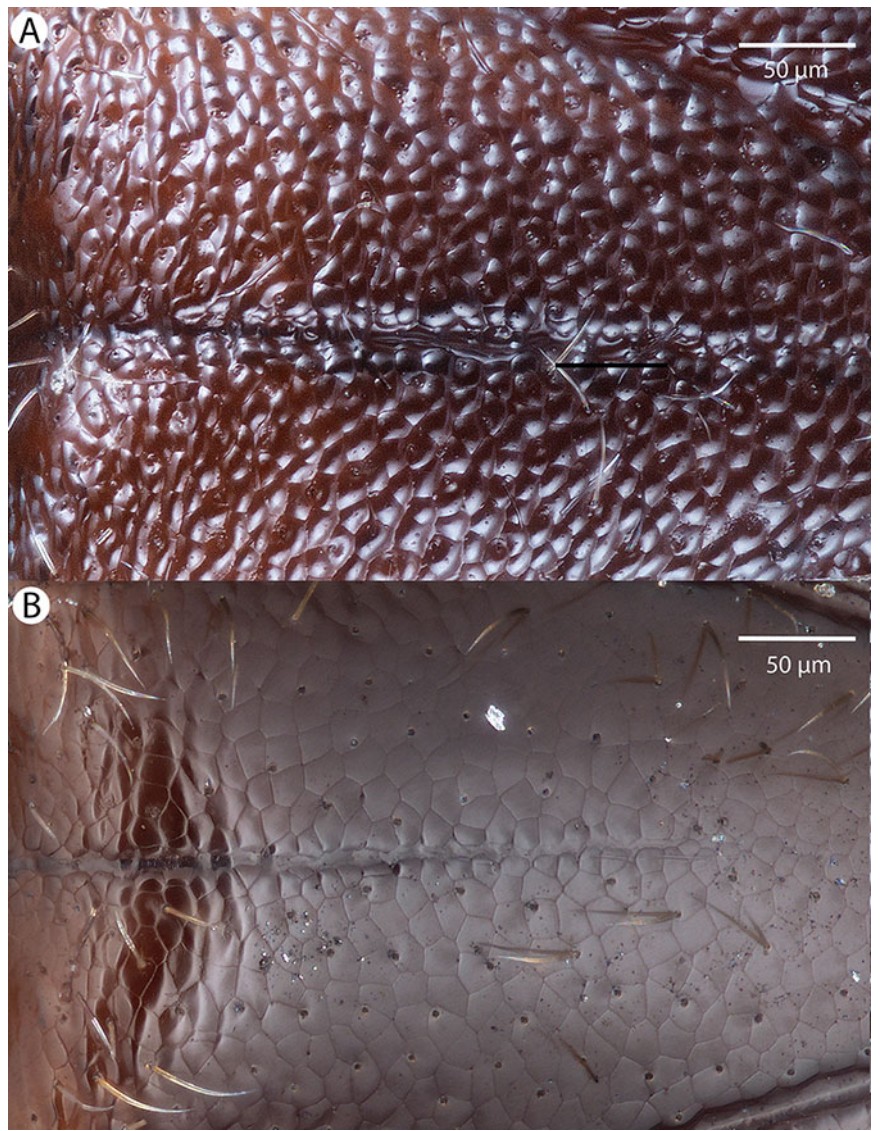

**Figure 51 Brightfield image showing the middle anteromesoscutum of *Conostigmus* species, dorsal view.** (A) *Conostigmus toliaraensis* Mikó and Trietsch sp. nov. (B) *Conostigmus pseudobabaiax* Mikó and Trietsch sp. nov.

Distal end of dorsomedian conjunctiva of the gonostyle-volsella complex shape: acute. Parossiculus count (parossiculus and gonostipes fusion): present (not fused with the gonostipes). Apical parossicular seta number: one. Distal projection of the parossiculus count: absent. Distal projection of the penisvalva count: absent. Dorsal apodeme of penisvalva count: absent. Harpe length: harpe shorter than gonostipes in lateral view. Distodorsal setae of sensillar ring of harpe length vs. harpe width in lateral view: setae as long or shorter than harpe width. Distodorsal setae of sensillar ring of harpe orientation: distomedially. Sensillar ring area of harpe orientation: medially. Lateral setae of harpe count: present. Lateral setae of harpe orientation: oriented distally. Distal margin of harpe in lateral view: shape: blunt. Lateral margin of harpe shape: widest point of harpe is at its articulation site with gonostyle-volsella complex.

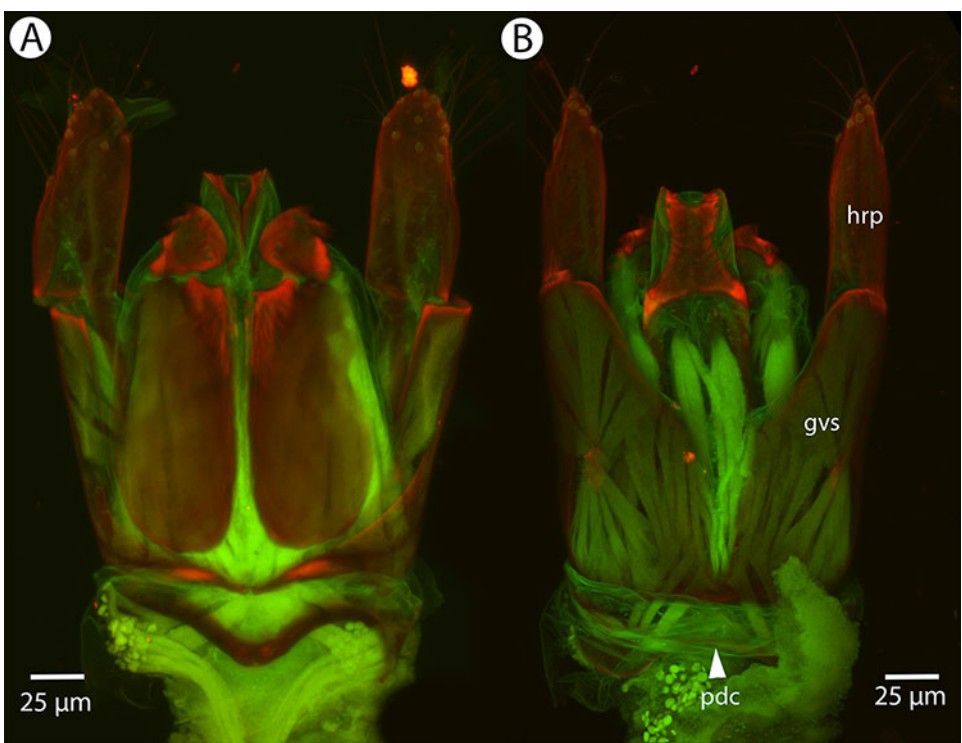

**Figure 52 CLSM volume rendered micrographs showing the male genitalia of *Conostigmus pseudobabaiax* Mikó and Trietsch sp. nov.** (A) Ventral view. (B) Dorsal view (dhs, dorsomedial setae of harpal setal ring; gvs, gonostyle/volsella complex; hrp, harpe; pdc, proximodorsal notch of cupula).

**Etymology:** From the Greek *pseudo-* (= false) and the specific name *babaiax*, indicating a close resemblance of *Conostigmus pseudobabaiax* and *C. babaiax*.

**Material examined:** Holotype male: MADAGASCAR: Ranomafana JIRAMA water works, Malaise trap near river, 16.10–8.11.2001, R. Harin'Hala, CASENT 2053690 (deposited in CAS). Paratypes (five males, six females): MADAGASCAR: five males, six females. CASENT 2006450–2006451, 2032774, 2041943, 2046097, 2046151, 2053381–2053382, 2053425, CASENT_2040937; PSUC_FEM 79736 (deposited in CAS, MRAC).

*Conostigmus toliaraensis* Mikó and Trietsch sp. nov.

Figures 53, 54, 55, 56 and 57

**Diagnosis:** *Conostigmus toliaraensis* sp. nov. shares the presence of a prognathous head (dorsal-most point of occipital carina is dorsal to posterior ocellus in lateral view) and the presence of transverse scutes on the ventral region of the frons with *C. babaiax* Dessart, 1996, *C. pseudobabaiax* sp. nov. and *Conostigmus longulus* Dessart, 1997. *Conostigmus toliaraensis*, *C. babaiax*, and *C. pseudobabaiax* sp. nov. differ from other *Conostigmus* species by the presence of ventromedian and ventrolateral white, setiferous patches on the frons. *Conostigmus pseudobabaiax* and *C. toliaraensis* differ from *Conostigmus babaiax* in OOL longer than LOL (in *Conostigmus babaiax* OOL is shorter than LOL). *Conostigmus toliaraensis* can be readily differentiated from *C. pseudobabaiax* by the following phenotypes: first female flagellomere 0.9× the length of pedicel (1.4× as long in

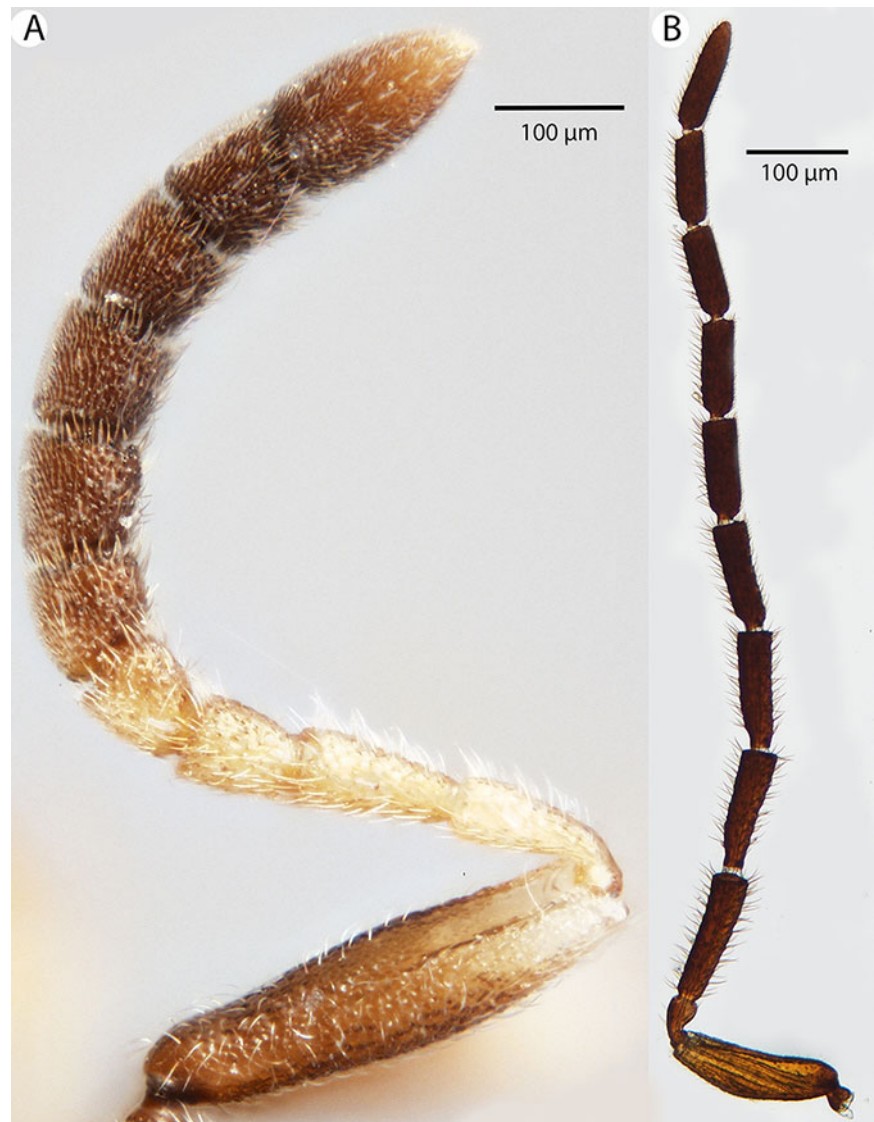

**Figure 53 Brightfield image showing the antenna of *Conostigmus toliaraensis* Mikó and Trietsch sp. nov.** (A) Female. (B) Male.

*C. pseudobabaiax*); male flagellomere 1 1.1× as long as second male flagellomere (1.3–1.4× as long in *C. pseudobabaiax*); scutes are strongly convex (flat in *C. pseudobabaiax*); proximodorsal notch of cupula as long as wide and harpe as long as gonostyle/volsella complex in lateral view (proximodorsal notch of cupula almost 2× as wide as long; harpe 0.7× length of gonostyle/volsella complex in *C. pseudobabaiax*).

**Description:** Body length: 2,000–3,450 μm. Color intensity pattern: ventral region of cranium is lighter than dorsal region of cranium. Color hue pattern: Distal part of scape, pedicel, F1–F3 ochre; legs except proximal metacoxa yellow; rest of body brown; Scape, hind leg except metacoxa ochre; fore and hind legs, distal metacoxa yellow; rest of body brown. Occipital carina sculpture: crenulate. Median flange of occipital carina count: absent. Submedial flange of occipital carina count: absent. Dorsal margin of

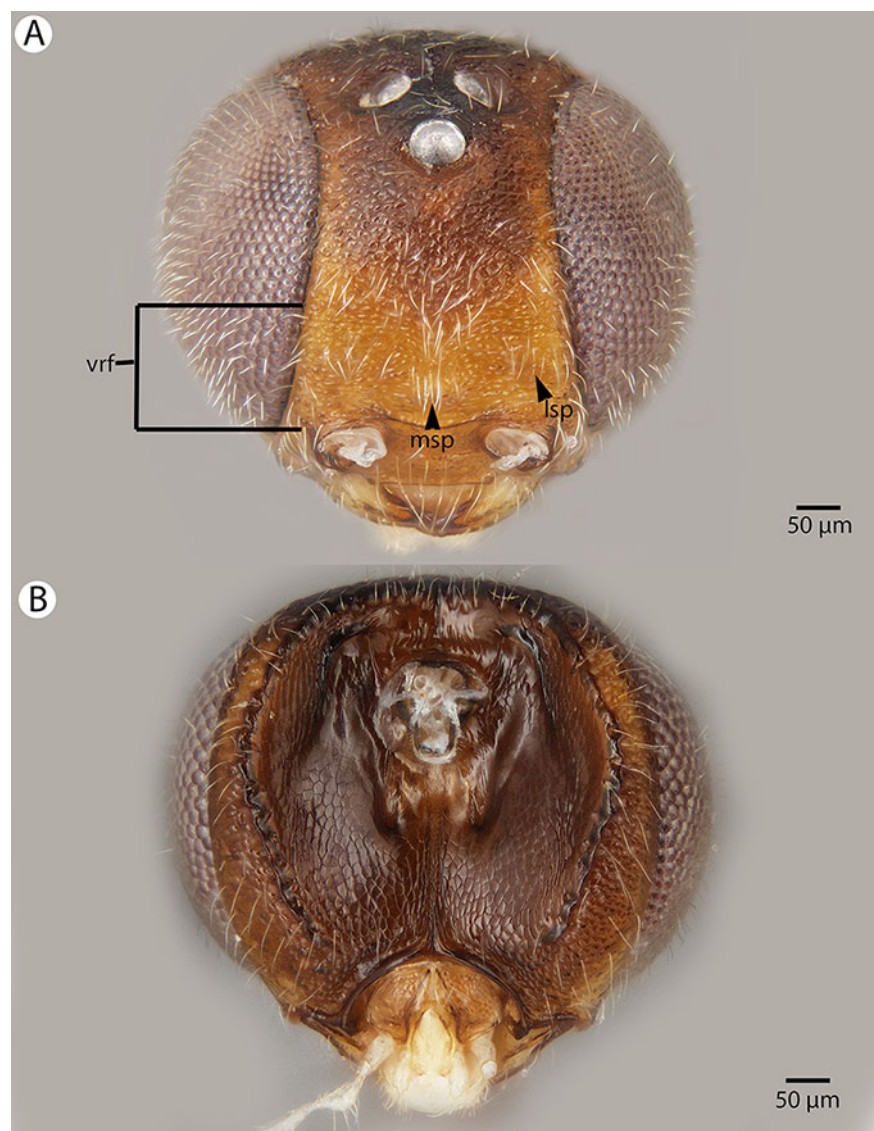

**Figure 54** Brightfield image showing the head of *Conostigmus toliaraensis* Mikó and Trietsch sp. nov. (A) Anterior view. (B) Posterior view.

occipital carina vs. dorsal margin of lateral ocellus in lateral view: occipital carina is dorsal to lateral ocellus in lateral view. Preoccipital lunula count: NOT CODED. Preoccipital carina count: absent. Preoccipital carina shape: NOT CODED. Preoccipital furrow count: present. Preoccipital furrow anterior end: Preoccipital furrow ends inside ocellar triangle. Postocellar carina count: absent. Male OOL: POL: LOL: 1.3–1.5:1:1. Female OOL: POL: LOL: 1.2–1.3:1.0:1.0. HW/IOS Male: 2.0–2.2. HW/IOS Female: 2.3–2.7. Setal pit on vertex size: smaller than diameter of scutes. Transverse frontal carina count: absent. Transverse scutes on frons count: present. Rugose region on frons count: absent. Randomly sized areolae around setal pits on frons count: absent. Antennal scrobe count: absent. Ventromedian setiferous patch and ventrolateral setiferous patch count: present. Facial pit count: no external corresponding structure present. Supraclypeal depression

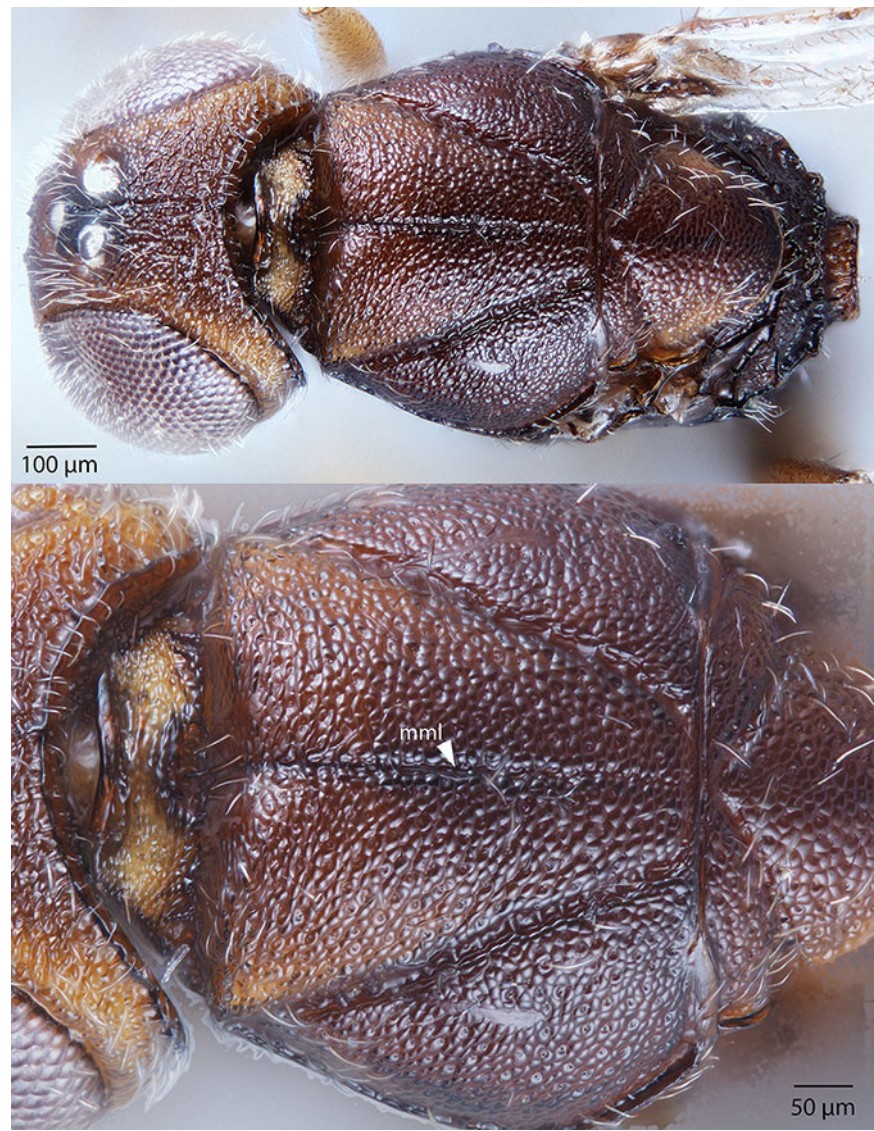

**Figure 55 Brightfield image showing the mesosoma of *Conostigmus toliaraensis* Mikó and Trietsch sp. nov.** Dorsal view (mml, median mesoscutal line).

count: absent. Supraclypeal depression structure: NOT CODED. Intertorular carina count: present. Intertorular area count: present. Median region of intertorular area shape: flat. Ventral margin of antennal rim vs. dorsal margin of clypeus: not adjacent. Torulo-clypeal carina count: absent. Subtorular carina count: absent. Mandibular tooth count: 2. Female flagellomere one length vs. pedicel: 0.9. Female ninth flagellomere length: F9 less than F7 + F8. Sensillar patch of the male flagellomere pattern: F4–F9; F5–F9. Length of setae on male flagellomere vs. male flagellomere width: setae shorter than width of flagellomeres. Male flagellomere one length vs. male second flagellomere length: 1.0–1.1; 1.1. Male flagellomere 1flagellomere one length vs. pedicel length: 2.5-3.0. Ventrolateral invagination of the pronotum count: present. Scutes on posterior region of mesoscutum and dorsal region of mesoscutellum convexity: convex. Notaulus posterior end

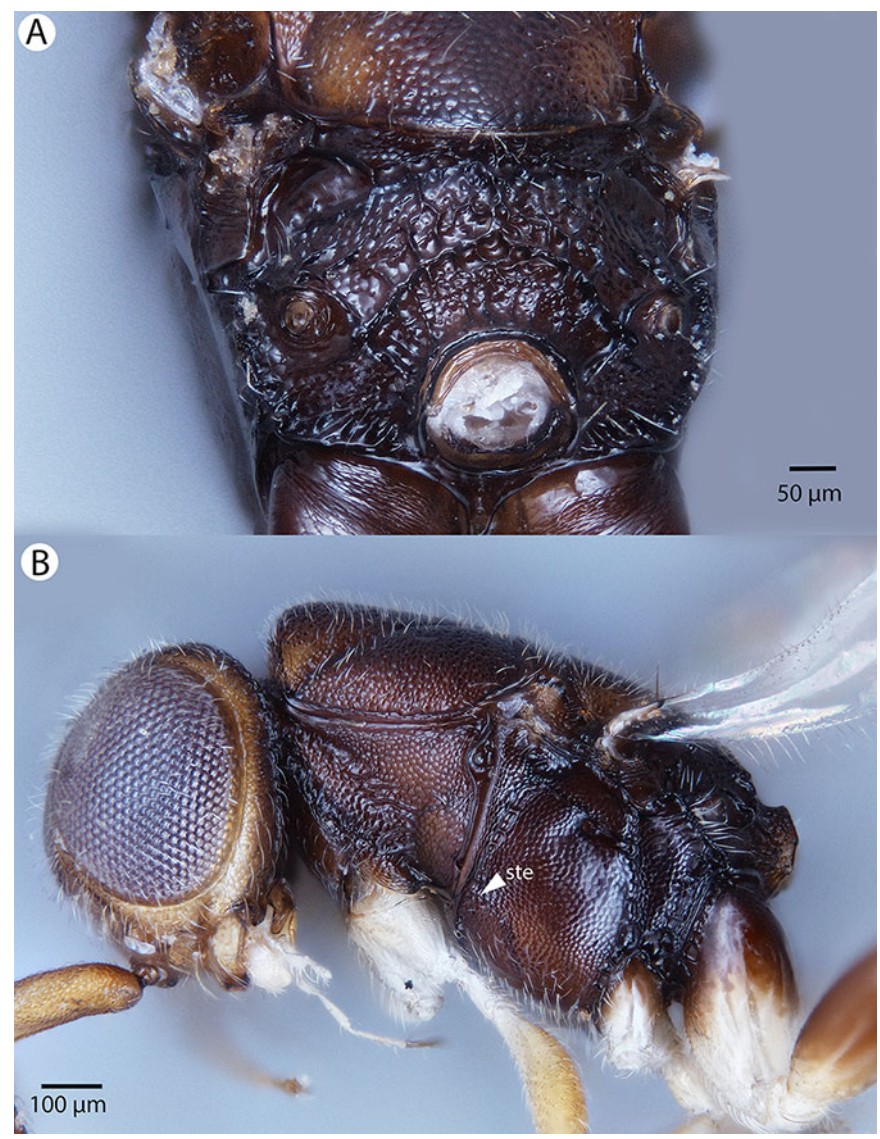

**Figure 56 Brightfield image showing the head and mesosoma of *Conostigmus toliaraensis* Mikó and Trietsch sp. nov.** (A) Mesosoma, posterior view. (B) Head and mesosoma, lateral view. (ste, sternaulus).

location: adjacent to transscutal articulation. Median mesoscutal sulcus posterior end: not adjacent to transscutal articulation (ends anterior to transscutal articulation). Scutoscutellar sulcus vs. transscutal articulation: adjacent. Axillular carina count: absent. Axillular carina shape: NOT CODED. Epicnemium posterior margin shape: anterior discrimenal pit absent; epicnemial carina interrupted medially. Epicnemial carina count: present only laterally. Sternaulus count: present. Sternaulus length: short, not reaching 1/2 of mesopleuron length at level of sternaulus. Speculum ventral limit: not extending ventrally of pleural pit line. Mesometapleural sulcus count: present. Metapleural carina count: present. Transverse line of the metanotum-propodeum vs. antecostal sulcus of the first abdominal tergum: adjacent sublaterally. Lateral propodeal carina count: present. Lateral propodeal carina shape: inverted "Y" (left and right lateral propodeal are adjacent

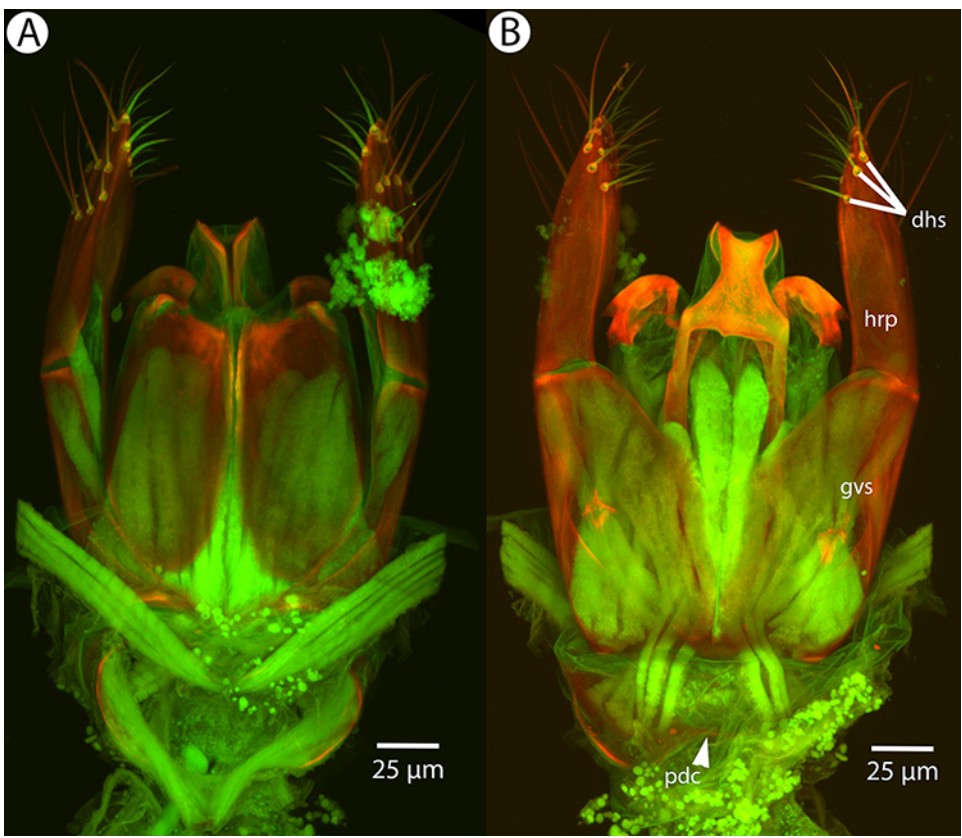

**Figure 57 CLSM volume rendered micrographs showing the male genitalia of *Conostigmus toliaraensis* Mikó and Trietsch sp. nov.** (A) Ventral view (B) Dorsal view (hrp, harpe; dhs, dorsomedial setae of harpal setal ring; gvs, gonostyle/volsella complex; pdc, proximodorsal notch of cupula).

medially posterior to antecostal sulcus of the first abdominal tergum, and connected to the antecostal sulcus by a median carina representing the median branch of the inverted "Y"); straight (left and right lateral propodeal carinae compose a carina that is not broken medially). Anteromedian projection of the metanoto-propodeo-metapecto-mesopectal complex count: absent. S1 length vs. shortest width: S1 wider than long. Transverse carina on petiole shape: concave. Distal margin of male S9 shape: convex. Proximolateral corner of male S9 shape: blunt. Cupula length vs. gonostyle-volsella complex length: cupula less than 1/2 the length of gonostyle-volsella complex in lateral view. Proximodorsal notch of cupula count: present. Proximodorsal notch of cupula shape: arched. Proximolateral projection of the cupula shape: blunt. Proximodorsal notch of cupula width vs. length: as long as wide. Distodorsal margin of cupula shape: straight. Dorsomedian conjunctiva of the gonostyle-volsella complex length relative to length of gonostyle-volsella complex: dorsomedian conjunctiva extending 2/3 of length of gonostyle-volsella complex in dorsal view. Dorsomedian conjunctiva of the gonostyle-volsella complex count: present. Distal end of dorsomedian conjunctiva of the gonostyle-volsella complex shape: acute. Parossiculus count (parossiculus and gonostipes fusion): present (not fused with the gonostipes). Apical parossicular seta number: one. Distal projection of the parossiculus count: absent. Distal projection of the penisvalva count: absent. Dorsal apodeme of penisvalva count: absent.

Harpe length: harpe as long as gonostipes in lateral view. Distodorsal setae of sensillar ring of harpe length vs. harpe width in lateral view: setae as long or shorter than harpe width. Distodorsal setae of sensillar ring of harpe orientation: distomedially. Sensillar ring area of harpe orientation: medially. Lateral setae of harpe count: present. Lateral setae of harpe orientation: oriented distally. Distal margin of harpe in lateral view: shape: blunt. Lateral margin of harpe shape: widest point of harpe is at its articulation site with gonostyle-volsella complex.

**Comments:** The length of the preoccipital furrow is variable in *Conostigmus toliaraensis Dessart, 1997*, from reaching the median ocellus (CAS2053309) to barely exceeding POL (CAS2040934). Two specimens from Foret Classee have narrower heads and bodies (distinct in HW/IOS ratio). Since the rest of the specimens are from Forret d'Ankazotsihitafototra, these two specimens might represent a different subspecies or species. The fact that there are only a few minute differences in the male genitalia morphology between *Conostigmus toliaraensis* and *C. pseudobabaiax* is unique, since male genitalia characters are traditionally used for species separation in Megaspilidae and in some cases provide the only diagnostic tool.

**Etymology:** The species epithet refers to the Toliara Province of Madagascar, where most of the specimens of this species were collected.

**Material examined:** Holotype male: CASENT 2053309 MADAGASCAR: Toliara Prov: Res. Speciale d'Ambohijanahary: Foret d'Ankazotsihitafototra: 35.2 km; NW Ambaravaranala; 1,050 m; 18°16′00″S, 45°24′24″E; 13–17.i.2003; MT; MISC BLF7019; Fisher, Griswold et al. California Academy of Sciences. Yellow pan trap- in montane rainforest (deposited in CAS).

Paratypes (nine females): MADAGASCAR: nine females. CASENT 2009754, 2040934-2040936, 2040983, 2041206, 2053310-2053311, 2053452 (CAS). (deposited in CAS MRAC).

## Identification key for Malagasy *Conostigmus* Dahlbom

1. **a.** Antennomeres gradually widening apically (Figs. 6B, 8B and 16A, females). . . . . **2**
   **aa.** Antennomeres not widening apically (Fig. 16B, males) . . . . . . . . . . . . . . . . . **12**
2. **a.** Scutes on ventral region of frons transverse (vrf: Figs. 1A, 11B, 22B and 48A). . . **3**
   **aa.** Scutes on ventral region of frons not transverse (Figs. 7A, 13A and 18A) . . . . . **6**
3. **a.** White setal patches on frons absent (Figs. 22A and 22B)
   **b.** Depression surrounding frontal pit present (dep: Figs. 22A and 22B)
   **c.** Transverse frontal carina present (tfc: Figs. 22A and 22B) . . . . . . . . . *Conostigmus longulus Dessart, 1997*
   **aa.** White setal patches on frons present (msp, ssp: Figs. 4A and 48A)
   **bb.** Depression surrounding frontal pit absent (Figs. 4A and 48A)
   **cc.** Transverse frontal carina absent (Figs. 4A and 48A) . . . . . . . . . . . . . . . . . . . . **4**
4. **aa.** LOL longer than OOL (Fig. 6A)
   *Conostigmus babaiax Dessart, 1997*
   **bb.** OOL shorter than LOL (Fig. 48) . . . . . . . . . . . . . . . . . . . . . . . . . . . . . . . . **5**

5. **a.** Flagellomere one length 0.9× pedicel length (Fig. 53A)
   **b.** Scutes on frons and mesonotum strongly convex (Figs. 55A, 55B and 51A)
   ***Conostigmus toliaraaensis*** Mikó and Trietsch sp. nov.
   **aa.** Flagellomere 1 1.4× as long as pedicel (Fig. 49A)
   **bb.** Scutes on frons and mesonotum flat (Figs. 50A, 50B and 51B)
   ***Conostigmus pseudobabaiax*** Mikó and Trietsch sp. nov.

6. **a.** Flagellomere nine as long as sum of lengths of flagellomeres six to eight (Fig. 16A)
   **b.** Rugous region on frons present (Figs. 13A and 13B)
   **c.** Subtorular carina present (stc: Fig. 13C)
   **d.** Median and submedial flanges of occipital carina present (mfc, sfc: Figs. 14A and 15B)
   **e.** Axillular carina present, carinae not adjacent posteriorly (not composing a U-shaped carina surrounding disc of mesoscutellum) (axc: Figs. 14B, 15A and 15B). . . . . . . . . . . . . . . . . . . . ***Conostigmus clavatus*** Mikó and Trietsch sp. nov.
   **aa.** Flagellomere nine shorter than sum of lengths of flagellomere seven and flagellomere eight (Figs. 6B and 8B)
   **bb.** Rugous region on frons absent (Figs. 3A, 7A and 11B)
   **cc.** Subtorular carina absent (Figs. 18A, 18B, 37A and 37B)
   **dd.** Median and submedial flanges of occipital carina absent (Figs. 7B and 8A)
   **ee.** Axillular carina absent (Figs. 5A, 10A and 19B) or axillular carinae continuous posteriorly (forming a U-shaped carina surrounding disc of mesoscutellum) (usc: Figs. 8A, 9A and 9B) . . . . . . . . . . . . . . . . . . . . . . . . . . . . . . . . . . . . . . **7**

7. **a.** Preoccipital carina present (poc: Figs. 7B, 14A and 15B)
   **b.** Anteromedian projection of the metanoto-propodeo-metapecto-mesopectal complex present (app: Fig. 8A)
   **c.** Randomly sized areolae around setal pits on frons present (aro: Fig. 7A)
   **d.** Axillular carinae continuous posteriorly (forming a U-shaped carina surrounding disc of mesoscutellum) (usc: Figs. 8A, 9A and 9B) . . . . . . ***Conostigmus ballescoracas*** Dessart, 1997
   **aa.** Preoccipital carina absent (Figs. 5A, 10A, 25A and 25B)
   **bb.** Anteromedian projection of the metanoto-propodeo-metapecto-mesopectal complex absent (Figs. 5A, 10A, 25A and 25B)
   **cc.** Randomly sized areolae around setal pits on frons absent (Figs. 4A, 11B and 18A)
   **dd.** Axillular carinae absent (Figs. 5A, 10A and 19B) . . . . . . . . . . . . . . . . . . . . . . **8**

8. **a.** Anterior neck of T1 (and corresponding S1) as long as wide (Fig. 28B)
   **b.** Median mesoscutal line marked by a row of punctures (mml: Fig. 28B)
   **c.** Sternaulus elongate (exceeding 3/4 of mesopleuron length measured at level of sternaulus)(ste: Figs. 29A and 29B)
   **d.** Postocellar carina present (pcc: Figs. 28B and 29B) . . . . . . . . ***Conostigmus lucidus*** Mikó and Trietsch sp. nov.
   **aa.** Anterior neck of T1 (and corresponding S1) much wider than long (Figs. 9B, 11A and 25B)

**bb.** Median mesoscutal line marked by a groove (mml: Figs. 5A, 5B, 9A, 9B, 15A and 15B)

**cc.** Sternaulus short, not reaching 1/2 of mesopleuron length measured at level of sternaulus (ste: Figs. 6B, 19A and 24B)

**dd.** Postocellar carina absent (Figs. 5A, 10A, 25A, 42A and 42B) . . . . . . . . . . . . . . **9**

9. **a.** Antennal scrobe present (asr: Figs. 10B and 11B)

    **b.** Depressions around setal pits on dorsal region of cranium and mesonotum larger than scutes (Fig. 10A) . . . . . . . .***Conostigmus bucephalus*** Mikó and Trietsch sp. nov.

    **aa.** Antennal scrobe absent (Figs. 4A, 7A, 13A and 13B)

    **bb.** Depressions around setal pits on cranium and mesonotum smaller than scutes. . . . . . . . . . . . . . . . . . . . . . . . . . . . . . . . . . . . . . . . . . . . . . . . .**10**

10. **a.** Head globular, almost as long as wide in dorsal view and as high as long in lateral view (Figs. 46A and 46B)

    **b.** OOL almost as long as LOL (Fig. 46A)

    **c.** Preoccipital sulcus absent (Fig. 46A) . . . . . . . .***Conostigmus missyhazenae*** Mikó and Trietsch sp. nov.

    **aa.** Head transverse, distinctly wider than long in dorsal view

    **bb.** OOL about 2× as long as LOL (Figs. 41A and 41B)

    **cc.** Preoccipital sulcus present (pos: Figs. 41A, 41B, 39A and 39B) . . . . . . . . . . **11**

11. **a.** Mandible with one tooth (Figs. 18A and 18B) ***Conostigmus fianarantsoaensis*** Mikó and Trietsch sp. nov.

    **aa.** Mandible with two teeth (Figs. 37A and 37B) ***Conostigmus madagascariensis*** Mikó and Trietsch sp. nov.

12. **a.** Scutes on ventral region of frons transverse (vrf: Figs. 1A, 11B, 22B and 48A). . . . . . . . . . . . . . . . . . . . . . . . . . . . . . . . . . . . . . . . . . . . . . . . . . . . . **13**

    **aa.** Scutes on ventral region of frons not transverse (Figs. 7A, 13A and 18A). . . . **15**

13. **a.** White setal patches on frons absent (Figs. 22A and 22B)

    **b.** Depression surrounding frontal pit present (dep: Figs. 22A and 22B)

    **c.** Transverse frontal carina present (tfc: Figs. 22A and 22B)

    **d.** Setal ring area of the harpe oriented dorsomedially (hrp: Figs. 26A and 26B)

    **e.** Dorsomedial setae of harpal setal ring elongate, apical ends adjacent medially, 2–3× as long as harpe width in lateral view (dhs: Fig. 26B) . . . . . . . . ***Conostigmus longulus*** *Dessart, 1997*

    **aa.** White setal patches on frons present (msp, ssp: Figs. 4A and 48A)

    **bb.** Depression surrounding frontal pit absent (Figs. 4A and 48A)

    **cc.** Transverse frontal carina absent (Figs. 4A and 48A)

    **dd.** Setal ring area of the harpe oriented medially (hrp: Fig. 57B)

    **ee.** Dorsomedial setae of harpal setal ring short, apical ends not adjacent medially, shorter than harpe width in lateral view (dhs: Fig. 57B) . . . . . . . . . . . . . . . . . . **14**

14. **a.** Flagellomere 1 1.1× as long as second flagellomere (Fig. 53B)

    **b.** Scutes on frons and mesonotum strongly convex (Fig. 51A)

    **c.** Proximodorsal notch of cupula as long as wide (pdc: Fig. 57B)

**d.** Harpe as long as gonostyle-volsella complex in lateral view (hrp, gvs: Fig. 57B) *Conostigmus toliaraensis* Mikó and Trietsch sp. nov.

**aa.** Flagellomere 1 1.3–1.4× as long as second flagellomere (Fig. 49B)

**bb.** Scutes on frons and mesonotum flat (Fig. 51B)

**cc.** Proximodorsal notch of cupula almost 2× as wide as long (pdc: Fig. 51B)

**ee.** Harpe 0.7 length of gonostyle/volsella complex in lateral view (hrp, gvs: Fig. 51B) . . . . . . *Conostigmus pseudobabaiax* Mikó and Trietsch sp. nov.

15. **a.** Subtorular carina present (stc: Fig. 13C)

    **b.** Axillular carina present (axc: Figs. 14B, 15A and 15B)

    **c.** Median and submedial flanges of occipital carina present (mfc, sfc: Figs. 14A and 15B)

    **d.** Rugulose sculpture on frons present (Figs. 13A and 13B)

    **e.** OOL/LOL: > :3.3 (Fig. 13B)

    **f.** Interorbital space wide (HW/IOS = 1.63–1.66)

    **g.** Distodorsal margin of cupula concave medially (ddm: Fig. 17B)

    **h.** Eyes bulging (Figs. 13, 14A and 15B); (Fig. 13B) *Conostigmus clavatus* Mikó and Trietsch sp. nov.

    **aa.** Subtorular carina absent (stc: Fig. 13C)

    **bb.** Axillular carina absent (axc: Figs. 14B, 15A and 15B)

    **cc.** Median and submedial flanges of occipital carina absent (mfc, sfc: Figs. 14A and 15B)

    **dd.** Rugulose sculpture on frons absent (Figs. 13A and 13B)

    **ee.** OOL/LOL < 3.3 (Fig. 13B)

    **ff.** Interorbital space narrow (HW/IOS > 1.8)

    **gg.** Distodorsal margin of cupula straight medially

    **hh.** Eyes not bulging (Figs. 13, 14A and 15B) . . . . . . . . . . . . . . . . . . . . . . . . . **16**

16. **a.** Anterior neck of T1 (and corresponding S1) as long as wide (Fig. 28B)

    **b.** Median mesoscutal line marked by a row of punctures (mml: Fig. 28B)

    **c.** Sternaulus elongate (exceeding 3/4 of mesopleuron length measured at level of sternaulus; ste: Figs. 29A and 29B)

    **d.** Postocellar carina present (pcc: Figs. 28B and 29B)

    **e.** Dorsomedian conjunctiva of the gonostyle/volsella complex absent (Fig. 30C)

    **f.** Proximodorsal notch of cupula absent (Fig. 30C)

    **g.** Parossiculus absent (parossiculus and gonostyle fused, Fig. 30C). *Conostigmus lucidus* Mikó and Trietsch sp. nov.

    **aa.** Anterior neck of T1 (and corresponding S1) much wider than long (Figs. 9B, 11A and 25B)

    **bb.** Median mesoscutal line marked by a groove (mml: Figs. 5A, 5B, 9A, 5B, 15A and 15B)

    **cc.** Sternaulus short, not reaching 1/2 of mesopleuron length measured at level of sternaulus (ste: Figs. 6B, 19A and 24B)

    **dd.** Postocellar carina absent (Figs. 5A, 10A, 25A, 42A and 42B)

**ee.** Dorsomedian conjunctiva of the gonostyle/volsella complex present
(dc: Figs. 21C and 43A)

**ff.** Proximodorsal notch of cupula present (pdc: Fig. 21B)

**gg.** Parossiculus present (parossiculus and gonostyle not fused, Fig. 1A). . . . . . . **17**

17. **a.** Preoccipital carina present (poc: Fig. 7B)

**b.** Anteromedian projection of the metanoto-propodeo-metapecto-mesopectal
complex present (app: Fig. 8A)

**c.** Randomly sized areolae around setal pits on frons present (aro: Fig. 7A)

**d.** Axillular carinae continuous posteriorly (forming a U-shaped carina
surrounding posteriorly and laterally mesoscutellar disc) (usc: Figs. 8A,
9A and 9B) . . . . . . . . . . . . . . . . . . . . . *Conostigmus ballescoracas Dessart, 1997*

**aa.** Preoccipital carina absent (Figs. 5A, 10A, 25A and 25B)

**bb.** Anteromedian projection of the metanoto-propodeo-metapecto-mesopectal
complex absent (Figs. 5A, 10A, 25A and 25B)

**cc.** Randomly sized areolae around setal pits on frons absent (Figs. 4A, 11B and 18A)

**dd.** Axillular carinae absent (Figs. 5A, 10A and 19B) . . . . . . . . . . . . . . . . . . . **18**

18. **a.** Head globular, almost as long as wide in dorsal view and as high as long in lateral
view (Figs. 46A and 46B)

**b.** OOL almost as long as LOL (Fig. 46A)

**c.** Preoccipital sulcus absent (Fig. 46A)

**d.** Proximal region of lateral margins of harpe diverging distally and widest
point of harpe is in its proximal 1/3rd (hrp: Figs. 47B and 47C) . . . . *Conostigmus
missyhazenae* Mikó and Trietsch sp. nov.

**aa.** Head transverse, distinctly wider than long in dorsal view

**bb.** OOL about 2× as long as LOL (Figs. 41A and 41B)

**cc.** Preoccipital sulcus present (pos: Figs. 41A, 41B, 39A and 39B)

**dd.** Lateral margins of harpe gradually converging distally, widest point of harpe is at
its articulation site with gonostyle-volsella complex (Figs. 30A, 36A and 52A) . . **19**

19. **a.** Cupula as long as gonostyle-volsella complex (cup: Fig. 36A)

**b.** Distal 3–4 setae in dorsal region of sensillar ring of harpe oriented distodorsally
(Figs. 36A–36C)

**c.** Distal margin of S9 concave

**d.** Distal end of dorsomedial conjunctiva of gonostyle-volsella complex not extending
1/2 of length of gonostyle-volsella complex (dc: Fig. 36C)

**e.** Parossiculus with two parossicular setae (pss: Fig. 36D) . . . . . . . . . *Conostigmus
macrocupula* Mikó and Trietsch sp. nov.

**aa.** Cupula at least 1/2 of gonostyle (Figs. 21A and 21B)

**bb.** Setae of sensillar ring of harpe oriented distomedially (Figs. 21 and 43)

**cc.** Distal margin of S9 convex

**dd.** Distal end of dorsomedial conjunctiva of gonostyle-volsella complex
extending 2/3 of length of gonostyle-volsella complex (dc: Fig. 43A)

**ee.** Parossiculus with one parossicular seta (pss: Fig. 43) . . . . . . . . . . . . . . . . . **20**

20. **a.** Mandible with one tooth (Figs. 18A and 18B)

    **b.** Setae on antenna shorter than or as long as width of flagellomeres (Fig. 40A)

    **c.** Proximolateral projection of cupula blunt (Fig. 21A)

    **d.** Proximodorsal notch of cupula v-shaped (notched) (pdc: Fig. 21C)

    **e.** Distal end of dorsomedial conjunctiva of gonostyle/volsella complex blunt (dc: Fig. 21C)

    **f.** Distal margin of harpe in lateral view acute (hrp: Fig. 21C) . . . . . ***Conostigmus fianarantsoaensis*** Mikó and Trietsch sp. nov.

    **aa.** Mandible with two teeth (Figs. 37A and 37B)

    **bb.** Setae on antenna longer than width of flagellomeres (Fig. 20A)

    **cc.** Proximolateral projection of cupula acute (ppc: Fig. 43B)

    **dd.** Proximodorsal notch of cupula arched (pdc: Fig. 43)

    **ee.** Distal end of dorsomedial conjunctiva of gonostyle-volsella complex acute (dc: Fig. 43A)

    **ff.** Distal margin of harpe in lateral view blunt (hrp: Fig. 43C) . . . . ***Conostigmus madagascariensis*** Mikó and Trietsch sp. nov.

## DISCUSSION

### Latitudinal diversity gradient and Malagasy *Conostigmus*

Including our data, almost an order of magnitude more *Conostigmus* species have been described from the Holarctic (n = 125) than from the Afrotropical region (n = 13) (*Johnson & Musetti, 2004*; *Dessart, 1997*). This biodiversity pattern suggests that *Conostigmus* joins other taxa known to be exceptions to the typical LDG: mollusks (*Valdovinos, Navarrete & Marquet, 2003*), nematodes (*Lambshead et al., 2000*), fig wasps (Agaonidae; *Hawkins & Compton, 1992*), galling insects (*Price et al., 1998*), bees (Anthophila; *Michener, 1979*), sawflies ("Symphyta"; *Kouki, Niemelä & Viitasaari, 1994*), some Lepidoptera (*Holloway, 1987*), psyllids and aphids (*Dixon et al., 1987*; *Eastop, 1977*; *Eastop, 1978*).

Ichneumonidae and Braconidae were also considered as exceptions to the typical latitudinal diversity gradient (*Gauld, 1986*; *Owen & Owen, 1974*; *Quicke & Kruft, 1995*). However, recent papers clearly show that the unusual distribution of these taxa are likely due to lack of comprehensive data from tropical regions (*Quicke, 2012*; *Veijalainen et al., 2012*).

*Noyes' (1989)* survey of two similarly-sized countries supports the validity of reverse LDG in Ceraphronoidea. Standardized sampling of the megaspilids of Sulawesi and Great Britain revealed a much higher diversity in the temperate (69 spp.) than in the tropical (nine spp.) region, as determined by Paul Dessart.

Deviation from the LDG in Ceraphronoidea has been only superficially examined, however, and could result from sampling bias. The only taxonomic review of *Conostigmus* species was published by *Dessart (1997)*. He treated the faunas of Africa, Asia, and Australia and examined 145 specimens compared to the many hundreds if not thousands of specimens examined for Palearctic species. Of the 36 species, nineteen are known

exclusively by holotypes, eight by the holotype and one paratype, and only one species (*Conostigmus canariensis*) was based on more than 10 specimens.

The present revision focuses solely on Malagasy *Conostigmus* and is based on observations of 159 specimens representing 12 species, more than five times as many as the earlier recorded *Conostigmus* species from Madagascar (*Dessart, 1997*). This species number is still just a small fragment of the known Palaearctic *Conostigmus* species (n = 97; *Johnson & Musetti, 2004*) and one fourth of the number of species recorded from the Atlantic Archipelago (n = 44; *Broad & Livermore, 2014*), which is almost half the size of Madagascar (315,159 vs. 587,041 km$^2$). Considering that Madagascar is a biodiversity hotspot (*Myers et al., 2000*), our study lends support to the hypothesis that Megaspilidae show a reverse latitudinal biodiversity gradient.

## The single layer epithelium and body size polyphenism

Insects are epidermal organisms (*Locke, 1998*) and the single-layered epidermis is responsible for their tremendous phenotypic diversity. Epidermal cells produce the cuticle, the acellular exoskeleton that is the subject of most morphological descriptions in insect systematics (*Deans et al., 2012*). For instance, in the present paper we exclusively used cuticle-related phenotypes. The dominance of cuticular characters in insect systematics descriptions is easy to explain: besides the remnants of some skeletal muscles, the cuticle is perhaps the only component of an insect body that can be accurately studied even on an improperly fixed specimen. This resilient replica of the pupal epidermis can be studied on specimens that are millions of years old (*Carpenter, 1992*).

The epidermis arises exclusively and solely from imaginal disks. The growth of imaginal disks, and thus the final cell number and cell size of the epidermis, is regulated in collaboration by insulin and ecdysone (*Nijhout & Grunert, 2010*; *Nijhout et al., 2007*; *Nijhout & Callier, 2015*) that are controlled mostly by environmental factors, such as temperature, oxygen level and nutrition. Oxygen concentration and temperature mostly influence body size through cell growth (*Callier & Nijhout, 2014*; *Heinrich et al., 2011*; *Harrison & Haddad, 2011*; *Peck & Maddrell, 2005*; *Azevedo, French & Partridge, 2002*; *Partridge et al., 1994*) while nutrition level seems to impact cell number through regulating proliferation (*Emlen, Lavine & Ewen-Campen, 2007*; *Liu et al., 2015*).

Ceraphronoidea exhibit substantial body size polyphenism, which varies by almost a factor of two in some species (*Mikó et al., 2013*; *Fergusson, 1980*; *Liebscher, 1972*). This tendency is followed by Malagasy *Conostigmus*, for example the IOS (interorbital distance, an anatomical line between the medial eye margins that reflects body size) reveals a two-fold difference in *C. longulus* (138–263 μm). Body size polyphenism is usually induced by variability in host body size in polyphagous and brood size in gregarious parasitoid Hymenoptera (*Quicke, 1997*; *Nalepa & Grisell, 1993*; *Medal & Smith, 2015*). Numerous ceraphronoid species are known to parasitize hosts with variable body size (*Fergusson, 1980*; *Gilkeson, McLean & Dessart, 1993*) and gregariousness is not uncommon (*Cooper & Dessart, 1975*; *Starý, 1977*; *Liebscher, 1972*; *Mackauer & Chow, 2015*; *Takada, 1973*). *Mackauer & Chow (2015)* A clear relationship between ceraphronoid body mass

and brood size was recently shown in the facultatively gregarious *Dendrocerus carpenteri* (Curtis, 1829), where the body mass of a single solitary specimen did not differ from the combined body mass of two gregarious specimens *Mackauer & Chow (2015)*.

Information on *Conostigmus* biology is very limited, but body size of their hosts (Syrphidae and boreid mecopterans) certainly allows the development of multiple parasitoid specimens (*Dessart, 1980*; *Cooper & Dessart, 1975*; *Weems & Howard, 1954*; *Kamal, 1926*; *Ulber et al., 2010*; *Panis, 2008*). These data suggest that differences in ceraphronoid body size is nutrition dependent thus body size polyphenism is most likely related to differences in cell number.

Wing trichomes (http://purl.obolibrary.org/obo/HAO_0002454) have a one to one match to epidermal cells (*Dobzhansky, 1929*; *Stevenson, Hill & Bryant, 1995*; *Partridge et al., 1994*; *Heinrich et al., 2011*) and they were traditionaly used in comparative evo-devo studies to estimate cell density and size in different *Drosophila* mutant specimens (*Stern & Emlen, 1999*; *Emlen, Lavine & Ewen-Campen, 2007*; *Nijhout & Callier, 2015*).

Sculptural elements of the cuticle likewise correspond to the patterns and geometry of epidermal cells (*Wigglesworth, 1973*; *Locke, 1959*; *Locke, 1967*) and they have never been explored as a potential source for understanding cellular processes in the developing imaginal disks.

## The nature of scutes

In Malagasy *Conostigmus* species, the head and the mesosoma are covered with repetitive, usually hexagonal and isodiametric, 6.6–25 μm wide elements, referred to as scutes (*Cals, 1974*; *Moretto, Minelli & Fusco, 2015*) or sculpticells (*Allen & Ball, 1979*). Arthropod taxa often exhibit scutes (*Meyer, 1842*; *Cals, 1974*; *Krell, 1994*) that are considered to be ancestral sculpture elements in Insecta (*Hinton, 1970*). The surface morphology of scutes (convex vs. concave; Figs. 51A and 51B) and the depth of the impressions separating them (Figs. 50, 31A and 31B) are important for separating Malagasy *Conostigmus* species while differences in their superficial density (Figs. 24, 25A and 25B; they are less dense in smaller specimens) is perhaps the most obvious intraspecific trait.

Due to their hexagonal shape and size, it has long been speculated that scutes to reflect the surface of epidermal cells (*Kölliker, 1856*; *Warren, 1903*). *Fusco, Brena & Minelli (2000)* studied the correspondence between scutes and epidermal cells in subsequent instars of lithobiomorph centipedes and demonstrated a one to one match between the cells and scutes. *Hinton (1970)*, *Cals (1973)*, *Cals (1974)* and *Blaney & Chapman (1969)* likewise found correspondences between the number of epidermal cell nuclei in mature adults and scutes in different insect groups, but *Blaney & Chapman (1969)* found 1–2 percent fewer epidermal cells than scutes and explained this discrepancy by ecdysial cell death based on the presence of some degraded cell nuclei. One-to-one correspondence between scutes and epidermal cells is also supported by the fact that elongate scutes correspond to elongate epidermal cells (*Hinton, 1970*).

*Locke (1959)*, *Locke (1967)* and *Wigglesworth (1973)* performed detailed histological and developmental studies to reveal the cellular origin of stellate folding, ripple patterns,

dome-like plaques, and setal pits and revealed that these structures are the product of multiple epidermal cells. Unfortunately, the relationships between scutes and epidermal cells have never been proved by similarly detailed examinations.

### Bigger cells or more cells?

Although the relationship between scutes and epidermal cells has been broadly acknowledged in insect systematics (*Ball, 1985*; *Allen & Ball, 1979*; *Burks et al., 2013*; *Krell, 1994*), no one has used this knowledge to understand body size polyphenism. According to our findings, scute size is independent of body size (*i.e.* the epithelium of smaller specimens is built by proportionally less scutes than that in larger specimens). The number of scutes along the IOS (interorbital space, shortest distance between compound eyes) of a smaller specimen is half the number of scutes along the same line in a specimen with an IOS two times as long (Figs. 31A and 31B).

It follows that there is no difference in epidermal cell size of the smaller and the larger specimens and therefore cell number differences must contribute exclusively to body size polyphenism in Malagasy *Conostigmus longulus Dessart, 1997* specimens. Based on our collective understanding of underlying developmental processes, the size difference in *Conostigmus longulus* is likely related to nutritional differences that likely result from the complexities of polyphagy and gregariousness.

We observed a substantial intraindividual variation in scute morphology: scutes and cell size on the frons are smaller than that on the mesoscutellum. This variation might reflect the difference in the growth of the head and wing imaginal disks contributing to allometric changes.

Intraspecific differences in body size often impact species diagnoses. Statements, such as "smaller specimens can be very difficult, if not impossible, to identify correctly because the morphology of typical specimens is not expressed" (*Al Khatib et al., 2014*: p. 809) and "in smaller specimens, the characters are subdued" (*Smith, 2012*: p. 215) are common in taxonomic descriptions and often refer cuticular specializations, such as carinae or grooves. Despite the importance of allometric reductions, developmental causes of these phenomena have never been revealed. In *Conostigmus longulus* the transverse carina of the frons is less expressed in smaller specimens (tfc: Figs. 22A and 22B), encumbering their identification. The carina is the product of the concerted action of 52 epidermal cells (26 columns in two rows) in small specimens and 156 cells (52 columns in three rows) in large specimens (Figs. 22A and 22B) suggesting that allometric reduction of cuticular specializations might be related to cell number and that more epidermal cells are able to produce more conspicuous structures. In this respect, the impact of epidermal cell density on the distinctness of cuticular specialization might be similar to the impact of pixel density on the resolution of digital images; one can see more details on an image with 1,200 dpi than on one with 256 dpi resolution.

### CONCLUSIONS

Our data suggest that *Conostigmus* show a reverse latitudinal biodiversity gradient, but we acknowledge that *Conostigmus* in the temperate zone remains poorly understood

(e.g., types of half the described Holarctic species are missing (*Johnson & Musetti, 2004*)). Species concepts are also based strictly on morphological data, which, for some taxa, can mask true species-level diversity (*Smith et al., 2008*).

The correspondence between scutes and epidermal cells has already been proved by developmental studies in centipedes (*Moretto, Minelli & Fusco, 2015*), but we need to validate this relationship in insects. To understand spatial relationships between cellular and subcellular components of the epithelium is now easier to achieve with the advent of contemporary 3D reconstruction techniques such as CLSM or serial block face scanning electron microscopy. Ceraphronoidea would be an especially feasible model for this kind of examination since the head and the mesosoma are almost uniformly covered with scutes, and it is relatively easy to establish sustainable colonies of multiple species (*Araj et al., 2006*; *Chow & Mackauer, 1999*). *Dendrocerus carpenteri* is facultatively gregarious (*Mackauer & Chow, 2015*) with brood size varying between one and three larvae making this taxon feasible even for simultaneous analyses of nutrition, oxygen level and temperature dependence of epidermal development.

Being able to understand cellular processes in the developing epithelium of adult insects by reading sculptural elements can provide invaluable information about the influence of environmental factors on allometric differentiation. Sculpture is not only one of the most important traits for insect classification, it also conserves the history of developmental processes in the single cell thick epithelium accountable for the tremendous morphological diversity in arthropods. Sculpture also remains available as a source of biological information long after a specimen has been collected or preserved as a fossil.

Therefore we believe that sculpture, a witness to developmental and evolutionary history of arthropods, could serve as a messenger between morphology based classical arthropod taxonomy and the 21st century insect ecology, evolutionary biology, and cell and developmental biology.

## ACKNOWLEDGEMENTS

We thank Brian Fisher, Charles Griswold, and Darryll Ubick, California Academy of Sciences, San Francisco for making this material available for study, Missy Hazen (Penn State Huck Institute of the Life Sciences, Microscopy and Cytometry Facility) for her assistance in Confocal Laser Scanning Microscopy and James Balhoff for his help in generating semantic statements in OWL. Stéphane Hanot, Akice-Marie Buset and Didier Van den Spiegel kindly sent us type specimens of the species deposited in Tervuren (Royal Museum of Central Africa, Belgium). Gavin Broad and Maciej Krzyżyński are also acknowledged for their valuable comments on the first draft of this manuscript.

### Funding

This material is based upon work supported by the U.S. National Science Foundation, under Grant Nos. DBI-0850223, DEB-0956049, DBI-1356381, DEB-1353252 and

Grant No. DEB-0072713 to B.L. Fisher and C.E. Griswold and Grant No. DEB-0344731 to B.L. Fisher and P.S. Ward. The funders had no role in study design, data collection and analysis, decision to publish, or preparation of the manuscript.

### Grant Disclosures

The following grant information was disclosed by the authors:
U.S. National Science Foundation: DBI-0850223, DEB-0956049, DBI-1356381, DEB-1353252.
B.L. Fisher and C.E. Griswold: DEB-0072713.
B.L. Fisher and P.S. Ward: DEB-0344731.

### Competing Interests

The authors declare that they have no competing interests.

### Author Contributions

- István Mikó conceived and designed the experiments, performed the experiments, analyzed the data, wrote the paper, prepared figures and/or tables.
- Carolyn Trietsch conceived and designed the experiments, performed the experiments, analyzed the data, wrote the paper, prepared figures and/or tables.
- Emily L. Sandall conceived and designed the experiments, performed the experiments, analyzed the data, wrote the paper, prepared figures and/or tables.
- Matthew Jon Yoder contributed reagents/materials/analysis tools, reviewed drafts of the paper.
- Heather Hines contributed reagents/materials/analysis tools.
- Andrew Robert Deans contributed reagents/materials/analysis tools, reviewed drafts of the paper.

### Data Deposition

GitHub: https://github.com/hymao/hymao-data/blob/master/miko2016_malagasy.owl.

### New Species Registration

The following information was supplied regarding the registration of a newly described species:
Publication:
urn:lsid:zoobank.org:pub:41133330-D364-4ABF-AFB3-DEE013E0EDF9.
Conostigmus bucephalus Mikó and Trietsch
LSID: urn:lsid:zoobank.org:act:FC799B3D-624D-4872-8F3E-0D4B7B0F6473.
Conostigmus clavatus Mikó and Trietsch
LSID: urn:lsid:zoobank.org:act:64A99084-8B0F-44E2-97B2-1B05E655606D.
Conostigmus fianarantsoaensis Mikó and Trietsch
LSID: urn:lsid:zoobank.org:act:040F2E40-7D7C-4124-98AC-36F300608E66.
Conostigmus lucidus Mikó and Trietsch
LSID: urn:lsid:zoobank.org:act:4FA60414-40DD-4195-BB26-B5CD3D6BAB8D.

Conostigmus macrocupula Mikó and Trietsch
LSID: urn:lsid:zoobank.org:act:CC00A26A-3FED-4614-B36F-D42D5AF92FF9.
Conostigmus madagascariensis Mikó and Trietsch
LSID: urn:lsid:zoobank.org:act:3669CA28-6149-4F9F-A7C0-9E64130DD406.
Conostigmus missyhazenae Mikó and Trietsch
LSID: urn:lsid:zoobank.org:act:6E1A4A22-B02D-4A7D-BBBD-D4C77681FB82.
Conostigmus pseudobabaiax Mikó and Trietsch
LSID: urn:lsid:zoobank.org:act:16A49F11-949C-4336-9894-7BB02916124D.
Conostigmus toliaraensis Mikó and Trietsch
LSID: urn:lsid:zoobank.org:act:77C12D21-D99E-4EF6-9710-2D2473BF7B15.

## Supplemental Information

Supplemental information for this article can be found online at http://dx.doi.org/
10.7717/peerj.2682#supplemental-information.

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
