# Peer review of "Malagasy Conostigmus (Hymenoptera: Ceraphronoidea) and the secret of scutes"

_PeerJ, doi:10.7717/peerj.2682_

## Round 0.1 · original submission · Minor Revisions

Dear István,

I'm glad to let you know that your article received very nice comments, as it represents an innovative and comprehensive example of a scientific work (in the era of “salami slicing”, i.e. data fragmentation).

The article contains a small number of easily correctable errors including grammar, missing references, and minor content clarification. Therefore my decision is “minor revisions”.

·

Basic reporting

Mostly great. Some minor problems with the English.

Experimental design

No problems at all. Innovative use of imaging technology to approach taxonomic problems.

Validity of the findings

No problems at all.

Additional comments

This is a lovely, innovative taxonomic paper, the first I am aware of looking at intraspecific variation at the level of epidermal cells. It is about time that more attention was given to ceraphronoids. The illustrations are beautiful and really liberate these Conostigmus from their usual fate of being small, dull wasps that are ignored. I do have some minor comments and corrections that I hope will be of use.

Sorry for the pedantic nature of many of the textual corrections. I just couldn’t stop myself. But there were a surprising number of mistakes in spelling and grammar.

There is no diagnosis of Conostigmus; some notes on how to recognise the genus in the Malagasy fauna is surely essential, given how poorly known ceraphronoids are.

The beginning of the discussion is inaccurate. You can’t state that Ichneumonidae and Braconidae are ‘known to be exceptions to the typical latitudinal diversity gradient’ when there are recent papers clearly showing that these ‘exceptions’ are likely due to lack of good data. See recent papers by Quicke (2012) and Veijalainen et al. (2012). It is the case for various groups, including Ichneumonoidea and Ceraphronoidea, that described species richness is a poor proxy for actual species richness.

Ceraphronoidea are not simply ‘ectoparasitoids’; a variety of life history strategies have been reported and it’s clear that this is a biologically very interesting but under-explored superfamily. As far as I am aware, all Megaspilidae, where known, are ectoparasitoids (e.g. Haviland, 1920), but Ceraphronidae includes endoparasitoids, with both ectoparasitoids and endoparasitoids apparently within the same genus (e.g. Aphanogmus: Parnell, 1963; Luhman et al., 1999; Evans et al., 2005).

Abbreviations of anatomical structures (i.e. morphological terminology) shouldn’t be in a supplementary file; I would argue these are essential to understand the paper. PeerJ is not very helpful in providing a PDF that treats text as images.

In species diagnoses, you should state whether one sex is unknown, as this effects the ability to diagnose the species. For example, C. macrocupula is diagnosed only by the long cupula, and is known only from males. Would a female be recognisable?

Some of the terminology is confusing, such as whether the notch of a cupula is notched… There is no guidance on the figures as to what this means. I was trying to work out the male genitalia terminology from fig. 43 but I don’t know if ‘pnc’ is supposed to be ‘pdc’. It doesn’t look like much of a notch.

Contrary to your opening sentence of the conclusions (line 1476), I don’t believe your data do say much about a latitudinal diversity gradient for Megaspilidae. You’ve only looked at one genus, albeit the largest genus. You don’t present wider data on the faunas of Dendrocerus or other, smaller genera.

Evans, G. A., P. Dessart and H. Glenn. 2005. Two new species of Aphanogmus (Hymenoptera: Ceraphronidae) of economic importance reared from Cybocephalus nipponicus (Coleoptera: Cybocephalidae). Zootaxa 1018: 47-54.
Haviland, M. D. 1920. On the bionomics and development of Lygocerus testaceimanus Kieffer, and Lygocerus cameroni Kieffer, (Proctotrypoidea-Ceraphronidae) parasites of Aphidius (Braconidae). Quarterly Journal of Microscopical Science 65: 101-127.
Luhman, J. C., R. W. Holzenthal and J. K. Kjaerandsen. 1999. New host record of a ceraphronid (Hymenoptera) in Trichoptera pupae. Journal of Hymenoptera Research 8: 126.
Parnell, J. R. 1963. Three gall midges (Diptera: Cecidomyidae) and their parasites found in the pods of broom (Sarothamnus scoparius (L.) Wimmer). Transactions of the Royal Entomological Society of London 115: 261-275.

Quicke, D. L. J. 2012. We know too little about parasitoid wasp distributions to draw any conclusions about latitudinal trends in species richness, body size and biology. PLoS ONE 7: 9 pp. doi: 10.1371/journal.pone.0032101

Veijalainen, A., Wahlberg, N., Broad, G. R., Erwin, T. L., Longino, J. T. and Sääksjärvi, I. E. 2012. Unprecedented ichneumonid parasitoid wasp diversity in tropical forests. Proceedings of the Royal Society of London Series B - Biological Sciences 279: 4694–4698.

·

Basic reporting

Submitted manuscript is consistent with PeerJ policies and scientific standards. Authors provided all needed information about hardware and software they used (with hyperlinks to the source if possible), also gave precise information about where (in which collection) specimens of Conostigmus were finally placed. Raw data is put on the Figshare. English used by the Authors is clear. Figures are relevant, well described and labeled. The quality of photographs reaches known limits of microhymenopteran photography. However disparate figures in submitted .pdf file and as source files, labeled as figures 1-6 are bothering me.

Experimental design

The submission fits the Scope of PeerJ journal. Also methods are described adequately and one should not have any issues with reproducing the studies.

Validity of the findings

No Comments

Additional comments

Authors provided rich submission with 9 new described species from Madagascar, revised genus Conostigmus occuring on the island, confirmed reverse Latitudinal Diversity Gradient in Conostigmus (and therefore in Megaspilidae family) and results of research on body size polyphenism. In the era of "salami slicing" and publishing less relevant data as possible, I find submitted manuscript as comprehensive example of scientific work.
Nevertheless there are still few lapses in the manuscript (labeled by a comment function in Adobe Reader, .pdf file attached) and I am recommending to straighten them.

---

## Round 0.2 · accepted · Accept

In my opinion, the manuscript is now suitable for publication in PeerJ as it is.